# Architecture, dynamics and biogenesis of GluA3 AMPA glutamate receptors

Aditya Pokharna[1,6], Imogen Stockwell[1,6], Josip Ivica[1], Bishal Singh[1,2], Johannes Schwab[3], Carlos Vega-Gutiérrez[4], Beatriz Herguedas[4], Ondrej Cais[1], James M. Krieger[5] & Ingo H. Greger[1✉]

AMPA-type glutamate receptors (AMPARs) mediate the majority of excitatory neurotransmission in the brain[1]. Assembled from combinations of four core subunits, GluA1–4 and around 20 auxiliary subunits, their molecular diversity tunes information transfer and storage in a brain-circuit-specific manner. GluA3, a subtype strongly associated with disease[2], functions as both a fast-transmitting $Ca^{2+}$-permeable AMPAR at sensory synapses[3], and as a $Ca^{2+}$-impermeable receptor at cortical synapses[4,5]. Here we present cryo-electron microscopy structures of the $Ca^{2+}$-permeable GluA3 homomer, which substantially diverges from other AMPARs. The GluA3 extracellular domain tiers (N-terminal domain (NTD) and ligand-binding domain (LBD)) are closely coupled throughout gating states, creating interfaces that impact signalling and contain human disease-associated mutations. Central to this architecture is a stacking interaction between two arginine residues (Arg163) in the NTD dimer interface, trapping a unique NTD dimer conformation that enables close contacts with the LBD. Rupture of the Arg163 stack not only alters the structure and dynamics of the GluA3 extracellular region, but also increases receptor trafficking and the expression of GluA3 heteromers at the synapse. We further show that a mammalian-specific GluA3 trafficking checkpoint determines the conformational stability of the LBD tier. Thus, specific design features define communication and biogenesis of GluA3, offering a framework to examine this disease-associated glutamate receptor.

AMPARs form both homomeric and heteromeric receptor complexes that differ in their distribution, organization and function within the brain[1,6]. These receptors can be segregated into two main subgroups—either permeable or impermeable to calcium ions. $Ca^{2+}$-impermeable (CI) AMPARs are far more abundant throughout the forebrain, and contain the GluA2 subunit that governs $Ca^{2+}$ permeability, subunit assembly and receptor structure. In CI-AMPAR tetramers, GluA2 preferentially occupies the inner B/D positions, giving rise to an interface between its NTDs[7,8] (Fig. 1a). The resulting compact NTD tier enables receptor localization at the synapse concomitant with efficient synaptic transmission[8–10].

By contrast, $Ca^{2+}$-permeable (CP) AMPARs are abundant in subcortical structures and in interneurons[1]. Their $Ca^{2+}$ signal mediates synaptic plasticity[11,12], but is also closely associated with various diseases[1,13]. In contrast to GluA2-containing receptors, CP AMPARs are structurally poorly defined, and are subject to different assembly and synaptic localization mechanisms[8,14]. CP AMPARs segregate into the slowly gating GluA1[14], and the fast GluA3 and GluA4 receptors[1], selectively enriched at sensory synapses in the thalamus and brain stem[3,15–19]. However, contrary to GluA4, GluA3 is present in both cortical CI and subcortical CP AMPARs.

GluA3 homomers are thought to localize to the auditory nerve (endbulb of Held) synapse[18], where their fast gating characteristics enable auditory signalling and sound localization[3,15,18,20]. CI GluA3 heteromers exhibit different trafficking and synaptic plasticity mechanisms compared with CI GluA1 heteromers[4,5]. GluA3 is X-chromosome linked, and its mutations contribute to multiple disorders including epilepsy, intellectual disability, aggression and schizophrenia[2,21–24], rendering this AMPAR subunit a central but structurally poorly defined drug target. Here we determined cryo-electron microscopy (cryo-EM) structures of GluA3 homomers in different gating states, which, combined with simulations and functional data in neurons, shed light on the operational principles of this central AMPAR subtype.

## Organization and dynamics of apo-state GluA3-G

To resolve the organization of GluA3 AMPARs, we performed cryo-EM structural analysis of the GluA3 flip splice form (RNA-edited at the R/G site (Gly747)) associated with the auxiliary subunit TARP-γ2, an AMPAR combination enriched at auditory synapses[25] (Extended Data Fig. 1 and Supplementary Table 1), according to published procedures[14,26]. GluA3 exists as two variants in vertebrates, GluA3(Gly439) and GluA3(Arg439) (hereafter, GluA3-G and GluA3-R) (Extended Data Fig. 2a). GluA3-R is unique to mammals and, in contrast to GluA3-G, exhibits poor secretory trafficking[27]. We determined the structures of both variants, providing insights into GluA3 architecture and control of secretory traffic (Extended Data Figs. 1–3 and 6).

[1]Neurobiology Division, Medical Research Council (MRC) Laboratory of Molecular Biology, Cambridge, UK. [2]Department of Genomic Medicine, University of Texas MD Anderson Cancer Center, Houston, TX, USA. [3]Structural Studies Division, Medical Research Council (MRC) Laboratory of Molecular Biology, Cambridge, UK. [4]Institute for Biocomputation and Physics of Complex Systems and Laboratory of Advanced Microscopies, University of Zaragoza, Zaragoza, Spain. [5]Biocomputing Unit, National Center of Biotechnology, CSIC, Madrid, Spain. [6]These authors contributed equally: Aditya Pokharna, Imogen Stockwell. ✉e-mail: ig@mrc-lmb.cam.ac.uk

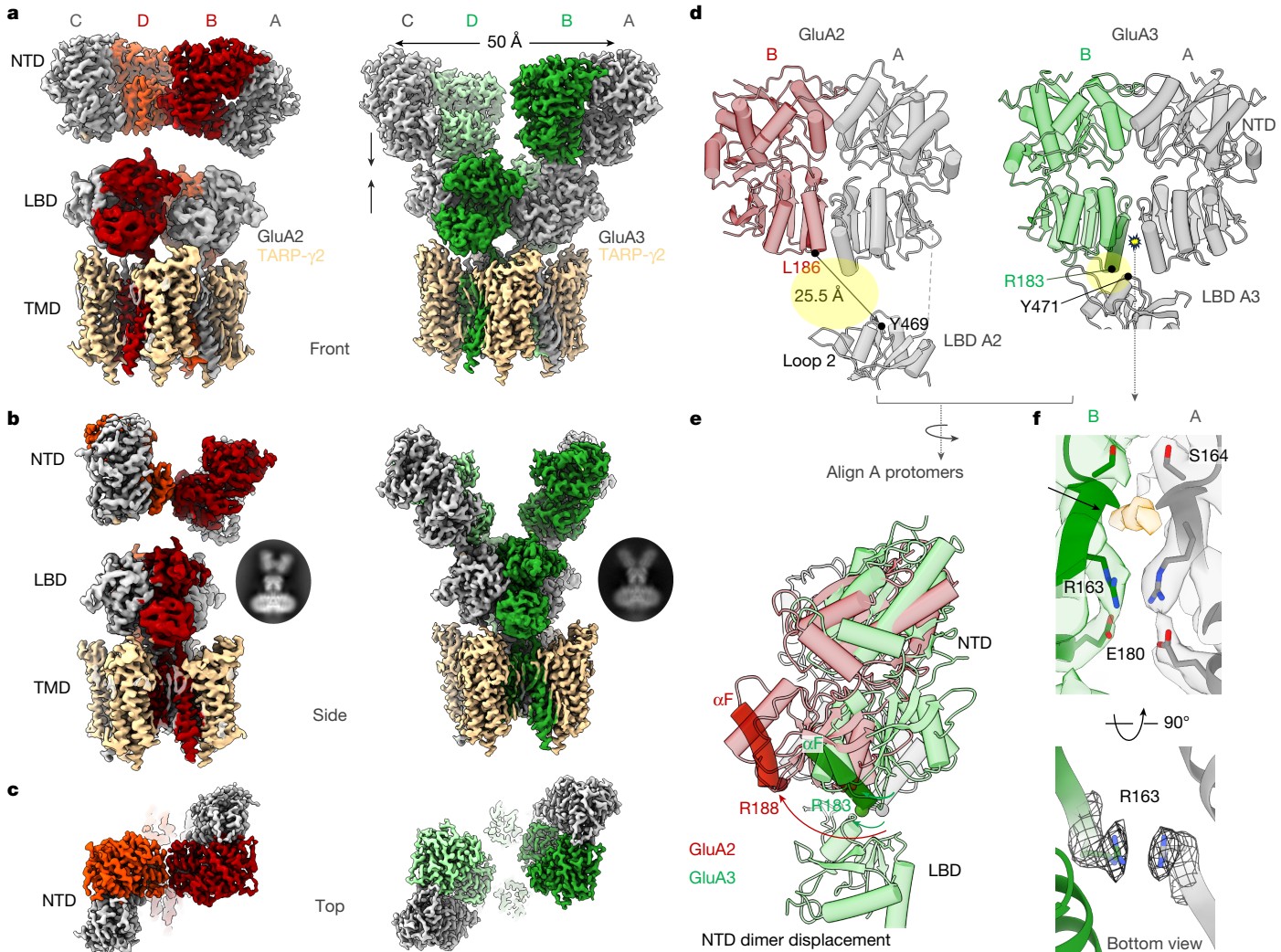

**Fig. 1 | The architecture of the GluA3–TARP-γ2 complex. a**, Composite cryo-EM maps of GluA2–TARP-γ2 (left) and GluA3-G–TARP-γ2 (right); the three domain layers are indicated on the left, and the individual chains (A–D) are indicated at the top. Chains A/C are shown in grey, and chains B/D are shown in red (GluA2) and in green (GluA3). The NTD tier is expanded by 23 Å relative to GluA2 (centre of mass between the A and C chains), and interfaces with the A/C chain LBDs. TMD, transmembrane domain. **b**, Side view of the GluA2–TARP-γ2 (left) and GluA3–TARP-γ2 (right) maps, showing the interface between the B/D chain NTDs that exist only in GluA2. Insets: typical 2D class averages. **c**, Top view of the GluA2 (left) and GluA3 (right) receptor complexes. **d**, Models of the A/B chain NTD dimers, illustrating the rightward translation of the GluA3 B-chain NTD (right) towards the A-chain LBD. Compared with GluA2 (PDB: 9B68), this leads to a 25.5 Å approximation of the Arg183 and Tyr471 GluA3 marker residues (equivalent to Leu186 and Tyr469 in GluA2). **e**, Superposition of the A-chain NTDs of GluA2 and GluA3 reveals the displacement of the GluA2 B-chain NTD relative to GluA3. This motion is highlighted (αF). The curved arrows mark the shift of GluA2 Arg188 relative to GluA3 Arg183 (spheres). The rotation would disengage the GluA2 NTD from interfacing with the LBD. **f**, Magnified view of the lower part of the GluA3 NTD dimer interface, showing the stacking interaction between the Arg163 side chains across the interface, which is enabled by the Glu180 countercharges (top). A ligand density above the Arg-stack is shown in yellow and is coordinated by the Ser164 main chain. Bottom, bottom view of the Arg163 stack.

GluA3-G–TARP-γ2 determined in the apo state diverges from existing AMPAR structures in its sequence-diverse NTD tier (Fig. 1a). Similar to GluA1 receptors[14], GluA3 lacks the interface between the inner B and D subunit NTDs that is typically observed in GluA2 (Fig 1a); as a result, the two GluA3 NTD dimers are separated. Furthermore, the GluA3 NTDs stack onto the LBDs—a close apposition of NTD and LBD is apparent throughout 3D classes, where we observe receptors with two upright NTD dimers, both interfacing with the LBD, or those with one upright dimer and a second one bending towards the LBD (Extended Data Fig. 2b). Bending motions occurred to various degrees and in NTD dimers not coupled to the LBD, suggestive of a conformational continuum.

To interrogate the conformational spectrum of the atypical GluA3 structure, we used DynaMight, a machine-learning approach designed to address continuous structural heterogeneity[28]. DynaMight generates a conformational landscape with a central region corresponding to a dominant conformation, and less-populated conformations in the periphery. GluA3 with two 'double upright' NTD dimers accumulated in the centre of the conformational landscape, while receptors with bent NTDs localized to the periphery (Extended Data Fig. 2c,d and Supplementary Video 1). Overall, receptors with two upright NTD dimers appear to be a prevalent GluA3 conformation.

## The GluA3 NTD interfaces with the LBD

We constructed a double-upright GluA3-G–TARP-γ2 receptor to a resolution of 2.5–3 Å (Fig. 1a–c and Extended Data Figs. 1 and 3a). The NTD tier was horizontally expanded by more than 23 Å (centre of mass) between the outer A/C-chain NTDs in contrast to GluA2 homomers (Protein Data Bank (PDB): 9B68)[26] or to the native GluA2/3 heteromer

(PDB: 6NJM)[29] (Extended Data Fig. 4a). Moreover, the NTD closely apposed the LBD, giving rise to a previously unseen NTD–LBD interface. Interface contacts are established exclusively with the LBDs of the A/C subunits (but not with the gating-dominant B/D LBDs), resulting in an outward translation of the NTD dimers that underlies the expanded appearance of the receptor (Fig. 1a,d). We measured a >20 Å translation of the inner NTDs towards an A/C LBD, when comparing GluA3 with either a GluA2 homomer or a GluA2/3 heteromer, using the GluA3 marker residues Arg183 (NTD) and Tyr471 (LBD) and the corresponding markers in GluA2 (Fig. 1d and Extended Data Fig. 4a).

When tracing the origin of this NTD–LBD coupling, we noticed an atypical, flat organization of the GluA3 NTD dimers that is not seen with any other AMPAR. This is evident after superposition of the GluA3 and GluA2 NTD dimers using the A protomers, showing an approximately 30° displacement of GluA2 relative to GluA3 (protomers B; Fig. 1e). NTD dimer displacement is conserved across AMPARs, including the GluA2–GluA3 heterodimer[29,30] (Extended Data Fig. 4b,c). Critically, the flat dimer conformation enables contacts of both GluA3 NTD protomers with an A/C LBD (Fig. 1d,e and Supplementary Video 2).

GluA3 exhibits the weakest NTD dimer interface among AMPARs[31,32], leading to a range of dimer conformations (see below), which have been ascribed to charge repulsion between the side chains of Arg163 projecting into the dimer interface[33] (Fig. 1f). This interfacial arginine is unique to GluA3 and is replaced by hydrophobic residues in all other AMPARs (Extended Data Fig. 4d), yielding displaced dimers of higher affinity[31,32].

In the intact GluA3 receptor, the Arg163 side chains stack through their guanidinium groups, with an inter-Cζ distance of about 3.5 Å (Fig. 1f). This unusual interaction is supported by a charge-compensating Glu180 together with a surrounding solvent network, and a ligand density coordinated by the Ser164 main chain, which we modelled as a chloride ion (Extended Data Fig. 4e). Together, these features hold the GluA3 dimer in a flat conformation, yielding an extended surface of the NTD base for association with the A/C LBDs (Fig. 1a,b,d).

## The GluA3 NTD–LBD interface

Focusing onto the A/B NTD dimer, the hydrophilic NTD–LBD interface (Extended Data Fig. 4f) is formed by the base of the NTD of chain B with loop 2 of the LBD (Fig. 2a). Arg183 in the NTD of chain A forms a salt bridge with Asp475 on the LBD, a contact that is not possible with a displaced NTD dimer conformation (Fig. 1e). Moreover, the NTD–LBD linker of chain A projects Arg395 towards LBD Asp475, further knitting the two domains together.

Rapid synaptic transmission can be mimicked with high-frequency glutamate pulses applied to AMPARs in excised HEK293 cell patches. GluA3–TARP-γ2 response amplitudes remained largely constant throughout a 100 Hz train, in marked contrast to GluA1–TARP-γ2 (Fig. 2b). This depression in GluA1 largely results from slow desensitization recovery, which is linked to its highly dynamic NTDs[14]. To assess whether NTD–LBD coupling influences GluA3 gating, we first introduced a stabilizing disulfide bridge between NTD Lys129 and LBD Glu458 (Fig. 2a). This bridge slowed entry into desensitization and increased equilibrium currents, which was not apparent with the single mutants K129C or E458C (Fig. 2c and Extended Data Table 1). Moreover, stabilizing the interdomain interface further facilitated the frequency response in a 100 Hz train compared with the GluA3 wild type (Fig. 2d). Conversely, perturbing the NTD–LBD interface with an N-glycan at Gln185 (Q185glyco = Q185N K187T) sped desensitization entry and slowed desensitization recovery (Extended Data Table 1), culminating in attenuated response amplitudes throughout the train (Fig. 2d). Thus, NTD–LBD coupling in GluA3 facilitates fast signal transmission.

Dysfunctional GluA3 is often associated with disease[1,2,21–24]. Disease mutations are scattered across the NTD and LBD surface, and locate to the NTD–LBD interface, such as to Arg188 (R188Q) in the NTD and Arg473 (R473T) in the LBD[34] (Extended Data Fig. 5a). These two mutations had a subtle kinetic phenotype when expressed with TARP-γ2 in HEK293 cells; Arg473 mutated to either Thr or Ala slowed recovery from desensitization (Extended Data Table 1), and reduced response amplitudes, suggesting a trafficking defect. Additional mutations map to the NTD dimer interface and throughout the ion channel[34], as well as to an interface uniquely appearing in desensitized GluA3 (Extended Data Fig. 5a,b; see below). Our structures provide a template to further study GluA3 disease mechanisms.

## Heterogeneity of ER-retained GluA3-R

Contrary to GluA3-G, the mammalian GluA3-R isoform is retained in the endoplasmic reticulum (ER), and is released either by assembly into heteromers or by mutation of Arg439 in the LBD to Gly[27,35]. Fluorescence-activated cell sorting (FACS) analysis confirmed ER retention of GluA3-R, while both GluA3-G and GluA2 strongly accumulate at the cell surface (Extended Data Fig. 6a). To assess the basis for this phenotype, we determined a cryo-EM structure of GluA3-R–TARP-γ2. In contrast to GluA3-G, apo state GluA3-R was prone to aggregation (Extended Data Fig. 6b) and exhibited substantial structural heterogeneity in its LBD tier, where dissociation of LBD dimers into highly mobile monomers was common throughout 3D classes (Extended Data Fig. 6c–e). Given the distance constraints, LBD mobility is unlikely to result from direct charge repulsion between the Arg439 residues (Extended Data Fig. 6f). Further refinement of a 3D class (comprising about 25%) with two intact LBD dimers revealed stacked Arg163 side chains, and flat NTD dimers closely apposed to the LBD. Thus, NTD coupling to the LBD is apparent in both GluA3-G and GluA3-R (Extended Data Fig. 6g).

We surmise that the exposed LBD dimer interfaces in GluA3-R facilitate assembly into heteromers. This is demonstrated by patch-clamp recordings of heteromers in HEK293 cells. Even with a fourfold excess of GluA3-R over GluA2 (4:1 plasmid ratio), the currents are mostly carried by non-rectifying GluA2–GluA3-R heteromers. Whereas, at the same ratio, both GluA3-G and GluA1 form homomers, reported by the dominance of rectifying responses (Extended Data Fig. 6h,i).

## NTD–LBD coupling occurs in activated GluA3

AMPAR activation is initiated by agonist associating with the A/C LBD clamshells[36], which interface with the NTD in the resting-state GluA3 (Fig. 1a). To assess this gating transition in GluA3-G, we captured the receptor in an open state (in the presence of L-glutamate and the desensitization blocker cyclothiazide, as described previously[37–39]) and monitored the conformational changes using standard 3D classification procedures (Extended Data Figs. 3b and 7a). Overall, we observed greater heterogeneity of both the NTD and LBD tiers compared with the apo state, with fewer classes exhibiting NTDs interfacing with the LBD (20% versus 45%). More frequent detachment of the NTD and LBD was also apparent through analysis with DynaMight (Extended Data Fig. 7b and Supplementary Video 3).

We further processed particles using a well-resolved NTD dimer (resulting from docking to the LBD), and these displayed clear features of activated AMPARs: closed LBD clamshells and a dilated gate that matches gate dimensions of other AMPAR subtypes (Extended Data Fig. 8a–e). We also observed selective interaction of the extracellular TARP β1 loop with the B/D LBDs (Extended Data Fig. 8a,b), while the other TARP pair targets the A/C KGK motif[40] through their β4 loop. These contacts are conserved in apo and active GluA3. Critically, the NTD–LBD interface of active GluA3-G closely resembled that of the apo-state receptor (root mean squared deviation (r.m.s.d.), 0.27 Å), as did the flat NTD dimer organization with stacked Arg163 side chains and putative chloride ligand (Extended Data Fig. 8f). The inter-tier coupling therefore occurs in both resting-state and active GluA3.

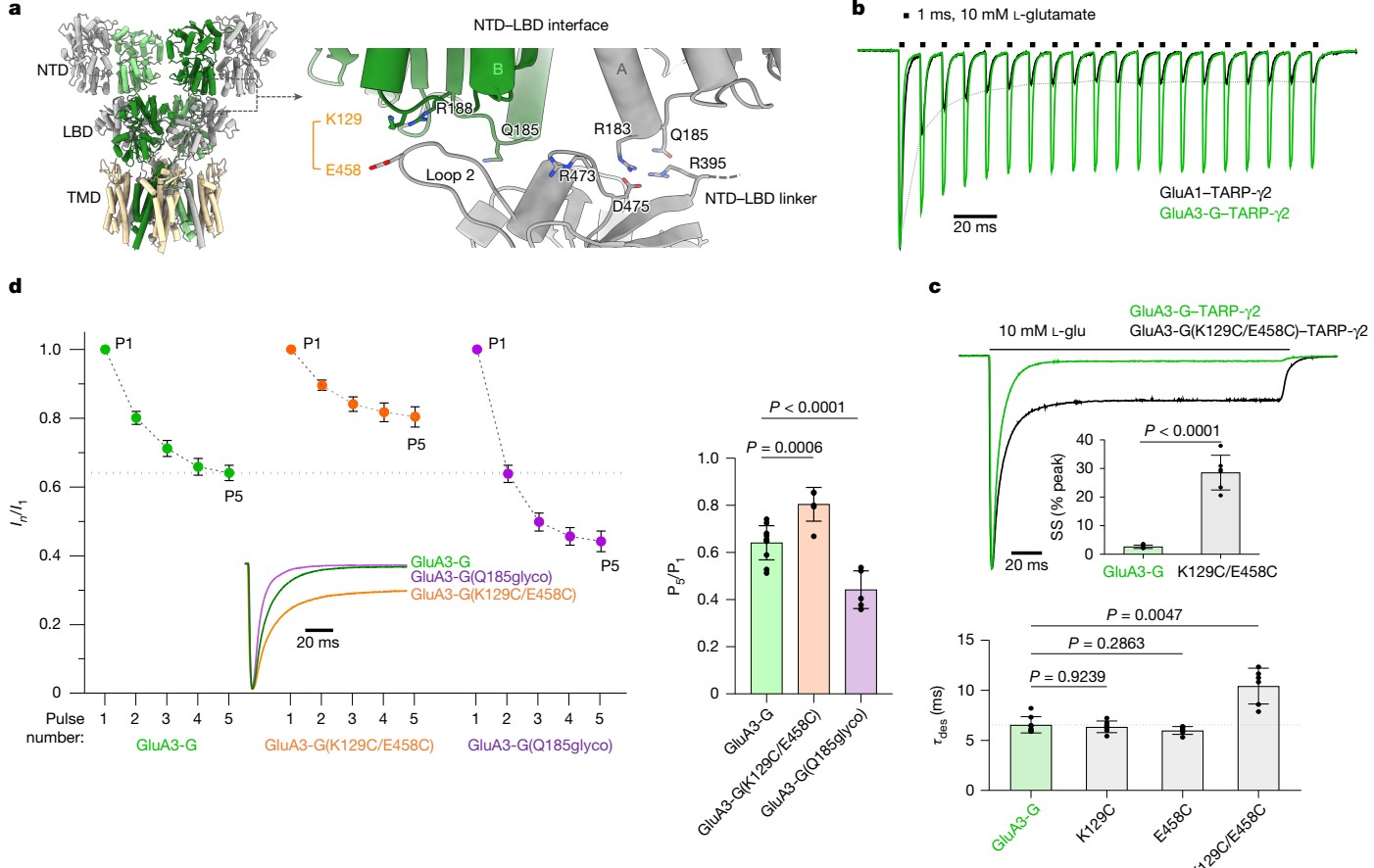

**Fig. 2 | Functional relevance of the GluA3 NTD–LBD interface. a**, Model of the apo-state interface, showing key interacting residues. LBD loop 2 and Arg395 in the NTD–LBD linker are also shown, both of which contribute to the interface. **b**, Representative overlaid current traces from outside-out patches of HEK293 cells expressing GluA3-R–TARP-γ2 (green) and GluA1–TARP-γ2 (black), elicited with 10 mM glutamate (1 ms pulses at 100 Hz). Currents are normalized to the first response (P1) in the application. **c**, Example peak-scaled whole-cell currents elicited by application of 10 mM L-glutamate for 200 ms at −60 mV from cells expressing GluA3-G–TARP-γ2 (green trace) and GluA3-G(K129C/E458C)–TARP-γ2 (black trace) (top). Inset: the mean ± s.d. steady-state (SS) current for GluA3-G–TARP-γ2 (n = 8 cells) and GluA3-R(K129C/E458C)–TARP-γ2 (n = 6 cells). Statistical analysis was performed using an unpaired two-tailed t-test; t = 12.16, d.f. = 12, P < 0.0001. Bottom, the mean ± s.d. desensitization time constants for GluA3-G–TARP-γ2 (n = 8 cells), GluA3-G(K129C)–TARP-γ2

(n = 7 cells), GluA3-G(E458C)–TARP-γ2 (n = 6 cells) and GluA3-G(K129C/E458C)–TARP-γ2 (n = 6 cells). Statistical analysis was performed using Welch's analysis of variance (ANOVA; $W_{3,8.572}$ = 11.01, P = 0.0010) followed by Dunnett's multiple-comparison test; P values are indicated. **d**, Current amplitudes of the first five responses evoked from the outside-out patches using the protocol as described in **b**, expressing GluA3-G–TARP-γ2 (n = 11 patches), GluA3-G(K129C/E458C)–TARP-γ2 (n = 6 patches) and GluA3-G(Q185glyco)–TARP-γ2 (n = 7 patches) (left). Currents are normalized to the first pulse. Data are mean ± s.d. A dashed line connecting the dots is included as a visual guide. Inset: aligned, peak-scaled whole-cell current responses to 10 mM, 200 ms glutamate application. Right, the mean ± s.d. current ratio between the fifth and first pulses (P5/P1) for the same data. Statistical analysis was performed using one-way ANOVA ($W_{(2,21)}$ = 38.70, P < 0.0001) followed by Dunnett's multiple-comparisons test; P values are indicated.

## Organization of desensitized GluA3

Desensitized GluA3-G also departs from known AMPAR structures. Lacking the stabilizing B/D NTD interface of GluA2 (Fig. 1a), the NTD dimers undergo a spectrum of conformations (Extended Data Fig. 9a,b). In the LBD tier, either one LBD dimer splits into monomers or both do, leading to a pseudo-four-fold symmetry as seen in GluA1 (ref. 14) (Fig. 3a and Extended Data Fig. 9a). In marked contrast to GluA1[14], desensitized GluA3 maintains its domain-swapped architecture between the NTD and LBD tiers[7], despite its structural flexibility. Domain unswapping in GluA1 is associated with slow desensitization recovery, consequently affecting synaptic transmission[8,14]. Thus, conservation of the domain-swapped state in GluA3 may contribute to rapid kinetics, facilitating high-frequency signal transmission (Fig. 2b).

We propose that maintenance of the domain-swap in desensitized GluA3 is linked to a previously unseen NTD–LBD arrangement, observed here in 30% of 3D classes (Fig. 3b–e and Extended Data Fig. 9a,b). This conformation could be resolved to 4 Å, and is established by an upward

rotation of the LBD of chain B towards its respective NTD, culminating in an atypical NTD–LBD apposition that involves LBD helix J (Fig. 3a,b and Supplementary Video 4). Helix J is a central regulatory element: it forms the core of the LBD dimer interface in resting-state and open-state AMPARs[36] (Fig. 3a (left)). It is the target of RNA editing at the R/G site and of alternative flip/flop splicing, determining both gating kinetics[41,42] and receptor biogenesis[43–45].

In desensitized GluA3-G, helix J projects the alternatively spliced Thr748 towards Gly235 in the NTD (Fig. 3d,e), an arrangement that is not possible in apo or open states. Our attempt to introduce a disulfide bridge between Gly235 and Thr748 led to barely resolvable currents (presumably due to the gating role of helix J within LBD dimers). Nevertheless, mutation of Gly235 (G235A and G235C) slowed desensitization entry and sped recovery (Fig. 3f, Extended Data Fig. 9d and Extended Data Table 1). This phenotype was specific to GluA3, and was not apparent in the equivalent GluA2 mutation (Extended Data Table 1). The GluA3-desensitized interface also harbours the Gly439/Arg439 trafficking checkpoint[27], as well as various disease-associated mutations, such

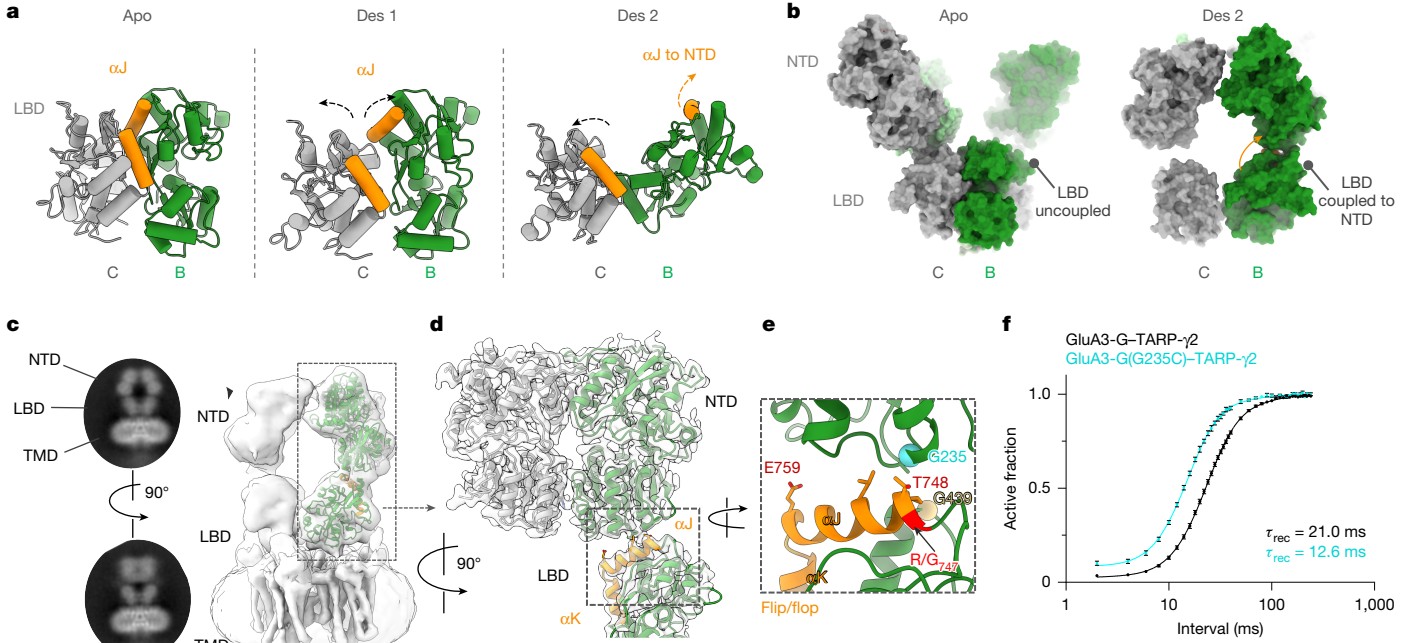

**Fig. 3 | Organization of desensitized GluA3. a**, Models of the GluA3 LBD dimer in apo (left) and desensitized (Des 1 and 2; middle and right) states. **b**, Atomic models of the full NTD and LBD layers for apo (left) and desensitized state 2 (right) are shown as a surface representation, highlighting the switch in NTD–LBD interactions that couples the B-chain NTD and LBD in the desensitized state. **c**, The dominant desensitized state features are shown with 2D class averages, a 3D map and a fitted atomic model, including the coupled chain-B NTD and LBD. **d**,**e**, Magnified views of the high-resolution NTD–LBD region, highlighting the key interacting regions including αJ and αK of the alternatively spliced flip/flop cassette in orange and the R/G RNA-editing site in red, and the Gly439/Arg439 mammalian switch site as a red or yellow sphere. The area indicated by a box in **d** is magnified in **e**. **f**, Recovery from desensitization was analysed from the pooled data (GluA3-G–TARP-γ2 (n = 9 cells) and GluA3-G(G235C)–TARP-γ2 (n = 7 cells)), fitted with a two-component Hodgkin–Huxley equation (black line, GluA3-G fit; cyan line, GluA3-G(G235C) fit). The slope for the fast and slow components was fixed at 4 and 1, respectively.

as A237T in the vicinity of Gly235, which is associated with epilepsy, as well as T748M and E759K, both on helix J[22] (Fig. 3e and Extended Data Fig. 5c). Together, these findings suggest functional relevance of this desensitized conformation.

The NTD dimers adopt multiple conformations in desensitized GluA3, ranging from dimer splaying to formation of a roof-shaped NTD tier, reminiscent of GluA2/3 heteromers[30] (Fig. 3c and Extended Data Fig. 9b). As the NTD anchors AMPARs at the synapse, these conformations could impact associations with synaptic-cleft components to shape synaptic transmission[8–10].

## Conformational landscape of the GluA3 NTD

As the flat NTD dimer conformation persists in all three gating states, we assessed the stability of this arrangement using all-atom molecular dynamics (MD) simulations. We measured dimer displacement using a (usually negative) torsion angle (Fig. 4a), capturing movement of the lower lobe centres from rotation of the subunits around an axis defined by the centres of the upper lobes. This torsion angle is near-zero for the flat conformation but increases to about 30° in displaced dimers[33], therefore closely matching GluA2 (Extended Data Fig. 4c). Arg163 stacking persisted in three independent 200 ns MD runs (Fig. 4b (left)), while mutation of Arg163 to isoleucine (R163I) led to dimer displacement (~25°) that closely matched other AMPAR NTDs (Fig 4b and Extended Data Fig. 4c). Dimer motions are reduced when simulating the NTD associated with the LBD (Extended Data Fig. 10a), highlighting the stabilizing influence of the LBD on NTD dimer conformation.

Isolated GluA3 NTD dimers can undergo a spectrum of conformations[33,46]. To further explore the energy landscape of the GluA3 NTD, we used metadynamics—an enhanced sampling MD approach that pushes a structure away from already visited conformations, and tracks

the underlying energies involved[47]. The conformational landscape of GluA3 NTD dimers is defined by two coordinates: dimer displacement torsion and interface opening angle (Fig. 4c and Methods). This landscape features a central, low-energy basin (purple) containing four distinct energy wells (1–4; black), surrounded by energetically unfavourable regions (red). The flat NTD dimer conformation, a key low-energy state, is located in well 3. The landscape also maps other known conformations. A displaced, GluA2-like dimer (PDB: 3P3W (chains B/D)) occupies well 2, and a splayed-open structure (PDB: 6FLR) is located in an intermediate region (Fig. 4c,d). The outer wells contain an 'extreme displaced' dimer (well 1), and another that displaces the opposite way with positive torsion (well 4). Less-populated regions outside this basin include more-displaced and open conformations reminiscent of NMDAR NTDs[48]. Notably, a high activation-energy barrier separates wells 2 and 3, yet this transition state region is occupied by intermediate crystal structures, including one containing phosphate ions[33,46] (Fig. 4d).

MD simulations of an intermediate crystal structure (PDB: 3O21, dimer CD) at physiological salt concentration revealed that a chloride ion binding between Arg163 and Arg164 creates a stable intermediate state (Extended Data Fig. 10b (right)). This observation supports the hypothesis that anions, such as chloride and phosphate, can stabilize intermediates and reduce the transition barrier, a finding that may aid the development of therapeutic ligands[46]. Together, these results reveal the stability of the flat GluA3 NTD, and emphasize the potential for the GluA3 NTD dimer to populate a wide range of states.

## Cryo-EM structure of GluA3(R163I)

As Arg163 is critical to GluA3 architecture and locates to a druggable site[46], we determined an apo-state cryo-EM structure of

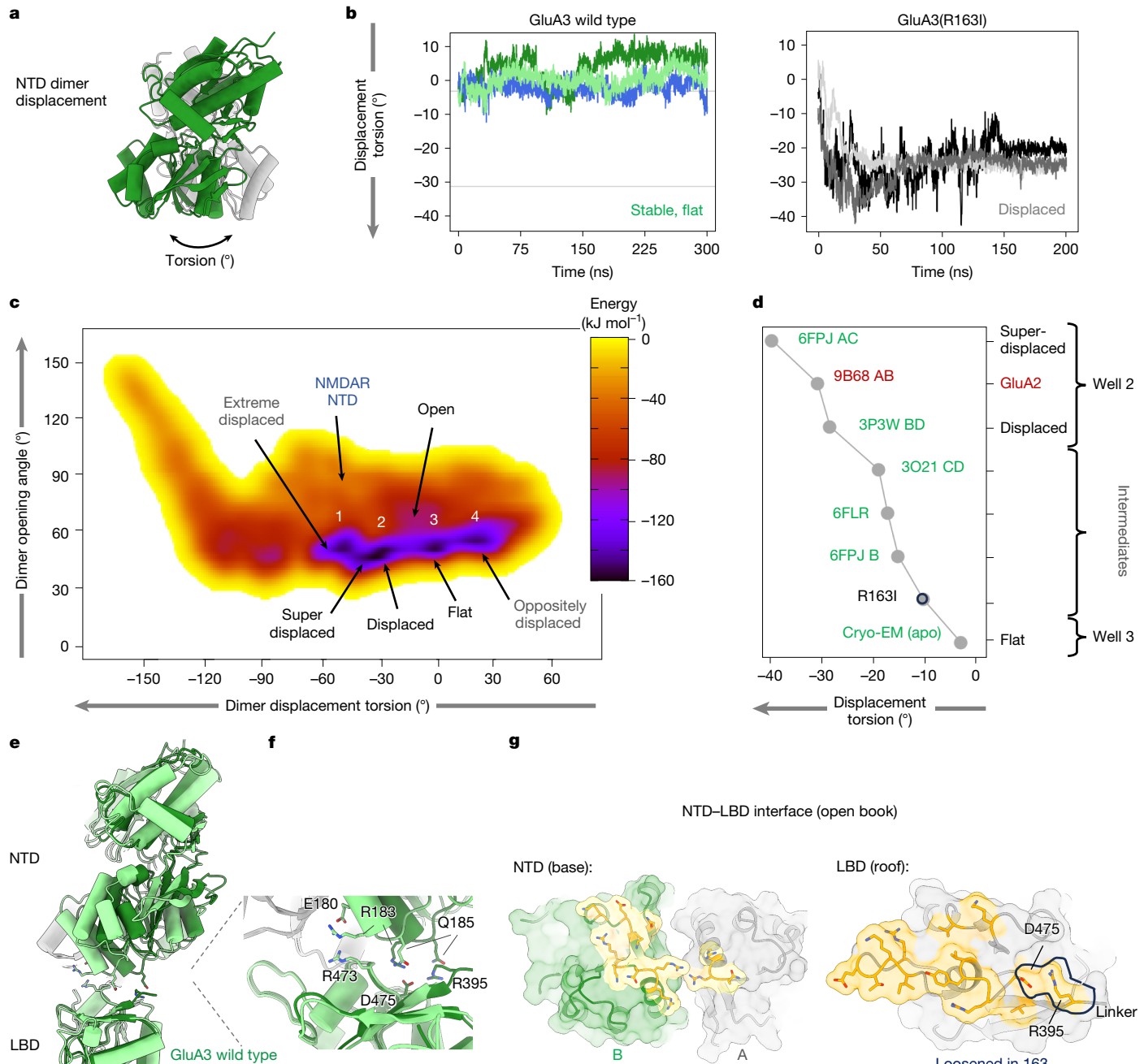

**Fig. 4 | Dynamics of the GluA3 NTD. a**, The displacement torsion angle is illustrated on a displaced GluA3 NTD dimer (PDB: 6FPJ dimer A/C). **b**, The displacement torsion is plotted for three NTD dimer simulations of GluA3 wild type (left) and R163I (right). Wild-type torsions remain around 0°, while the mutant shifts to more displaced with torsions around −25°. Plots for simulations including LBD are provided in Extended Data Fig. 10a, with NTD dimer r.m.s.d. values with and without the LBD. **c**, A free-energy landscape from metadynamics shows the most energetically favourable conformations and structural variability for wild-type NTD dimers. The x and y axes represent displacement (as in **a** and **b**; more displaced on the left) and opening. Colours show low energies in black and purple for more favourable conformations and higher energies in red and yellow. Four wells are labelled in white with known structures in black (wells 2 and 3) and proposed conformations in grey (wells 1 and 4). A rare conformation resembling NMDARs is labelled. **d**, Experimental GluA3 structures are plotted by displacement torsion, from superdisplaced at the top left to flat at the bottom right. Previously published structures are indicated by PDB codes. GluA2 is shown in red and the GluA3(R163I) cryo-EM structure, which is slightly more displaced than flat apo wild-type dimers, is highlighted by a black circle. Some structures are shown in Extended Data Figs. 4a and 8b with some heterodimers. **e**, Overlay of GluA3 wild type (dark green) and GluA3(R163I) (light green), showing displacement of mutant NTD relative to the wild type. **f**, Magnified NTD–LBD interface showing reorientation of residues, including Glu180, Arg183, Gln185 and Asp475. **g**, The wild-type NTD–LBD interface is shown in yellow open-book format, with the NTD on the left and LBD on the right. The black-contoured region shows a loss of contacts in GluA3(R163I) (Arg395 and Asp475).

the GluA3(R163I) receptor (Fig. 4e,f and Extended Data Fig. 3d). GluA3(R163I) exhibited semidisplaced NTDs, similar to a GluA3 NTD crystal structure in which the Arg163 side chains coordinate a phosphate ion (PDB: 6FPJ)[46], and to GluA3 NTD simulations that include the LBD (Fig. 4d and Extended Data Fig. 10a,c). This NTD displacement also caused reconfigurations in the NTD–LBD interface (Fig. 4e–g), including the NTD–LBD linker region, as documented in the interface footprint (Fig. 4g). The GluA3(R163I) structure highlights the contribution

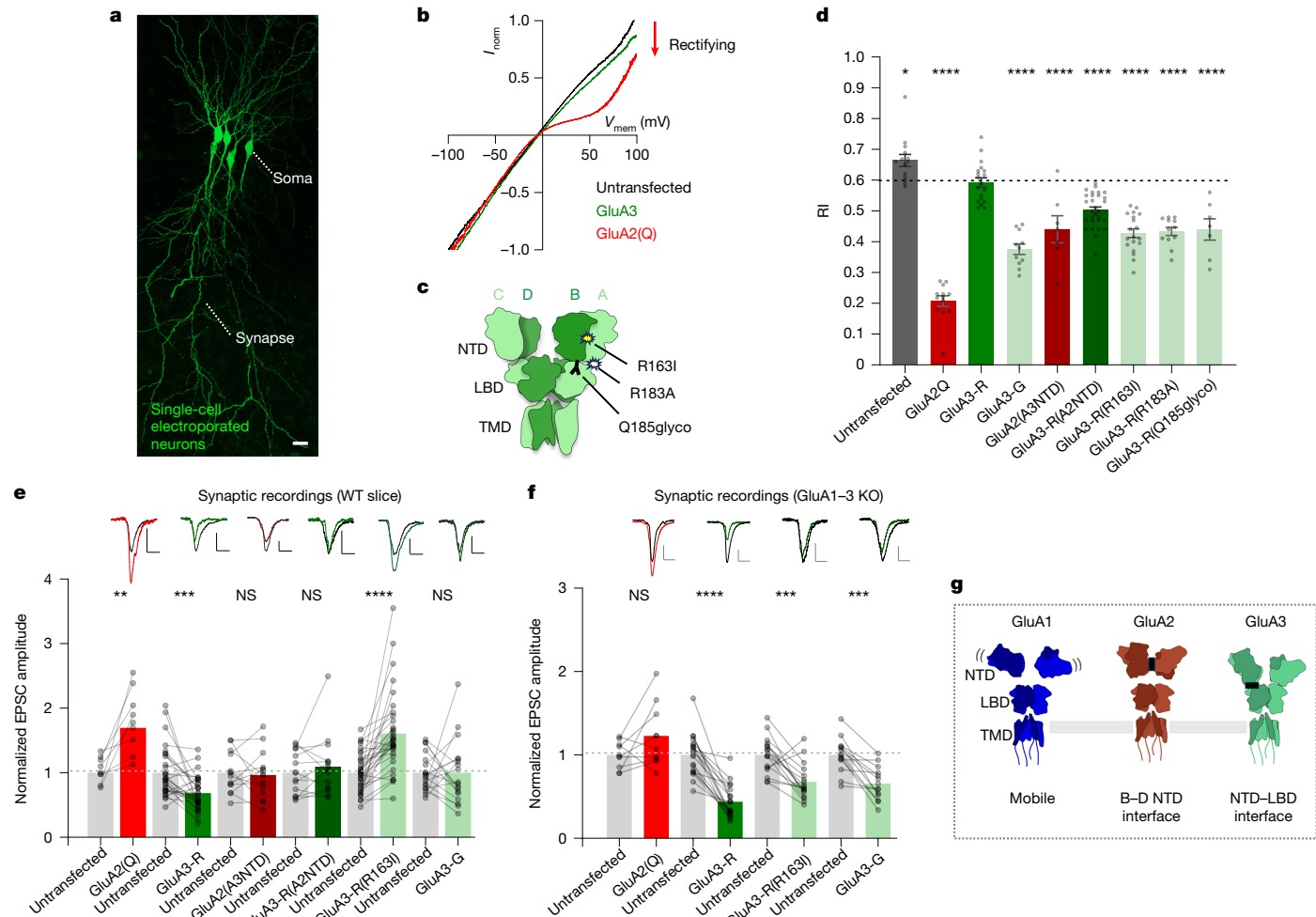

**Fig. 5 | Surface trafficking and synaptic transmission of GluA3 in CA1 neurons. a**, Single-cell electroporated CA1 pyramidal neurons in organotypic slice were used to measure AMPAR currents from somatic patches or synaptic recordings. Scale bar, 20 μm. **b**, $I/V$ curves of glutamate-evoked AMPAR currents from somatic outside-out patches. An inward rectification, as shown by GluA2(Q), indicates predominant surface expression of exogenous AMPAR. **c**, Schematic of a GluA3 homomer showing the positions of the R163I, R183A and Q185glyco mutations at the NTD and NTD–LBD interface of the A and B subunits. **d**, The RI of AMPAR currents from somatic outside-out patches of single-cell electroporated or untransfected cells. Statistical analysis was performed using one-way ANOVA ($F_{8,122} = 54.4$, $P < 0.0001$) with Dunnet's multiple-comparison test comparing with the mean of GluA3-R. Data are

mean ± s.e.m. **e**, Normalized EPSC amplitudes from dual synaptic recordings of a transfected and neighbouring untransfected neuron. Statistical analysis was performed using paired $t$-tests. Example traces from untransfected (grey) and transfected (coloured) cells are shown above the corresponding bar. WT, wild type. **f**, Normalized EPSC amplitudes from AMPAR knockout (KO) and rescue in *Gria1-3*$^{fl/fl}$ tissue. Dual recordings were performed from untransfected (Cre-negative) and transfected (Cre-positive + AMPAR single-cell electroporation) cells. Scale bars, 20 pA (vertical) and 20 ms (horizontal). All statistical details and prenormalized data are provided as source data. Data are mean ± s.e.m. **g**, Model depicting the different NTD organizations and interfaces present in GluA1–3 homomers. *$P < 0.05$, **$P < 0.01$, ***$P < 0.001$, ****$P < 0.0001$; NS, not significant.

of Arg163 to the GluA3 architecture. We next investigated the influence of Arg163 on synaptic GluA3.

## The NTD determines GluA3 traffic in neurons

Trafficking of AMPAR subtypes includes transport to the cell surface, delivery into synapses and synaptic anchoring[8,49,50]. At hippocampal synapses, GluA2/3 heteromers mediate baseline transmission, while GluA3 homomers appear largely excluded[5]. We assessed the surface expression of GluA3 NTD variants in organotypic hippocampal slices by measuring rectification indices (RIs) from somatic membrane patches (Fig. 5a,b,d). A reduced RI reports surface expression of exogenous receptors unedited at the Gln586/Arg586 (Q/R) site, relative to untransfected neurons dominated by non-rectifying Q/R-edited heteromers[5,51]. Whereas GluA2(Q) (Gln586) is readily expressed at the cell surface (GluA2: $0.21 \pm 0.02$, $n = 13$; untransfected: $0.66 \pm 0.02$, $n = 14$)[9,10], this was not the case for GluA3-R ($0.59 \pm 0.01$, $n = 19$), but was apparent

for GluA3-G ($0.38 \pm 0.02$, $n = 11$) (Fig. 5d), consistent with data from HEK293 cells[27] (Extended Data Fig. 6a).

In contrast to HEK293 cells[27], the GluA3 NTD contributes to trafficking in neurons. Swapping the GluA3 NTD onto GluA2 reduced the surface expression of the GluA2(GluA3-NTD) chimera, while the reverse swap facilitated expression of GluA3(GluA2-NTD) (GluA2(GluA3-NTD): $0.44 \pm 0.04$, $n = 7$; GluA3(GluA2-NTD): $0.50 \pm 0.01$, $n = 29$), uncovering a negative influence of the NTD of GluA3 (Fig. 5d). This is due to NTD conformation, as surface levels increased for the dimer interface mutant GluA3-R(R163I), and for the NTD–LBD interface mutants R183A and Q185glyco (GluA3-R(R163I): $0.43 \pm 0.01$, $n = 19$; GluA3-R(R183A): $0.43 \pm 0.01$, $n = 12$; GluA3-R(Q185glyco): $0.44 \pm 0.03$, $n = 7$) (Fig. 5c,d). This was specific to GluA3, as the equivalent NTD–LBD interface mutations were of no consequence in GluA2. The NTD dimer mutation (I157R) did not decrease the somatic RI and, in contrast to GluA3, did not impact the synaptic transmission of GluA2 (see below) (Extended Data Fig. 11c,d).

## GluA3(R163I) boosts synaptic transmission

Transfected GluA2(Q) robustly increases Schaffer-collateral-evoked excitatory postsynaptic currents (EPSCs) relative to a nearby untransfected neuron[9,10], but this is not seen with GluA3-R (Fig. 5e). Swapping the NTDs partially reverses this behaviour, emphasizing the synaptic anchoring ability of the GluA2 NTD[9,10,52] and demonstrating an apparent inability of the GluA3 NTD to anchor at the CA3–CA1 synapse. In contrast to GluA3-R, GluA3-G did not decrease EPSCs relative to untransfected cells, reflecting the increased surface expression of GluA3-G as a homomeric receptor (Fig. 5e). Notably, GluA3-R(R163I) significantly increased EPSCs, nearly reaching the relative current amplitudes of GluA2(Q) (Fig. 5e). Thus, this point mutation in the NTD dimer interface outweighs the impact of transplanting the entire GluA2 NTD (GluA3-R(GluA2-NTD)).

To resolve this unexpected phenotype, we next tested the impact of R163I on ion-channel conductance using non-stationary fluctuation analysis. We found that both GluA3-R and GluA3-R(R163I) (expressed in HEK293 cells and associated with TARP-$\gamma$2) exhibited comparable measures of conductance and open probability ($P_{open}$: 0.66 and 0.74; conductance: 30 and 31 pS for GluA3-R–TARP-$\gamma$2 and GluA3-R(R163I)–TARP-$\gamma$2; $n$ = 4 patches) (Extended Data Fig. 11a,b). Thus, ion-channel function is not substantially altered by the R163I mutation.

As the NTD dimer interface orchestrates AMPAR biogenesis, a mutation here may alter assembly[31], explaining the synaptic phenotype of GluA3-R(R163I). Indeed, GluA3-G(R163I) did not increase EPSCs (Extended Data Fig. 11e,f), probably due to its preferential assembly into GluA3 homomers (Extended Data Fig. 6h,i). The recordings above were performed in wild-type slices in the presence of endogenous AMPAR subunits. To assay the role of R163I in receptor biogenesis, we generated GluA1-3-knockout neurons using *Gria1-3*$^{fl/fl}$ mice[53]. In this AMPAR-null background, GluA3-R(R163I) homomers rescued EPSCs to a greater extent than GluA3-R (GluA3-R(R163I): 0.68 ± 0.05, $n$ = 16; GluA3-R: 0.44 ± 0.05, $n$ = 19), but not to the extent of GluA2 (1.2 ± 0.1, $n$ = 9) (Fig. 5f). EPSC amplitudes were now similar to GluA3-G (0.66 ± 0.06, $n$ = 13), matching their similar surface trafficking behaviour (Fig. 5d). This difference in GluA3-R(R163I) phenotype depending on the presence of endogenous AMPARs suggests that modifying the NTD dimer interface at Arg163 reroutes biogenesis in favour of heteromeric assembly, enabling GluA3-R accumulation at synapses to increase transmission.

## Conclusion

The GluA3 homomer adopts an architecture that is not observed in other AMPARs. Its unique NTD/LBD-coupled organization substantially differs from the detached NTD tier of GluA2 receptors, which acts as an efficient synaptic anchor[8–10], and from the highly mobile GluA1 NTDs[14] (Fig. 5g). Domain coupling is expected to enrich the allosteric landscape of GluA3 and its localization at synapses enriched with GluA3 (such as sensory synapses and interneurons). Our mutational analysis further supports the functional relevance of this interface (Fig. 2), while our structures offer a framework for the development of GluA3-selective modulators. These could target (1) the Arg163 site in the NTD dimer interface, a hotspot for small ligands (like $PO_4^{3-}$)[46] (Fig. 1f) or (2) the NTD–LBD interface, which shapes gating kinetics (Fig. 2). Moreover, our data further highlight the critical role of the NTD in AMPAR subunit assembly[31,32,54]. How the R163I mutation alters the NTD dimer interface to aid GluA3 heteromerization is an open question that will shed light on the first step of AMPAR biogenesis.

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

## Methods

### Expression and purification of GluA3–TARP-γ2 and associated mutants

All cDNA constructs were generated using in vivo assembly (IVA) cloning[55]. The GluA3-TARP-γ2 tandem construct was engineered by fusing TARP-γ2 (*Rattus norvegicus* cDNA sequence) to the C terminus of GluA3 (rat cDNA sequence, R/G-edited flip isoform) within the pRK5 vector, connected by a Gly-Ser-Gly-Ser-Gly linker sequence. The GluA3 construct was tagged with a Flag epitope at the N terminus, immediately following the signal peptide. Moreover, an eGFP tag and a human rhinovirus 3C (HRV 3C) protease cleavage site was appended to the C terminus of TARP-γ2 for visualization of expression and purification. Site-directed mutagenesis was performed to generate the R439G and R163I mutants. These mutants were cloned using IVA cloning based on the wild-type pRK5 constructs (R/G-edited, flip isoform). Expi293 cells were transfected with the aforementioned plasmids for all receptors studied. These cells were regularly tested for mycoplasma contamination. To prevent AMPA receptor-mediated excitotoxicity, AMPAR antagonists ZK200775 (5 nM, Tocris, 2345) and kynurenic acid (0.2 mM, Sigma-Aldrich, K335-5G) were added to the culture medium. Then, 40–48 h after transfection, cells were collected and lysed for 3 h in lysis buffer containing 50 mM Tris, 150 mM NaCl, 1.2% digitonin (w/v; Sigma-Aldrich, 300410-5 G) at pH 8, 2 mM PMSF and 4× protease inhibitor (Roche, 05056489001). Insoluble material was removed by ultracentrifugation (Type 45 Ti Fixed-Angle Titanium Rotor) at 200,000*g* for 1 h. The clarified lysates were incubated with anti-GFP beads for 3 h.

After washing with glyco-diosgenin (GDN) (Anatrace, GDN101) buffers (50 mM Tris pH 8, 150 mM NaCl, 0.1% GDN, 1 mM ATP and 1 mM MgCl$_2$; and 50 mM Tris pH 8, 150 mM NaCl and 0.1% GDN) in quick succession, the protein was eluted from the beads by digestion with 0.03 mg ml$^{-1}$ 3C protease at 4 °C overnight. The eluate was concentrated to 0.5 ml and loaded onto the Superose-6 10/300 size-exclusion chromatography column. The eluted fractions were pooled and concentrated to 2–3 mg ml$^{-1}$ and vitrified on the cryo-EM grids.

### Cryo-EM grid preparation and data collection

A volume of 2.5–4.5 μl of protein solution, depending on the sample, was applied to QuantiFoil R0.6/1 (300 mesh) or UltraFoil R1.2/1.3 (300 mesh) grids and plunged into liquid ethane using a Vitrobot Mark 4 (Thermo Fisher Scientific). The standard parameters were as follows: blot force = 7, blot time = 4 s, temperature = 4 °C, humidity = 100%, wait time = 20 s and no drain time. The QuantiFoil and UltraFoil grids were glow discharged at 25 mA for 25 s and 90 s, respectively, using the PELCO easiGlow system to render them hydrophilic before vitrification.

For the open state, the protein was initially incubated with 300 μM cyclothiazide (Tocris, 0713), a positive allosteric modulator preventing desensitization, for at least 30 min on ice, followed by rapid mixing with a 1 M L-glutamate stock solution (pH 7.4) to a final concentration of 100 mM before grid application.

To induce the desensitized state for both GluA3-R and GluA3-G, 10 mM quisqualate (Tocris, 0188) was swiftly added to the protein to a final concentration of 1 mM before grid application.

### Data acquisition for GluA3-G–TARP-γ2 apo state

In total, 63,921 multiframe videos were acquired using the 300 kV Thermo Fisher Scientific Titan Krios microscope equipped with a Gatan K3 direct electron detector. The detector operated in electron-counting mode with a 20 eV slit width for the BioQuantum energy filter. A 100 μm objective aperture was also inserted. Data were collected over three sessions.

Videos were dose-fractionated into 40 frames over an exposure time of 1–1.2 s, resulting in a total electron dose ranging from 38.4 to 48.2 e$^-$ Å$^{-2}$). Data collection was performed using EPU (Thermo Fisher Scientific) software in the faster acquisition aberration-free image shift (AFIS) mode with 5 s delay after stage shift and 1 s delay after image shift. Images were recorded at a magnification of ×105,000 in super-resolution mode with binning 2, yielding an effective pixel size of 0.826 Å per pixel. The defocus range was set between −1.2 and −2.5 μm.

### Data acquisition for GluA3-G–TARP-γ2 open state

In total, 27,709 multiframe videos were acquired using the 300 kV Thermo Fisher Scientific Titan Krios microscope equipped with a Gatan K3 direct electron detector configured as above.

Videos were dose-fractionated into 40 frames over an exposure time of 1 s, resulting in a total electron dose of 37.6 e$^-$ Å$^{-2}$. Images were recorded at a magnification of ×105,000 in super-resolution mode with binning 2, yielding an effective pixel size of 0.826 Å per pixel. The defocus range was set between −1.0 and −2.4 μm.

### Data acquisition for GluA3-G–TARP-γ2 desensitized state

In total, 31,470 multiframe videos were acquired using the 300 kV Thermo Fisher Scientific Titan Krios microscope equipped with a Gatan K3 direct electron detector configured as above. Data were collected over two sessions.

Videos were dose-fractionated into 40 frames over an exposure time of 1 s, resulting in a total electron dose of 36.4 e$^-$ Å$^{-2}$. Images were recorded at a magnification of ×105,000 in super-resolution mode with binning 2, yielding an effective pixel size of 0.826 Å per pixel. The defocus range was set between −1.2 and −2.5 μm.

### Data acquisition for GluA3-G(R163I)–TARP-γ2 apo state

In total, 34,581 multiframe videos were acquired using the 300 kV Thermo Fisher Scientific Titan Krios microscope equipped with a Falcon 4i direct electron detector. The detector was operated in electron-counting mode with a 10 eV slit width for the SelectrisX energy filter.

Data were collected in the EER format and compressed into 40 frames over an exposure time of 3.36 s, resulting in a total electron dose of 40 e$^-$ Å$^{-2}$. Images were recorded at a magnification of ×130,000 in super-resolution mode with binning 2, yielding an effective pixel size of 0.955 Å per pixel. The defocus range was set between −1.2 and −2.5 μm.

### Data acquisition for GluA3-R–TARP-γ2 apo state

In total, 34,514 multiframe videos were acquired using the 300 kV Thermo Fisher Scientific Titan Krios microscope equipped with a Gatan K3 direct electron detector configured as above.

Videos were dose-fractionated into 40 frames over an exposure time of 1.1 s, resulting in a total electron dose of 41.27 e$^-$ Å$^{-2}$. Images were recorded at a magnification of ×105,000 in super-resolution mode with binning 2, yielding an effective pixel size of 0.826 Å per pixel. The defocus range was set between −1.2 and −2.5 μm.

### Cryo-EM image processing

The general schematic of the image processing workflow for the different datasets is shown in Extended Data Figs. 1 and 7a and is nearly identical for each dataset. All data were processed using cryoSPARC (v.4.41)[56] and RELION (v.5.0)[57].

Dose-fractionated videos were initially exported to cryoSPARC live for preprocessing. After patch motion correction and contrast transfer function (CTF) estimation, micrographs with a total motion above 100 px and CTF fits below 10 Å were discarded, along with ice-contaminated micrographs. Particles were picked from the remaining micrographs using the blob picker tool. These particles were extracted at a box size of 512 px (downsampled by a factor of 4 to 128 px) and subjected to 2D classification.

The best-resolved 2D classes underwent multiple rounds of ab initio refinement to obtain a good reference volume with the same overall shape as an AMPAR. This volume, along with generated bait classes (noise with very few particles), were used as references for iterative

heterogeneous refinements. Particles that did not align with the good volume were filtered out, and the remaining particles were used to generate a consensus map.

This consensus map was used to create 2D templates for improved template-based picking, and the above steps were repeated to obtain a new consensus map and corresponding particles. These particles were rescaled to the original box size of 512 px and used for homogeneous and non-uniform refinement with dynamic masks.

These aligned particle stacks were exported to RELION using the Python script csparc2star.py[58]. Particles were re-extracted (with a binning factor of 4) for focused 3D refinement and classification downstream. $C_1$ symmetry was applied throughout processing up to this point.

The following masks were generated from the consensus maps imported from cryoSPARC: TMD tetramer with 4 TARP-γ2, TMD-LBD tetramer with 4 TARP-γ2 and LBD-NTD tetramer. Additional 3D classifications without image realignment were performed using these masks to separate different conformations or further clean-up the datasets. Several rounds of 3D classification were conducted in RELION, using different class numbers (5–24), symmetries and regularization parameters (T values ranging from 2 to 80) to achieve the widest spread of various conformations. Particles selected from the best-resolved or unique classes were rescaled to the original pixel size of 0.826 (or 0.955 for dataset 4). These particles underwent focused 3D refinement and global/local CTF refinement in cryoSPARC to generate high-resolution maps for each domain of the receptor (Extended Data Figs. 1 and 7a). The half maps generated from these refinements were then used to estimate the local resolution of these maps in cryoSPARC (Extended Data Fig. 3).

For datasets 1, 2, and 4, due to the symmetric nature of the AMPAR tetramer, all particles corresponding to a docked NTD dimer on the LBD aligned together on one side of the receptor, resulting in an asymmetrical appearance with only one well-resolved NTD dimer. To better resolve the NTD–LBD interface for the docked dimer, masks of the NTD dimer and/or NTD dimer with the upper lobe of the LBD were used for 3D classification. As described earlier, particles from the best-resolved classes were selected for final high-resolution refinements.

For dataset 1, to obtain a complete map of the GluA3-G–TARP-γ2 receptor in the apo state, $C_2$ symmetry with symmetry relaxation was applied during 3D refinement to the particles in the consensus map. The resulting particles from this refinement were further classified without image alignment using an LBD–NTD tetrameric mask. One of the classes displayed both NTD dimers docked on the LBD and well-resolved. This conformation was refined and served as a template to create a composite map of the receptor, as depicted in Fig. 1.

In dataset 3 (the desensitized state), the LBD layer showed significant heterogeneity, in contrast to the other states. Various conformations with different degrees of LBD dimer rupture were identified through 3D classification. Two extreme classes from this dataset (Extended Data Fig. 7a (class 1 and 2)) were extracted and refined according to the same workflow.

### Heterogeneity analysis of the NTD using DynaMight

To estimate the dynamics of the NTD in datasets 1 and 2 (apo and open states of GluA3-G–TARP-γ2, respectively), we used a slightly modified version of DynaMight[28] integrated into RELION. In the version used, the decoder does not predict the displacement of each individual Gaussian, but instead outputs Euler angles and a shift vector for groups of Gaussians. For grouping the Gaussians, users can provide masks of specific domains or regions, for which the rigid transformations are estimated by the VAE framework. Compared to the standard DynaMight version, this enforces a stronger prior on the deformations that is more suitable for disordered membrane regions and also turned out to be beneficial for smaller particles.

In our study, we created five masks from the consensus reconstruction (TMD, LBD dimer 1, LBD dimer 2, NTD dimer 1, NTD dimer 2) for

both the open and apo states and used 10,000 Gaussians for both datasets. In our modified DynaMight, the consensus structure is progressively updated during deformation estimation. This is achieved by generating new star files for each region based on the current deformation estimate, followed by Fourier reconstruction and combination of the five maps.

The source code for the modified DynaMight is publicly available on GitHub[59] and further implementation details will be published separately.

### Model building and refinement

UCSF ChimeraX[60], PHENIX (v.1.20)[61], COOT (v.0.9.8.95)[62], Refmac-Servalcat[63] and PyMOL 2.5 (Schrödinger) were used for all molecular modelling and refinement.

The following PDB models were used as the starting models to construct the models presented in this work: 3O21 (GluA3 NTD dimer), 5IDE (GluA3 LBD-TMD), 3DLN (GluA3 LBD bound to glutamate), 4F29 (GluA3 LBD bound to quisqualate) and 8CIS (TARP-γ2 chains only). Each of these models was separated into its corresponding monomers and rigid-body fitted into the associated maps, and the monomers were then combined into a single model, all within UCSF ChimeraX.

The maps used in each case had $C_1$ symmetry and were autosharpened in PHENIX with a conservative resolution filter, that is, the lowest resolution in the local-resolution estimate of that map (Extended Data Fig. 3).

The preliminary models then underwent real-space refinement in PHENIX, followed by an all-atom refinement in COOT with the Geman–McLure α set to 0.1 (ref. 62). Outliers and poorly-fit areas were manually inspected and corrected in COOT. Most side chains were removed from the regions below 4 Å, and entire residues were removed from poorly resolved disordered regions. Clashes, nonrotameric side chains and geometry outliers were corrected. This process was performed iteratively alongside PHENIX real-space refinement until the Clashscores, CaBLAM, and Ramachandran statistics ceased to improve. The TARP-γ2 NTD–LBD models were also refined against unsharpened and unweighted half maps using the Refmac-Servalcat pipeline. COOT was also used to add glycans to the apo state NTD–LBD model and to insert the R163I and R439G mutations. Model validation was carried out using MolProbity[64]. All figures and videos in the paper were created with UCSF ChimeraX, cryoSPARC v.4.4 and COOT. PyMOL v.2.5 (Schrödinger) was used to fix some problems with chain and residue IDs. The pore profile of the apo and open state was generated using HOLE[65] integrated in COOT.

### MD simulations

An initial refined Cryo-EM structure of the apo NTD dimer with an associated LBD monomer was prepared for simulation by modelling the missing atoms and residues, including the NTD–LBD linker, using MODELLER (v.10.4)[66] using the Scipion-Chem framework[67] in Scipion (v.3.0)[68], using UniProt[69] sequence P19492 (default flop variant) with the signal peptide removed and a GT linker replacing the TMD between residues Lys508 and Pro632 as in LBD crystallization constructs. In silico mutations were made in PyMOL v.2.5 (Schrödinger) using the mutagenesis and sculpting wizards. System preparation and MD simulations for both the full NTD–LBD tri-domain system and an extracted NTD dimer were performed using GROMACS 2023[70,71] with the July 2021 release of the CHARMM36m all-atom force field[72] and the standard CHARMM-modified TIP3P water model[73,74] on which the CHARMM36m protein force field is based. The two systems were placed in rhombic dodecahedron boxes with 1.0 nm and 1.4 nm padding, respectively, to account for interactions of the protein molecules with copies across the periodic boundary given their dynamics. They were solvated with water and 0.15 M $Na^+$ and $Cl^-$ ions with additional ions to neutralize the system, leading to total sizes of about 100,000 and 280,000 atoms for the NTD dimer and NTD–LBD tri-domain, respectively. Electrostatic interactions were treated with the particle mesh Ewald formalism[75,76]

using a short-range cut-off of 1.2 nm and van der Waals interactions were treated with potential switching between 1.0 and 1.2 nm with a long-range neighbour list cut-off of 1.4 nm and dispersion correction applied to the energy and pressure, as recommended for CHARMM force fields[77]. A similar set-up approach was used for the parallel dimer (PDB: 3O21 dimer CD), but using an earlier version of MODELLER outside Scipion and GROMACS v.5.0.4 with the CHARMM22 force field for the protein with the energy correction map for backbone dihedral angles[78], sometimes referred to as the CHARMM27 force field, as implemented in GROMACS[77].

Energy minimization was run for 5,000 steepest descent steps with position restraints on the protein heavy atoms of 1,000 kJ mol nm$^{-2}$ to relax the solvent around the protein. This was followed by 1 ns of restrained NVT (constant number of atoms, volume and temperature) equilibration, keeping the same restraints, using the Bussi stochastic velocity rescaling thermostat[79,80] with two coupling groups corresponding to protein and non-protein atoms and a time constant of 0.1 ps, equilibrating to a temperature of 300 K while maintaining constant volume. Next, the system was subjected to 1 ns of restrained NPT (constant number of atoms, pressure and temperature) equilibration with the same restraints, using the stochastic cell rescaling barostat[81] with a time constant of 0.5 ps and a compressibility of $4.5 \times 10^{-5}$ bar$^{-1}$, equilibrating the system to a pressure of 1.0 bar. Finally, the restraints were removed and the system was equilibrated for a further 1 ns, before 100–300 ns production MD runs in the NPT ensemble (constant number of atoms, pressure and temperature). All of the steps after NVT equilibration used the same thermostat and barostat with the same parameters, including a 2 fs time step with the leap-frog integrator and bonds containing hydrogen atoms constrained with LINCS[82].

## NTD dimer free-energy metadynamics simulations

Metadynamics is an enhanced sampling simulation technique that uses a history-dependent bias potential to fill free-energy landscapes of complex systems such as proteins[83]. This causes the protein to disfavour conformations that it has already visited and allows one to reconstruct the visited region of the energy landscape as a function of a set of (usually two) collective variables. This bias potential takes the form of Gaussian hills that fill an energy landscape at regions that the structure visits. Whereas the initial landscape is characterized by energy wells where the structure likes to sit, deposition of hills enables escape from these wells and faster exploration of conformational space. Tracking the bias deposition allows one to reconstruct the energy landscape visited.

We used this approach to study the region of the free-energy landscape of the isolated GluA3 NTD dimer related to displacement and opening by designing two appropriate collective variables (CVs): a torsion angle for rotation of the subunits relative to each other and an angle for the opening. These were both based on centres of mass (COMs) of the upper lobe (residues 117 to 243 and 354 to 380) and lower lobe (residues 1 to 116 and 244 to 353) using their Cα atoms. The displacement torsion angle starts from the COM of the lower lobe of the first subunit (com2) then passes through the COMs of the two upper lobes (com1 and com3 for the same and other subunit, respectively), which define the rotation axis, and ends with the COM of the lower lobe of the second subunit (com3), capturing the rotation of the two lower lobes relative to each other. These COMs are defined based on groups of atoms taken from a GROMACS index file index.ndx, as are two more COMs for the combined COMs of both upper lobes together (com13) and both lower lobes together (com24).

The opening angle was defined using the two lobe COMs (com2 and com4) together with a joint COM for the two upper lobes combined (com13), creating two vectors (vector1 from com13 to com2 and vector2 from com13 to com4) between which the angle was calculated. The TORSION framework within Plumed was used for this opening angle, allowing it to be projected back onto a plane containing the COMs of each upper lobe (com1 and com 3) and of both lower lobes together

(com24) to remove any angle change arising from displacement. This plane was defined using a vector perpendicular to the plane with the axis keyword, which was calculated from the positions of a ghost atom (a point without mass that Plumed uses for measurements as defined with the GHOST command). The ghost atom was positioned in a reference coordinate frame such that it was at a distance of 0.5 Å from the COM of both lower lobes together (com24) perpendicular to the plane formed together with the COMs of each upper lobe (com1 and com3) by setting ATOMS=com24,com1,com3 and COORDINATES=0,0.5,0. Thus, the plane was initially defined implicitly within the GHOST command using the positions of these three COMs to define the placement of a ghost atom (g2), which was then used within to define the plane within the TORSION command using the perpendicular axis vector between the combined lower lobe COM (com24) and the ghost atom (g2). This then allowed the angle between vector1 and vector2 to be projected back onto the plane perpendicular to the axis vector to remove contributions from displacement.

Initial metadynamics runs were initiated from the model of the parallel GluA3 NTD dimer (PDB: 3O21 dimer CD). First, 1 ns runs with a large energy deposition rate (1.2 kJ mol$^{-1}$ high Gaussian hills of width 0.35 radians (20°) every ps) over just one collective variable confirmed that they were behaving as expected. Additional test runs were also used to establish the final less substantial bias deposition protocol used where narrower hills were deposited every 10 ps using diffusion-based adaptive Gaussians where the shape and width is dependent on the mean square displacement of the CVs over a defined time interval[84], also set to 10 ps. Initial structures for the final run were selected from these test runs.

We used the well-tempered metadynamics scheme[85,86] to help to ensure convergence rather than overfilling of the energy landscape. Well-tempering makes the hill height decrease over time while depositing in the same place; after escaping from a well and sampling a new place in the landscape the hills return to the starting height and begin decreasing again. This results in the CVs sampling a distribution with a temperature $T + \Delta T$. A bias factor $\gamma = (T + \Delta T)/T$ that determines the extent of sampling was set to 10 with $T = 300$ K and an initial height of 1.2 kJ mol$^{-1}$.

Metadynamics simulations were run in the canonical (NVT) ensemble using the Bussi thermostat[79,80] and no pressure coupling using Plumed (v.2.1.3)[87] patched onto GROMACS (v.5.0.4)[71], using the multiple walkers framework[88], allowing faster sampling of the energy landscape through the use of a shared bias potential.

Interactions were calculated using the CHARMM27 force field[77,78] with the TIP3P water model[74], modified for CHARMM force fields[73]. The water box extended at least 10 Å away from the protein in any direction. Sodium and chloride ions were added up to a concentration of 10 mM (plus additional ions to neutralize the system) by random replacement of water molecules. Electrostatics was treated with the particle-mesh Ewald algorithm[75,76] using a short-range cut-off of 12 Å and van der Waals interactions were switched off between 10 and 12 Å. Bonds containing hydrogen atoms were constrained with LINCS[82]. Virtual sites for hydrogens[89] and the Verlet interaction cut-off scheme[90] were used for efficient simulations.

Each metadynamics run was preceded by a single run of 5,000 steps of steepest descent minimization and then four runs were initialized with 1 ns of equilibration with restraints on the protein heavy atoms, and 1 ns of unrestrained equilibration (conventional MD without metadynamics biasing). The final energy landscapes were reconstructed using the histogram method for well-tempered metadynamics (as implemented in Plumed sum_hills), which has been shown to work well for multiple walker metadynamics[88].

## Analysis of simulations and existing structures

MD simulations were corrected for atoms moving over the periodic boundaries using Gromacs tools and imported into ProDy[91,92] (v.2.5.0)[93].

The structures were aligned over the NTD Cα atoms, which were used for calculating the r.m.s.d. from the starting structure. Displacement torsion angles were calculated using COMs of the upper lobe (residues 117 to 243 and 354 to 380) and lower lobe (residues 1 to 116 and 244 to 353) using Cα atoms as in the metadynamics. The experimental structures including a GluA2 dimer were aligned into an ensemble, allowing the use of the same code with the same residue selections.

## HEK293 cell electrophysiology

DNA constructs: sequences for rat GluA3 and rat GluA2 were flip variants. All cDNA constructs used for transfection were generated using IVA cloning as previously described[1]. Constructs were cloned in pRK5 vectors. The *N*-glycan in GluA3s (GluA3-G(Q185glyco) and GluA3-R(Q185glyco)) was introduced by mutating NTD residues Asn185 and Lys187 (mature peptide without signal sequence) to Asp and Thr, respectively.

HEK293T cells (ATCC, CRL-11268, 58483269: identity authenticated by short-tandem-repeat analysis; mycoplasma negative), were cultured at 37 °C under 5% $CO_2$ in DMEM (Gibco; high glucose, Gluta-MAX, pyruvate, 10569010) supplemented with 10% FBS (Gibco) and penicillin–streptomycin. Cells were transfected using Effectene (Qiagen) according to the manufacturer's protocol; the total plasmid DNA was 1 μg. The transfection ratio of AMPAR to TARP was 1:2. To avoid AMPAR-mediated toxicity, 30 μM 2,3-dioxo-6-nitro-1,2,3,4-tetrahydrobenzo[f]quinoxaline-7-sulfonamide (NBQX; Tocris, 1044; or HelloBio, HB0443) was added to the cell medium during the transfection procedure. Currents from the cells were recorded 48 h after transfection for the GluA3-R construct and 16–24 h after transfection for GluA3-G.

Recording pipettes were pulled with a P-1000 horizontal puller (Sutter Instruments) using borosilicate glass electrodes (1.5 mm outer diameter, 0.86 mm inner diameter, Science Products). Glass electrodes were heat-polished with an MF-830 microforge (Narishige) to final resistances of 2–4 MΩ (whole-cell recordings) and 6–12 MΩ (outside-out patches). Electrodes were filled with an internal solution containing CsF (120 mM), CsCl (10 mM), EGTA (10 mM), HEPES (10 mM), $Na_2$-ATP (2 mM) and spermine (0.1 mM), adjusted to pH 7.3 with CsOH. The extracellular solution contained NaCl (145 mM), KCl (3 mM), $CaCl_2$ (2 mM), $MgCl_2$ (1 mM), glucose (10 mM) and HEPES (10 mM), adjusted to pH 7.4 using NaOH.

Currents were recorded with the Axopatch 700B amplifier (Molecular Devices), prefiltered at 10 kHz with a 4-pole Bessel filter (amplifier built-in), sampled at 100 kHz with the Digidata 1550B (Molecular Devices), stored on a computer hard drive and analysed using the pClamp 11.2 software pack (Molecular Devices).

On the day of recording, cells were plated onto poly-L-lysine-treated glass coverslips. Fast perfusion experiments were performed with a double barrel application tool made from theta-tube borosilicate glass (Science Products) cut to a diameter of approximately 300 μm. The theta tube was mounted on a piezoelectric translator (Physik Instrumente) and command voltage (9 V) was filtered with a 250 Hz Bessel filter to reduce mechanical oscillations. The theta tube was filled with pressure-driven solutions (ALA Scientific Instruments). The speed of solution exchange at the theta tube interface was measured as 20–80% rise time of the current generated with 50% diluted extracellular solution. It was on average about 300 μs for whole-cell recordings and 120 μs for outside-out patches. Cells were voltage-clamped at a nominal −60 mV (voltage not corrected for junction potential of 8.5 mV). Series resistance in a whole-cell recording was never higher than 8 MΩ and was compensated by 80–90%.

Desensitization time constants were obtained by fitting current decay (Chebyshev algorithm, built-in Clampfit 11.2, Molecular Devices) of the glutamate application from 90% of the peak to the baseline/steady-state current with one or two exponentials. Where biexponential fits were used, weighted $\tau_{w,\mathrm{des}}$ is reported, calculated as follows:

$\tau_{w,\mathrm{des}} = \tau_{f}(A_{f}/(A_{f} + A_{s})) + \tau_{s}(A_{s}/(A_{f} + A_{s}))$, where $\tau_{f(s)}$ and $A_{f(s)}$ represent the fast (slow) component time constant and coefficient, respectively.

Recovery from desensitization was measured using a two-pulse protocol. A conditioning pulse of 10 mM glutamate with a duration of 200 ms was followed by 15 ms glutamate pulses delivered at intervals increasing by 2 ms initially and then, from 40 ms after the conditioning pulse onwards, by 10 ms. The peak current amplitudes of the 15 ms pulses (normalized to the amplitudes of the conditioning pulse) were fitted with a sum of two Hodgkin–Huxley terms $y = y_0 + a_1 \times (1 - \exp(-x \times k_1))^{m_1} + (y_{\max} - a_1 - y_0) \times (1 - \exp(-x \times k_2))^{m_2}$, where $k_1$ and $k_2$ are rates of recovery and $m_1$ and $m_2$ are slopes[94]. Recovery profiles for GluA3 are very steep and good fits could be obtained by fixing the slopes $m_1$ and $m_2$ to 4 and 1. The $y_{\max}$ was constrained to 1. The weighted tau of recovery was calculated as: $\tau_{w} = ((\tau_1 \times a_1) + \tau_2 \times (y_{\max} - a_1 - y_0))/(y_{\max} - y_0)$.

Non-stationary fluctuation analysis was performed on the desensitizing current phase of macroscopic currents evoked with glutamate pulses (10 mM, 200 ms) from outside-out patches containing GluA3-G + TARPγ8 and GluA3-G(R163I) + TARPγ8. The variance ($\sigma^2$) of 20–80 successive responses was grouped in ten amplitude bins, plotted against the mean current, and fitted with the parabolic function $\sigma^2 = i\bar{I} - \bar{I}^2/N - \sigma_o^2$, where $i$ is the single-channel current, $I$ is the mean current, $N$ is the number of channels and $\sigma_o^2$ is the background variance. The weighted mean single-channel conductance ($\gamma$) was obtained from the single-channel current and the holding potential (−60 mV, not corrected for the liquid junction potential).

## Flow cytometry

Surface expression of AMPARs was analysed 48 h after transfection using the Fortessa flow cytometer. HEK293T cells were plated into a 12-well plate and transfected using Effectene (Qiagen) according to the manufacturer's protocol. The AMPAR constructs used for flow cytometry included an HA tag positioned after the signal peptide.

Transfected cells were washed three times with FACS buffer (PBS containing 5% FBS, 1% BSA and 0.05% sodium azide), followed by incubation with an anti-HA antibody conjugated with APC (Miltenyi Biotec, 130-123-553) for 1 h at 4 °C. To remove unbound antibody, cells were washed three additional times with FACS buffer and then resuspended in 300 μl of PBS containing 0.05% sodium azide. APC was excited by 640 nm laser and the emission signal was collected using channel pass filter 670/14. The gating strategy is shown in Supplementary Fig. 2. The mean geometric fluorescence of untransfected cells (APC signal) was subtracted from that of AMPAR-expressing cells. For comparison, fluorescence values of all conditions were normalized to the average of the GluA3-R samples measured on the same day.

## Animals

Hippocampal tissue for organotypic slice culture was extracted from either wild-type C57BL/6JOla (MGI, 3691859) or *Gria1-3*[fl/fl] mice as specified. For generation of the *Gria1-3*[fl/fl] line, mice with floxed loci at Gria1 (JAX, 019012), Gria2 (EM, 09212) and Gria3 (EM, 09215) genes were interbred to give mice homozygous for all floxed alleles, as described previously[10]. All experimental procedures were performed under project license PPL PP5747704 in accordance with the UK Animals (Scientific Procedures) Act of 1986 and approved by the Animal Welfare and Ethical Review Body (AWERB) committee of the MRC Laboratory of Molecular Biology. All animals were housed with unlimited access to food and water on a 12 h–12 h light–dark cycle at room temperature (20–22 °C) and 45–65% humidity. Sample size was based on previous experience with this sort of experiment[14] and there was no randomization or blinding.

## AMPAR knockout by neonatal viral injection

Viral injection into *Gria1-3*[fl/fl] neonates was performed as described previously[95]. In brief, postnatal day 0/1 (P0/1) pups were anaesthetized with isoflurane and injected using a pulled-glass pipette into each

hippocampus with 0.5 µl AAV-hSyn-Cre-eGFP at $3 \times 10^{12}$ genome copies per ml (Addgene, 105540). Pups were returned to the home cage until P6–8, when the hippocampi were extracted for organotypic slice culture.

## Organotypic slice culture

For preparation of organotypic slice culture, hippocampi from P6–8 mice were dissected in ice-cold Gey's balanced salt solution containing 175 mM sucrose, 150 mM NaCl, 2.5 mM KCl, 0.85 mM $NaH_2PO_4$, 0.66 mM $KH_2PO_4$, 2.7 mM $NaHCO_3$, 0.28 mM $MgSO_4$, 2 mM $MgCl_2$, 0.5 mM $CaCl_2$ and 25 mM D-glucose at pH 7.3. Hippocampi were cut using a McIlwain tissue chopper into 300 µm slices that were then grown on Millicell cell culture inserts (Merck) in culture medium (78.5% MEM, 15% heat-inactivated horse serum, 2% B27+ supplement, 2.5% 1 M HEPES, 1.5% 0.2 M GlutaMax supplement, 0.5% 0.05 M ascorbic acid, 1 mM $CaCl_2$, 1 mM $MgSO_4$) at 37 °C and 5% $CO_2$.

## Single-cell electroporation

Single cells from the CA1 region of organotypic hippocampal slices were transfected using an adapted version of the method described previously[96]. DNA plasmids were diluted to 33 ng µl$^{-1}$ at a 1:7 ratio of pN1-eGFP to AMPAR-expressing plasmid in intracellular solution (125 mM KGlu, 20 mM KCl, 4 mM $MgCl_2$, 10 mM HEPES, 4 mM $Na_2$-ATP, 0.3 mM Na-GTP, 0.2 mM EGTA) and back-filled into borosilicate microelectrode pipettes (5–8 MΩ). Slices were placed into the recording chamber sterilized with 70% ethanol and filled with HEPES-based artificial cerebrospinal fluid (aCSF; 140 mM NaCl, 3.5 mM KCl, 1 mM $MgCl_2$, 2.5 mM $CaCl_2$, 10 mM HEPES, 10 mM mM glucose, 1 mM Na-pyruvate, 2 mM $NaHCO_3$). Cells were briefly kept in cell-attached mode and DNA was introduced with a short burst of current pulses (60 pulses at 200 Hz). For transfection of *Gria1-3$^{fl/fl}$* slices, AMPAR-knockout cells were selectively electroporated based on the visualization of nuclear Cre–eGFP. Slices were returned to incubation in their original culture medium supplemented with 5 µg ml$^{-1}$ gentamycin until recording.

## Slice electrophysiology

Transfected hippocampal slice cultures were used for electrophysiological recording 3–4 days after single-cell electroporation. Slices were perfused with aCSF (10 mM D-glucose, 26.4 mM $NaH_2CO_3$, 126 mM NaCl, 1.25 mM $NaH_2PO_4$, 3 mM KCl, 4 mM $MgSO_4$, 4 mM $CaCl_2$) saturated with 95% $O_2$/5% $CO_2$. Then, 100 µM D-AP5, 1 µM SR-95531 and 2 µM 2-chloroadenosine were added to the aCSF for synaptic recording. Borosilicate pipettes (3–5 MΩ whole cell, 5–8 MΩ outside-out) were filled with intracellular solution containing 135 mM $CsMeSO_4$, 4 mM NaCl, 2 mM $MgCl_2$, 10 mM HEPES, 4 mM $Na_2$-ATP, 0.4 mM Na-GTP, 0.15 mM spermine, 0.6 mM EGTA, 0.1 mM $CaCl_2$, adjusted to pH 7.3 with CsOH. All whole-cell recordings were dual, involving simultaneous recording of a neighbouring pair of transfected (GFP positive) and untransfected cells. EPSCs were evoked by Schaffer collateral stimulation at 0.2 Hz in the stratum radiatum at the CA3–CA1 border. EPSCs represent an average of 20 sweeps. Recordings were excluded if the series resistance exceeded 20 MΩ or varied by more than 20%. Somatic RI recordings were made from outside-out patches subjected to fast-exchange perfusion in HEPES-based aCSF (see the 'Single-cell electroporation' section) containing 100 µM cyclothiazide, with or without 1 mM L-glutamate. In voltage-clamp mode, a 500 ms holding potential ramp from −100 mV to +100 mV was applied and current amplitudes at −60 mV, 0 mV and +40 mV, averaged over three sweeps, were used to calculate the RI using ($RI = -(I^{+40} - I^0)/(I^{-60} - I^0)$). Recordings were made using pClamp10 (Molecular Devices) with a Multiclamp 700B amplifier (Axon Instruments), and digitized using a Digidata 1440A (Axon Instruments).

## Reporting summary

Further information on research design is available in the Nature Portfolio Reporting Summary linked to this article.

## Data availability

Cryo-EM coordinates and corresponding EM maps have been deposited in the PDB and EMDB under the following accession codes: apo GluA3-G–γ2 (9HPC, EMD-52325 (LBD–TMD) and 9HPE, EMD-52327 (NTD–LBD)); active/open state GluA3-G–γ2 (9HPK, EMD-52332 (LBD–TMD) and 9HPD, EMD-52326 (NTD–LBD)); desensitized GluA3-G–γ2 NTD–LBD (9HPF, EMD-52328); apo GluA3-G(R163I)–γ2 NTD–LBD (9HPG, EMD-52329). A composite map of the apo GluA3-G–γ2: was deposited under the following codes: 9QFH and EMD-53109. Conventional MD and metadynamics simulations have been deposited to the MDDB and Zenodo. The MD runs were each submitted as a separate record to both databases and have the following MDDB and Zenodo record IDs: apo WT NTD dimer (A01Z9, 14361420 (run 1)[97]; A01ZA, 14361707 (run 2)[98]; and A01ZB, 14361850 (run 3)[99]); apo NTD dimer with in silico mutation R163I (A01ZC, 14364637 (run 1)[100]; A01ZD, 14364695 (run 2)[101]; and A01ZE, 14364713 (run 3)[102]); apo WT NTD–LBD tridomain (A01ZI, 14391563 (run 1)[103]; A01ZJ, 14391600 (run 2)[104]; and A01ZK, 14391602 (run 3)[105]); apo NTD–LBD tridomain with in silico mutation R163I (A01ZL, 15228474 (run 1)[106]; A01ZM, 14397392 (run 2)[107]; and A01ZN, 15228571 (run 3)[108]); intermediate state WT NTD dimer (A020M, 15224099 (run 1)[109]; A020N, 15224387 (run 2)[110]; and A020O, 15224400 (run 3)[111]). The four metadynamics walkers were submitted to the same Zenodo[112] record under ID 14425779 but have four different MDDB IDs (A0202, A0203, A0204 and A0205). Source data are provided with this paper.

## Code availability

ProDy is open source software that is available at GitHub (https://github.com/prody/ProDy), and through the Python package index (PyPI) with pip and the conda-forge channel for conda. Torsion angle measurements for single structures and trajectories were performed using standard structure analysis and trajectory analysis methods (using the getDihedral function) as described on the ProDy website. The source code for our modified DynaMight is publicly available at GitHub (https://github.com/3dem/DynaMight), and further implementation details will be published separately.

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

**Acknowledgements** We thank P. Emsley, L. Catapano and V. Kasaragod for suggestions with model building and Coot figure presentation; O. Paulsen for comments on the brain slice recordings; J. Watson, A. Hall and A. Scrutton for comments on the manuscript; and the members of the Biological Services teams at the LMB and Ares facilities, the LMB scientific computing, the EM facility and the LMB Flow Cytometry facility for support. This work was supported by grants from the Medical Research Council (MC_U105174197) and the Wellcome Trust (223194/Z/21/Z) to I.H.G.; and H2020 Marie Skłodowska-Curie Actions (101024130) and the Spanish Research Council (CSIC; IFERC004) to J.M.K.; C.V.-G. is funded by grant PRE2020-092922 and B.H. is funded by PID2019-106284GA-I00 and RYC2018-025720-I, from MICIN/AEI 0.13039/501100011033 and "ESF Investing in your future" and "ERDF/EU".

**Author contributions** I.H.G. conceived and supervised the study. I.H.G. wrote the paper with input from all of the authors. A.P. performed protein purification and cryo-EM data collection together with B.S. and C.V.-G. EM data processing and model building was done by A.P. with help from J.M.K. and B.S. A.P. analysed cryo-EM data together with J.S. and B.H. I.S. performed and analysed neuronal recordings. J.I. performed and analysed HEK293 cell recordings. J.I. and A.P. performed and analysed the flow cytometry experiments. J.M.K. conducted and analysed all MD simulation. O.C., J.I., A.P. and I.S. created various DNA constructs.

**Competing interests** The authors declare no competing interests.

**Additional information**
**Correspondence and requests for materials** should be addressed to Ingo H. Greger.

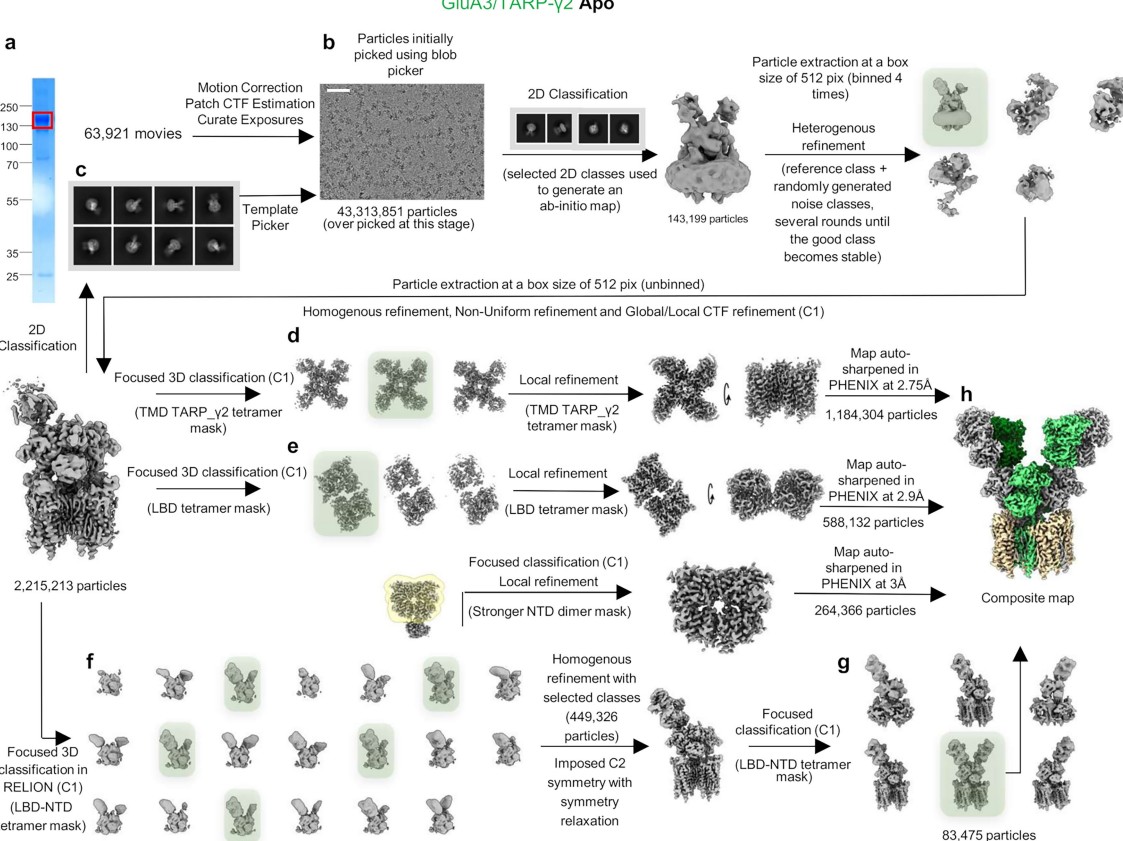

**Extended Data Fig. 1 | Single particle cryo-EM data processing workflow of apo-state GluA3/γ2. a**, A representative 4-12% Bis-Tris SDS-PAGE gel stained with Coomassie blue, indicating elution of GluA3$_{G439}$/γ2 in one of the SEC fractions. For gel source data, see Supplementary Fig. 1. **b**, Representative motion-corrected micrograph of apo state GluA3$_{G439}$/γ2 (scale bar, 50 nm). **c**, Representative 2D class averages of apo state GluA3$_{G439}$/γ2 used for template-based picking. Raw movies were imported in cryoSPARC live for motion correction, CTF estimation, particle picking and exposure curation. Selected particles underwent repeated heterogenous refinement to filter out noise particles and obtain a consensus map of the receptor. After homogenous and non-uniform refinement, the particles corresponding to this map were imported to RELION for further processing. **d**, **e**, and **f**, Focused refinement and classification scheme used for TMD, LBD and an isolated NTD dimer (in yellow) with soft masks, respectively. The best-resolved classes with similar conformations were selected from the focussed classifications. Focused refinement was performed on these particles from these selected classes to improve the resolution using the same masks. Focused refinement of the NTD layer was repeated, applying C2 symmetry and symmetry relaxation with a soft mask that included both NTD dimers. **g**, The resulting particles were 3D classified again with the same NTD mask, but this time with C1 symmetry. The class with both NTD dimers well-resolved was selected. This class was used as a template to create a composite map (shown in **h**) of the apo state GluA3$_{G439}$/γ2, with locally refined TMD, LBD, and NTD maps merged.

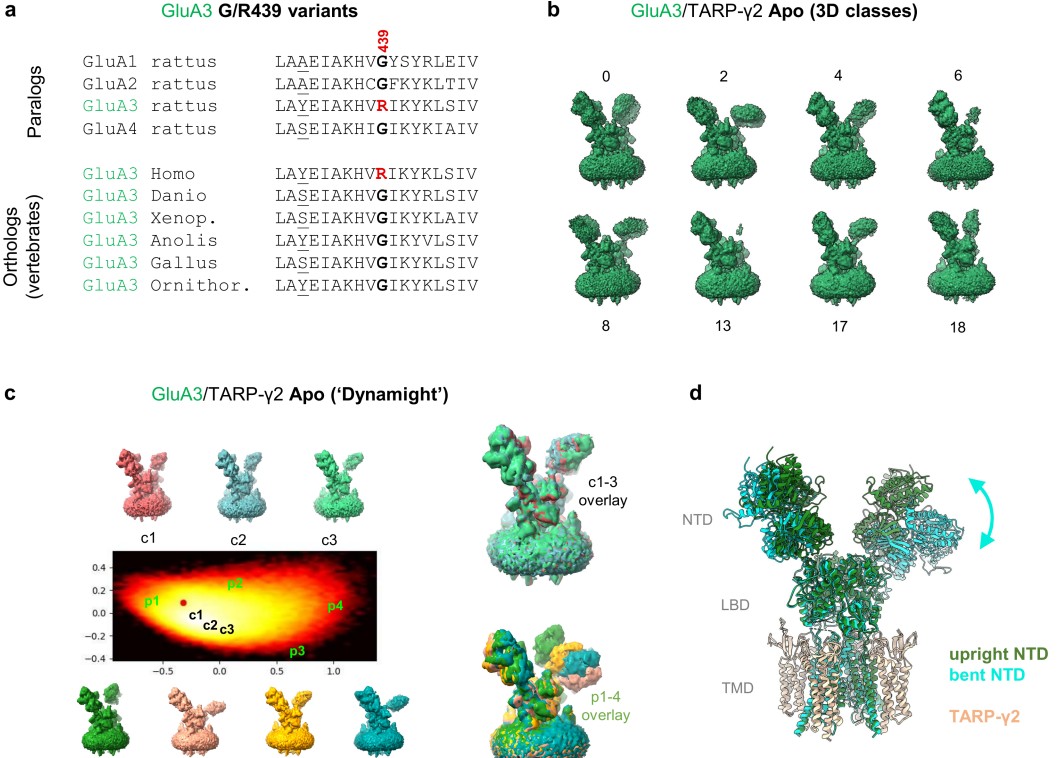

**a** GluA3 **G/R439 variants**

```
                                         439
Paralogs    GluA1 rattus     LAAEIAKHVGYSYRLEIV
            GluA2 rattus     LAAEIAKHCGFKYKLTIV
            GluA3 rattus     LAYEIAKHVRIKYKLSIV
            GluA4 rattus     LASEIAKHIGIKYKIAIV

Orthologs   GluA3 Homo       LAYEIAKHVRIKYKLSIV
(vertebrates) GluA3 Danio    LASEIAKHVGIKYRLSIV
            GluA3 Xenop.     LASEIAKHVGIKYKLAIV
            GluA3 Anolis     LAYEIAKHVGIKYVLSIV
            GluA3 Gallus     LASEIAKHVGIKYKLSIV
            GluA3 Ornithor.  LAYEIAKHVGIKYKLSIV
```

**b** GluA3/TARP-γ2 **Apo (3D classes)**

**c** GluA3/TARP-γ2 **Apo ('Dynamight')**

**d**

**Extended Data Fig. 2 | Conformational spectrum of the heterogenous NTD of GluA3/γ2. a**, Sequence alignment of GluA3 paralogs and orthologs around G439. The G439 variant is predominant in non-mammalian vertebrates and in other AMPAR subtypes. **b**, Representative 3D classes of apo state GluA3$_{G439}$/γ2, showcasing the flexible NTD layer of the receptor. **c**, The log density plot (in the middle) of the latent conformational space of the apo state GluA3$_{G439}$/γ2 from Dynamight. The numbers correspond to the receptor conformation at the given location on the plot. c1-3 (overlaid, top) are the dominant conformations of the receptor near the centre of the plot and p1-4 (overlaid, bottom) are the rarer conformers on the periphery of the plot. **d**, Overlaid atomic models from applying the Dynamight deformations illustrate the two main conformations with two upright NTD dimers (green) or one upright and one bent NTD dimer (cyan). TARP-γ2 is shown in light brown for both conformations.

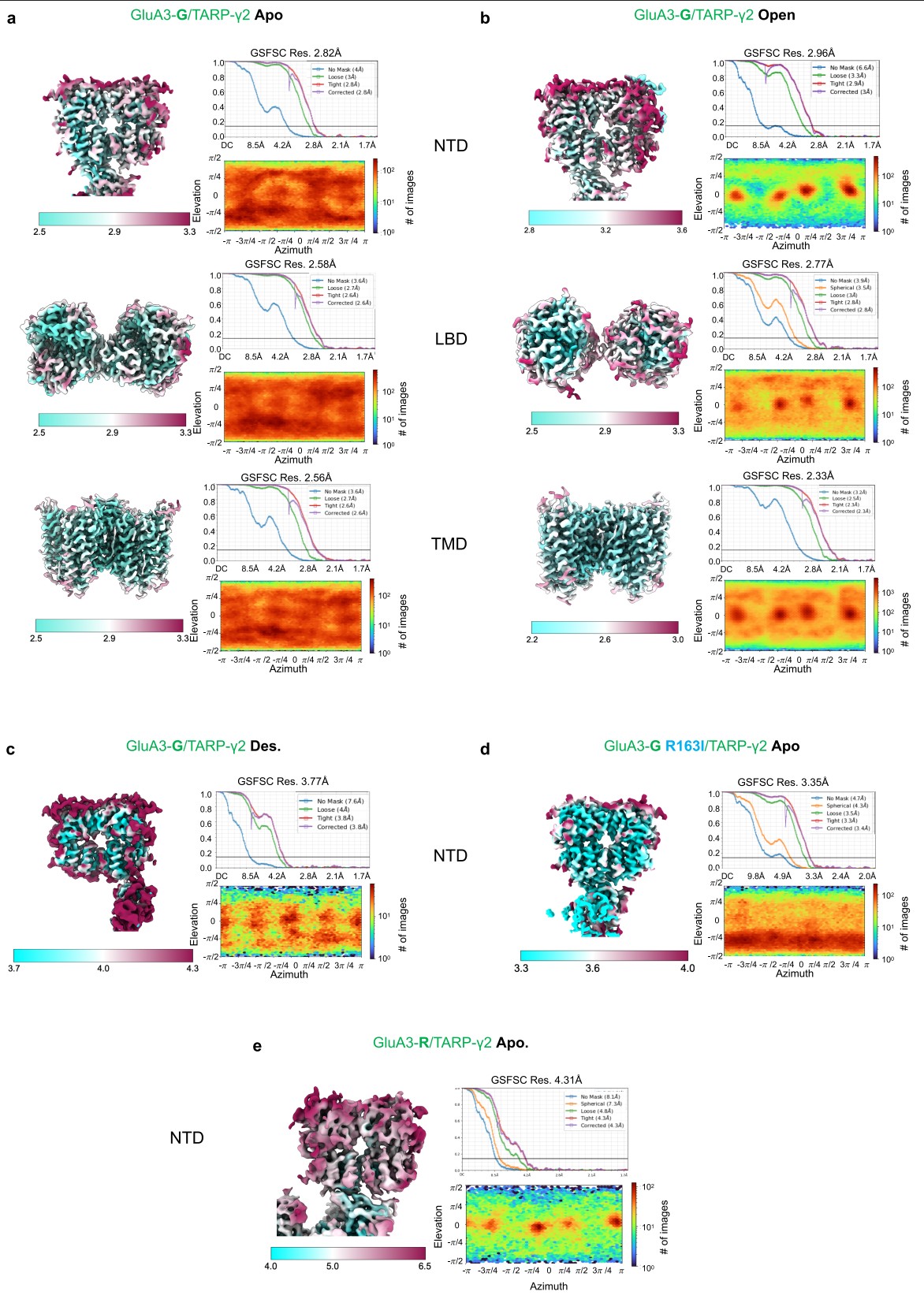

**Extended Data Fig. 3 | Analysis of the GluA3-G/γ2, GluA3-R/γ2 and GluA3-G_R163I/γ2 cryo-EM maps in the apo, open and desensitized states. a**, Local resolution maps of the NTD dimer, LBD and TMD, top to bottom respectively, of the apo state GluA3-G/γ2, coloured based on local resolution estimate as indicated by the colour bar. To the top-right of the local resolution map: Fourier shell correlation (FSC) curves of the corresponding map with the FSC value cut-off at 0.143 and the colours of the curves corresponding to the FSC values with different masks applied before Fourier transform. To the bottom-right of the local resolution map: viewing angle distribution of the particles used for the cryo-EM reconstruction, quantifying the diversity of orientations in the dataset. **b**, Local resolution maps, viewing angle distribution and the FSC curves of the open state GluA3-G/γ2 NTD, LBD and TMD, top to bottom respectively. **c**, Local resolution maps, viewing angle distribution and the FSC curves of the desensitized state GluA3-G/γ2 NTD. **d**, Local resolution maps, viewing angle distribution and the FSC curves of the apo state GluA3-G_R163I/γ2 NTD. **e**, Local resolution maps, viewing angle distribution and the FSC curves of the apo state GluA3-R/γ2 NTD.

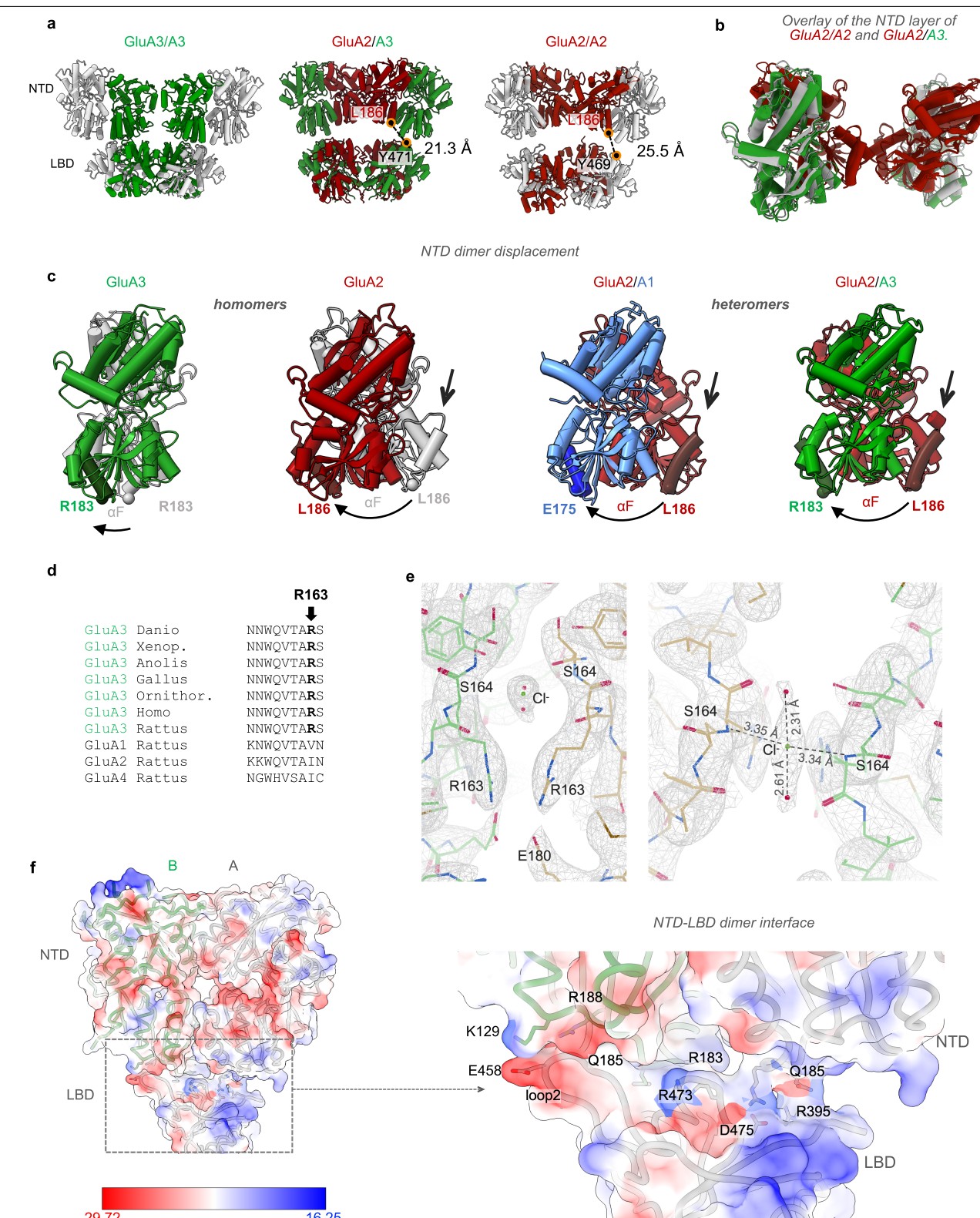

**a**

GluA3/A3    GluA2/A3    GluA2/A2

NTD

LBD

21.3 Å    25.5 Å

L186    Y471    L186    Y469

**b** *Overlay of the NTD layer of GluA2/A2 and GluA2/A3.*

*NTD dimer displacement*

**c**

GluA3    GluA2    GluA2/A1    GluA2/A3

*homomers*    *heteromers*

R183    R183    L186    L186    E175    αF    L186    R183    αF    L186
αF    αF

**d**

R163

GluA3 Danio     NNWQVTA**R**S
GluA3 Xenop.    NNWQVTA**R**S
GluA3 Anolis    NNWQVTA**R**S
GluA3 Gallus    NNWQVTA**R**S
GluA3 Ornithor. NNWQVTA**R**S
GluA3 Homo      NNWQVTA**R**S
GluA3 Rattus    NNWQVTA**R**S
GluA1 Rattus    KNWQVTAVN
GluA2 Rattus    KKWQVTAIN
GluA4 Rattus    NGWHVSAIC

**e**

S164    S164    S164
Cl⁻    Cl⁻    3.35 Å    2.31 Å
R163    R163    2.61 Å    3.34 Å    S164
E180

*NTD-LBD dimer interface*

**f**

B    A

NTD

LBD

-29.72    16.25

K129    R188    NTD
E458    Q185    R183    Q185
loop2    R473    R395
D475    LBD

**Extended Data Fig. 4** | See next page for caption.

**Extended Data Fig. 4 | The GluA3 NTD dimer and the NTD-LBD interface.**
**a**, Atomic models (left to right) of the LBD and the NTD layer of GluA3 homomer (this study), GluA3/A2 heteromer (PDB: 5FWY) and GluA2 homomer (PDB: 3HSY). Residues L186 on the GluA2 NTD, Y469 on GluA2 LBD and Y471 on the GluA3 LBD are used as markers (black-orange balls) to denote the distance between the NTD and the LBD layer. **b**, Overlay of the NTD layer of the GluA3/A2 heteromer and GluA2 homomer. **c**, Atomic models (left to right) of the NTD dimers of GluA3 homomer (this study), GluA2 homomer (PDB: 3HSY), GluA1/A2 heteromer (PDB: 7OCC), and GluA3/A2 heteromer (PDB: 5FWY). Helix F residues R183, L186, and E175 for GluA3, GluA2, and GluA1 respectively, are used as markers to indicate the intradimer displacement. **d**, Sequence alignment of GluA3 paralogs and orthologs around R163. R163 is ubiquitously found in GluA3 across the vertebrate species and is unique amongst the AMPAR subtypes. **e**, Two zoomed views of the candy-shaped ligand density around R163 and S164 reveal potential interactions that justify the tentative modelling of a chloride ion (green sphere). The red spheres in the density map represent water molecules that complete the electrostatic charge network. All distances between the atom pairs are indicated by dashed lines. **f**, Coulombic electrostatic potential (ESP) map of the NTD dimer and interacting LBD region from GluA3/γ2 in the apo state, generated using ChimeraX. The map (zoomed, right) highlights regions of positive and negative electrostatic potential, blue and red, respectively, with the key residues involved in charge interactions at the NTD-LBD interface labelled. The colour gradient represents the ESP ranging from −29.72 to +16.25 kcal/(mol·e).

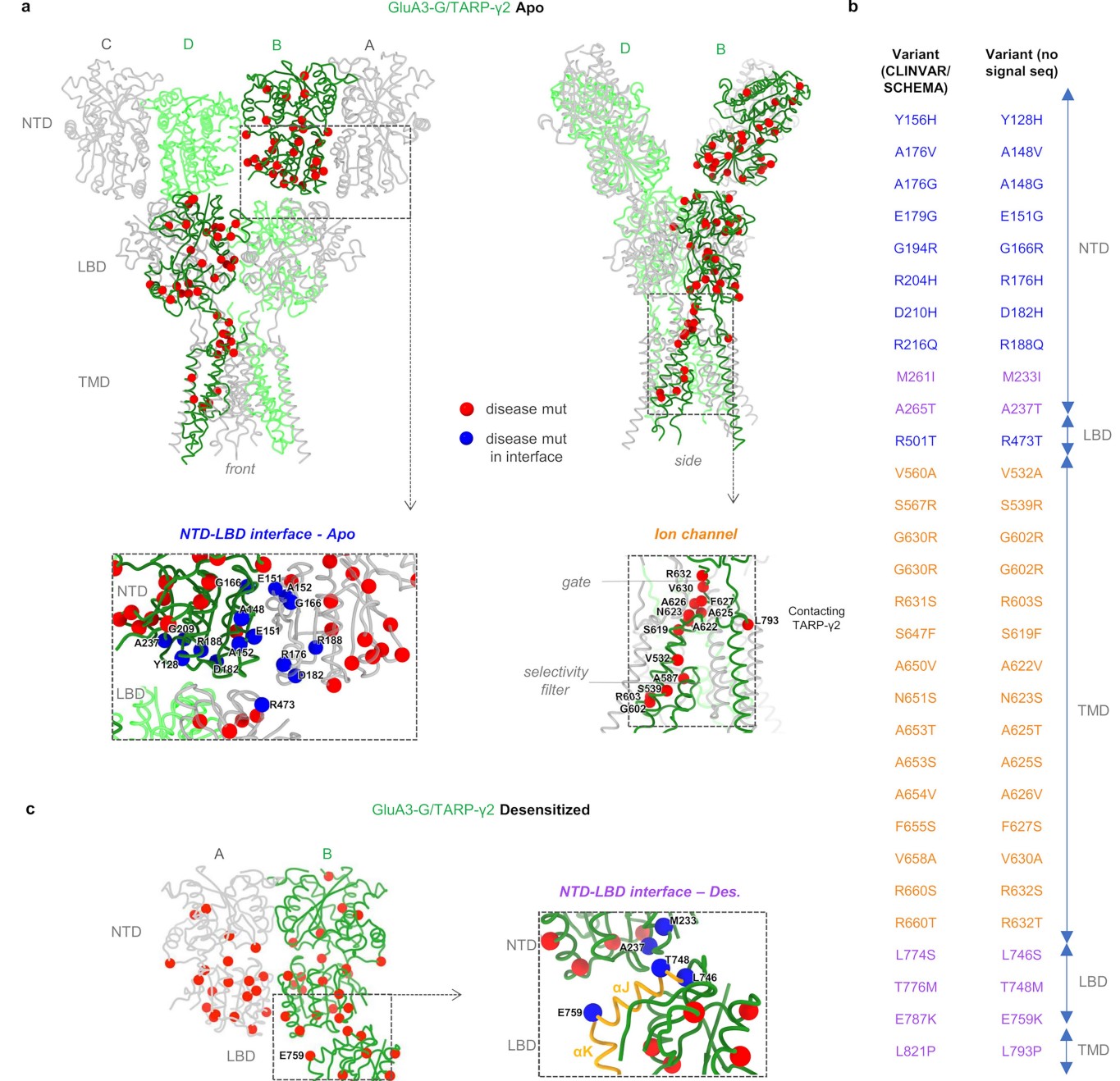

**Extended Data Fig. 5 | Location of human disease mutations on the GluA3-flip receptor.** a, Front (left) and side (right) views of apo-state GluA3-flip. The four subunit chains (A-D) are colour-coded, disease mutations are shown as red spheres. These were derived from the ClinVar (https://www.ncbi.nlm.nih.gov/clinvar/) and Schema (https://schema.broadinstitute.org/gene/ENSG00000125675) databases. Bottom panels: boxed areas present zoomed sections of the NTD-LBD interface (left) and the ion channel (right); mutations locating to either the NTD dimer or the NTD-LBD interface are shown as blue spheres and are labelled (based on mouse GluA3, minus the

signal sequence). b, Left panel: NTD-LBD section of desensitized GluA3. This conformation harbours an alternative NTD-LBD interface, involving the alternatively spliced LBD helix J (see also main text Fig. 3). Mutations locating to this interface are shown in blue (right panel).c, The variants highlighted in the interfaces are listed in pairs in the table, showing the CLINVAR variant on the left and the corresponding variant without the signal sequence on the right. The colours blue, orange and purple correspond to the location of the variant; NTD-LBD interface in the apo state, ion channel in the apo state and NTD-LBD interface in the desensitized state respectively.

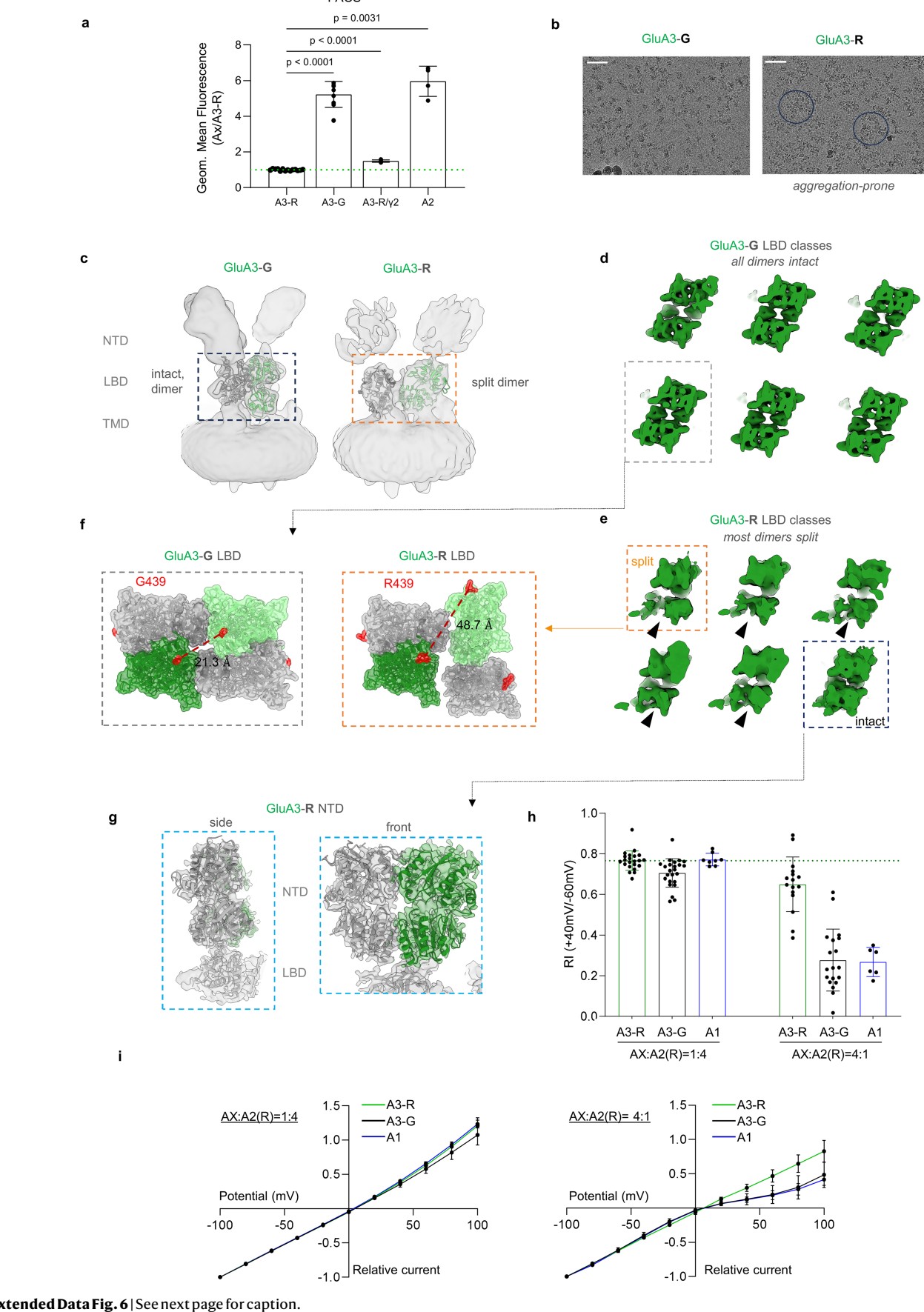

**Extended Data Fig. 6** | See next page for caption.

**Extended Data Fig. 6 | Structural and functional comparison of GluA3-R and GluA3-G. a**, Summary graph (mean ± SD) for flow cytometry data of the surface expression of GluA3-R, -G, GluA2, and GluA3-R/TARPγ2. Geometric mean fluorescence of each sample was normalized to the average value for GluA3-R on that day (indicated by green dashed line). Welch's ANOVA followed by Dunnet's multiple comparison test, ($W_{3, 8.598}$ = 245.9); ****$p$ < 0.0001. **b**, Representative motion-corrected micrographs of the apo-state GluA3-G/γ2 (left) and GluA3-R/γ2 (right) (scale bar, 50 nm). The R-form receptor is prone to aggregation on the cryoEM grids. The blue circles highlight the aggregates on the GluA3-R grids. **c**, Selected 3D classes of GluA3-G/γ2 (left) and GluA3-R/γ2 (right) in the apo state, with the LBD fitted in. The R-form exhibits the splitting of the LBD dimer. **d**, Representative 3D classes of the apo state of GluA3-G/γ2, showcasing that the dimer-of-dimer conformation of the LBD layer is preserved at all times. **e**, Representative 3D classes of the apo state of GluA3-R/γ2, showcasing that the LBD dimers inhabit a continuous spectrum of states ranging from intact LBD dimers (black box) to monomeric, split LBD dimers (orange box). The double-sided arrows point towards the split LBDs in the individual classes. **f**, The surface representation of the models fitted into the selected classes of GluA3-G/γ2 (**c**) and GluA3-R/γ2 (**d**). The red dotted lines denote the distance between R439 (left) and G439 (right) of the B/D subunit, which are 21.3 Å and 48.7 Å, respectively. **g**, The resolved apo state NTD dimer of the GluA3-R form (side and front view, left and right respectively) remains flat and docked on the LBD in a similar way. **h**, Summary bar graphs (mean ± SD) showing the rectification index (RI), defined as the ratio of peak currents recorded at +40 mV to those at −60 mV, determined for two transfection conditions: 4:1 and 1:4 ratios of GluA3-G/-R/A1 to GluA2(R). **i**, Averaged current–voltage (I–V) relationships of AMPARs co-expressed at the indicated transfection ratios, with all currents normalized to the amplitude at −100 mV. Error bars denote SEM.

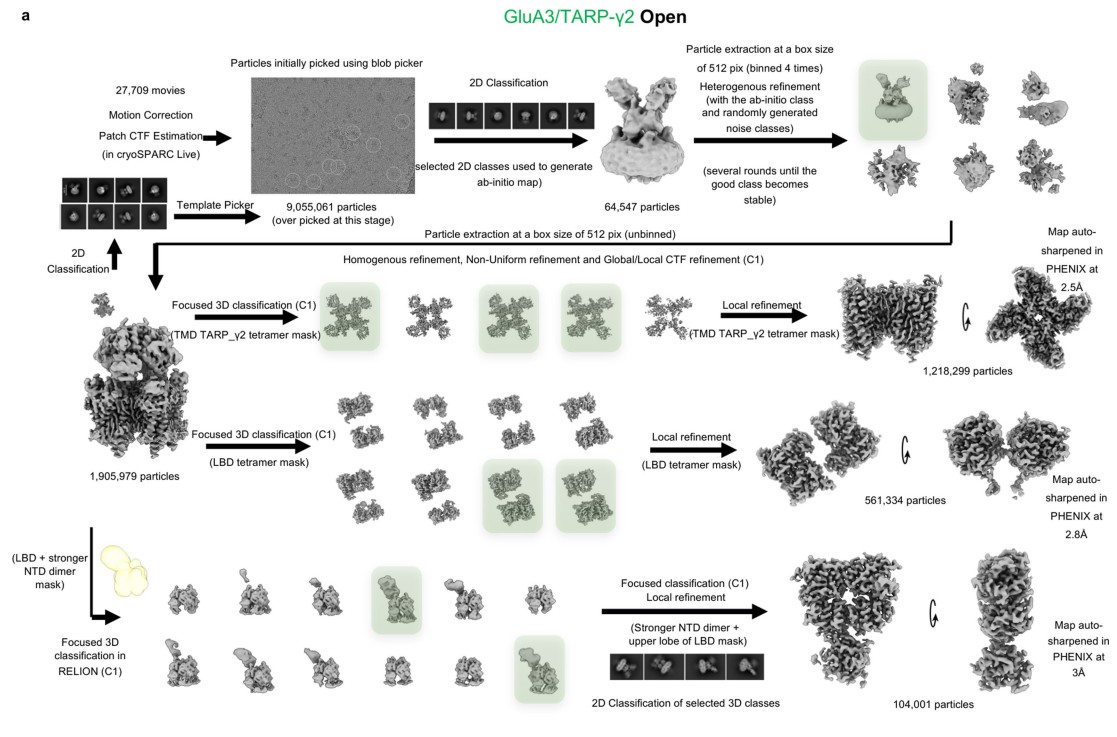

**a**

GluA3/TARP-γ2 **Open**

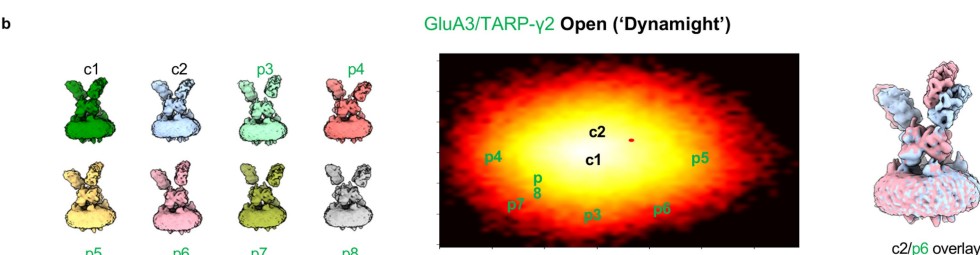

**b**

GluA3/TARP-γ2 **Open ('Dynamight')**

c1  c2  p3  p4

p5  p6  p7  p8

c2/p6 overlay

**Extended Data Fig. 7 | Single particle cryo-EM data processing workflow of open-state GluA3-G/γ2 and analysis of its dynamics. a**, The general cryo-EM image processing workflow for the open state is nearly identical to the apo state, as described in Extended Data Fig. 1. Focused refinement was performed on the NTD, LBD and TMD layers to get improved resolution. **b**, The log density plot (in the middle) of the latent conformational space from Dynamight analysis of the open state GluA3-G/γ2. The numbers correspond to the receptor conformation at the given location on the plot. c1 and c2 are the dominant conformations of the receptor near the centre of the plot and p3-8 are the rarer conformers on the periphery of the plot. c2 and p6 are overlaid on the right to display the stark difference between the two conformers at the NTD level.

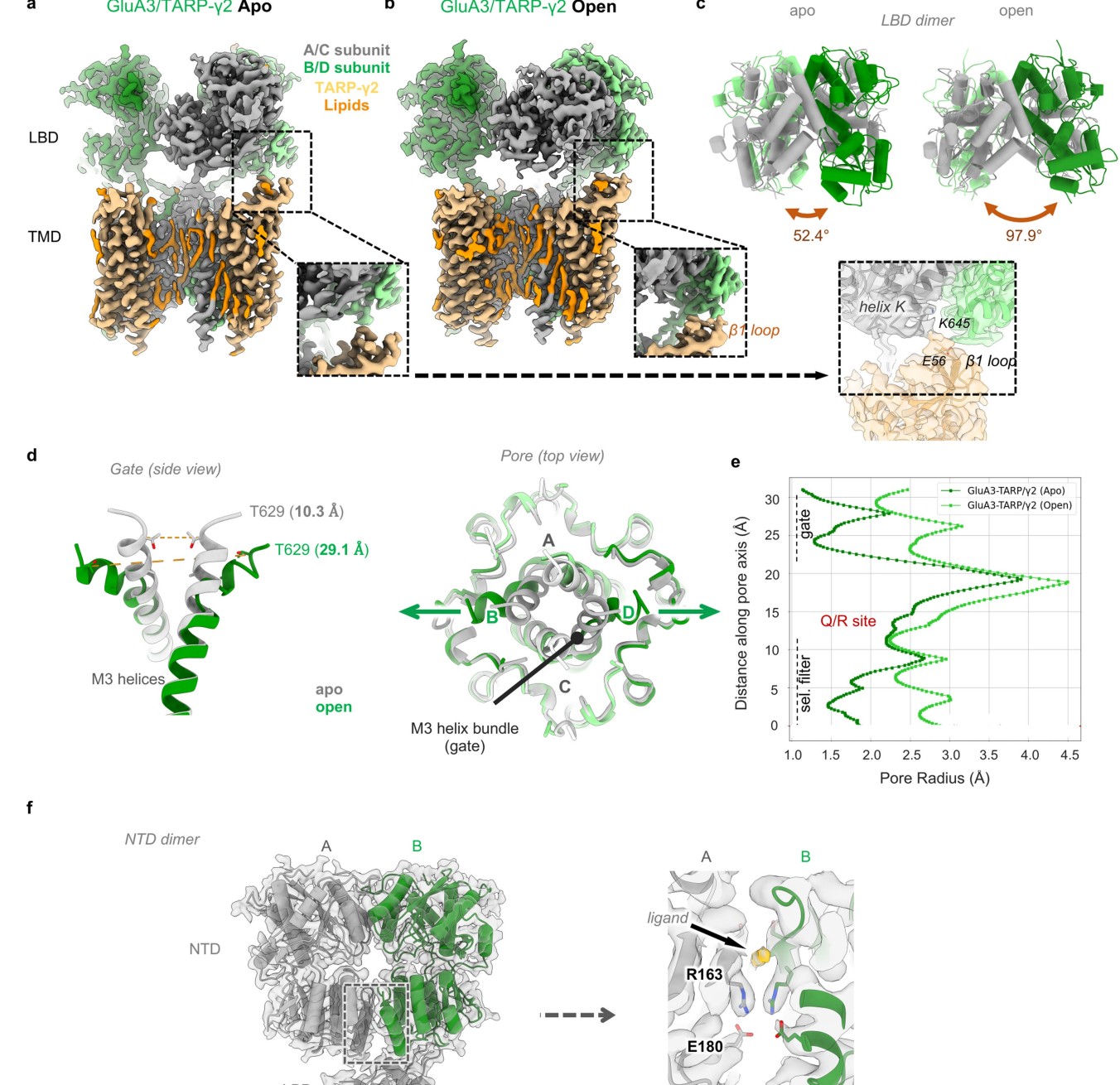

**Extended Data Fig. 8 | Closer look at the conformational changes in GluA3 open state.** A side-by-side comparison of the cryoEM maps of the TMD-LBD layer of GluA3-G/γ2 in the **a**, apo and the **b**, open state. The maps are colour-coded to highlight the different GluA3 subunits, TARP-γ2 and the lipids. The extracellular region of TARP-γ2 in proximity to helix K of the A/C subunit and β1 loop of the B/C subunit is zoomed in to show close contact between the LBD and TARP-γ2, possibly mediated by residues K645 and E56. **c**, A side-by-side comparison of the GluA3-G apo (left) and open (right) LBD layers shows a closing of the LBD clamshells that creates a greater opening at the bottom of the dimers to pull open the TMD ion channel. Both structures are coloured with the A/C chains in grey and the B/D chains in shades of green. **d**, An overlay of the TMD of GluA3 apo (grey) and open (green), showing the M3 gate opening from the side (left, showing just the gating-dominant B/D chains) and top (right, with open A/C chains in light green and open B/D chains in dark green). M3 gate opening is quantified by the change in Cα distance of the chain B/D T629 atoms, which is similar to GluA1 (30.3 Å for open [PDB id: 8C2H] vs 9.8 Å for resting [PDB id: 8C2I] using equivalent residue T621) and GluA1/A2 (29.5 Å for open [PDB id: 6QKZ] vs 10.5 Å for resting [PDB id: 7QHB] using equivalent residue T625 in GluA2). **e**, The pore profile of GluA3-G/γ2 in the apo (dark green) and the open (light green) state is shown with the gate, selectivity filter and the Q/R editing site (in red) outlined. **f**, The resolved open state NTD dimer remains flat and docked on the LBD in a similar way (left), maintaining stacking of R163, stabilized by E180 and an extra density modelled as chlorine atom (right).

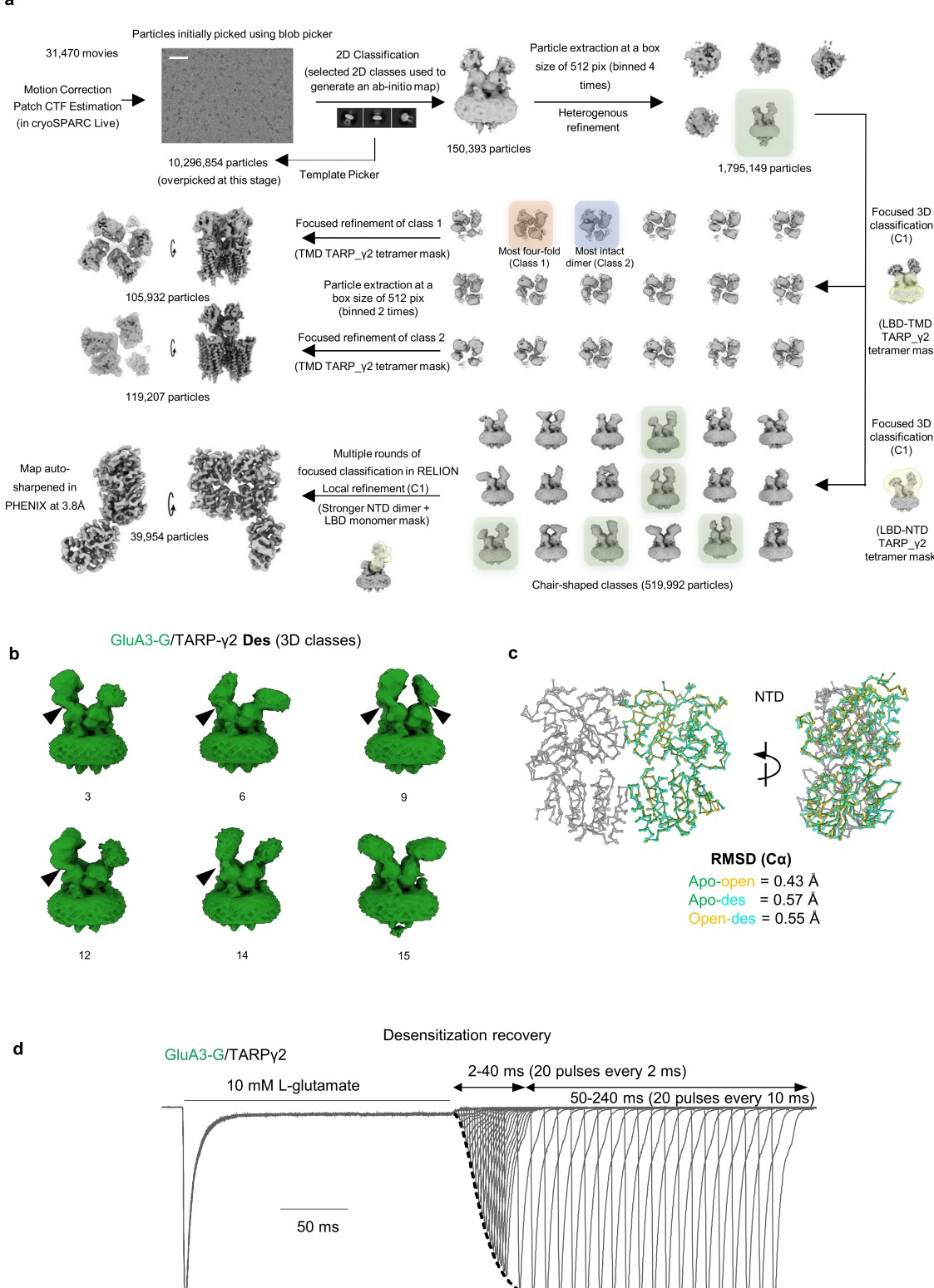

**GluA3-G/TARP-γ2 Desensitized**

31,470 movies

Motion Correction
Patch CTF Estimation
(in cryoSPARC Live)

Particles initially picked using blob picker

2D Classification
(selected 2D classes used to
generate an ab-initio map)

Particle extraction at a box
size of 512 pix (binned 4
times)

150,393 particles

Heterogenous
refinement

1,795,149 particles

10,296,854 particles
(overpicked at this stage)

Template Picker

Focused 3D
classification
(C1)

Focused refinement of class 1

(TMD TARP_γ2 tetramer mask)

Most four-fold
(Class 1)

Most intact
dimer (Class 2)

105,932 particles

Particle extraction at a
box size of 512 pix
(binned 2 times)

(LBD-TMD
TARP_γ2
tetramer mask)

Focused refinement of class 2

(TMD TARP_γ2 tetramer mask)

119,207 particles

Focused 3D
classification
(C1)

Map auto-
sharpened in
PHENIX at 3.8Å

Multiple rounds of
focused classification in RELION
Local refinement (C1)
(Stronger NTD dimer +
LBD monomer mask)

39,954 particles

(LBD-NTD
TARP_γ2
tetramer mask)

Chair-shaped classes (519,992 particles)

**b**

**GluA3-G/TARP-γ2 Des** (3D classes)

3    6    9

12    14    15

**c**

NTD

**RMSD (Cα)**

Apo-open = 0.43 Å
Apo-des   = 0.57 Å
Open-des = 0.55 Å

**d**

**Desensitization recovery**

GluA3-G/TARPγ2

2-40 ms (20 pulses every 2 ms)

10 mM L-glutamate

50-240 ms (20 pulses every 10 ms)

50 ms

**Extended Data Fig. 9 | Single particle cryo-EM data processing workflow of desensitized GluA3-G/γ2. a**, The cryo-EM image processing workflow for the desensitized state closely mirrors that of the apo state (Extended Data Fig. 1). Focused refinement was performed on the NTD-LBD and two distinct LBD conformations—partially broken dimeric and four-fold symmetric—to achieve improved resolution. **b**, Representative 3D classes of the desensitized state GluA3-G/γ2, highlighting the unique conformations specific to this state. **c**, Backbone representation of the atomic models of the GluA3 NTD dimers (front view, left and side view, right) in the apo, open and desensitized state, green, orange and cyan, respectively overlaid on each other. The RMSD of each pair is given below and the low value of each comparison (~0.5 Å) shows that the dimer conformation is preserved across the gating states. **d**, Protocol for recovery from desensitization. Example trace of paired pulse current elicited from whole cell expressing GluA3-G/TARP- γ2 at −60 mV. The dashed black line represents fit to the peak ratios with two-component Hodgkin-Huxley equation with a time constant (τw) of 20.4 ms.

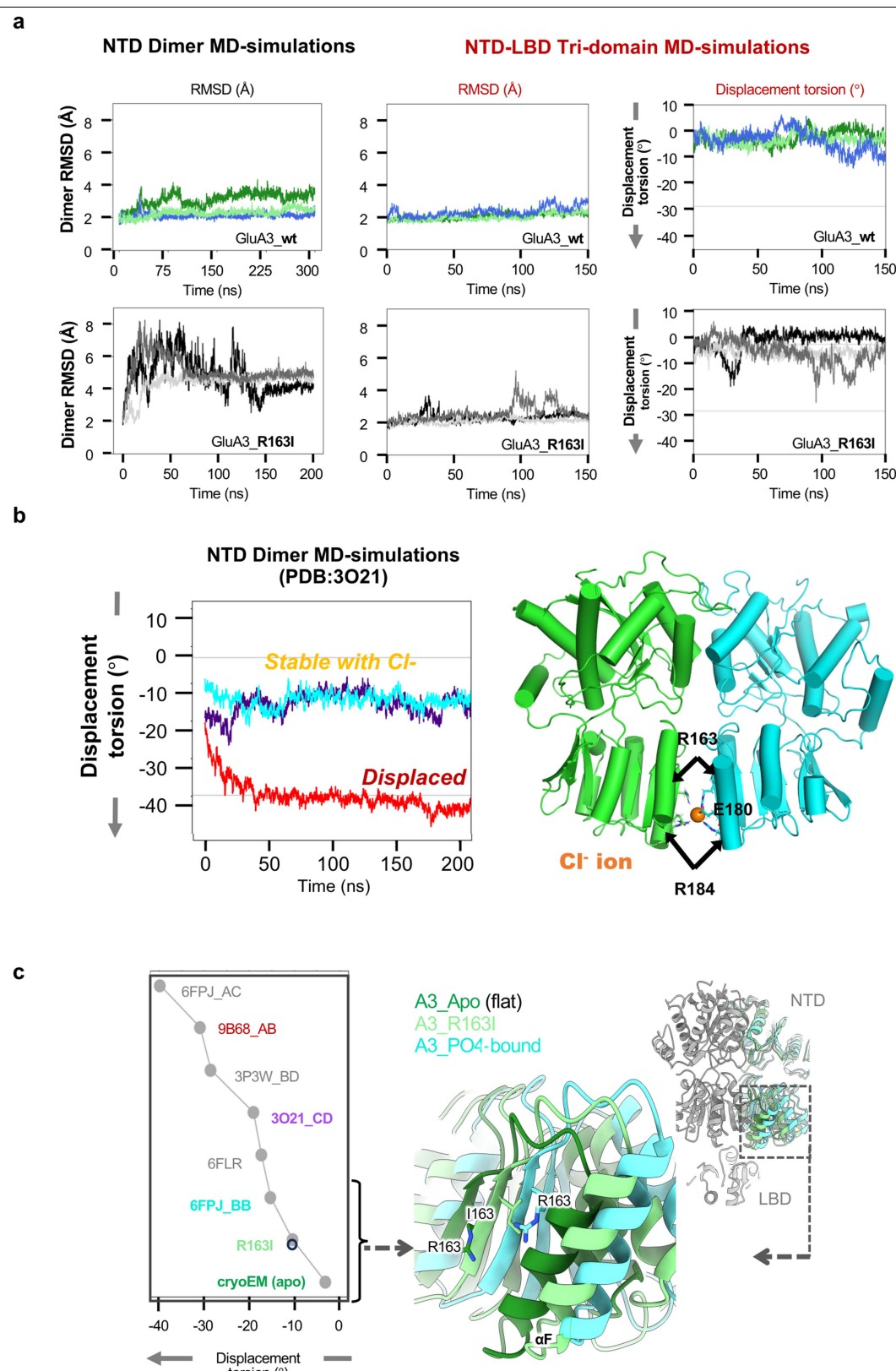

**Extended Data Fig. 10 | NTD dimer conformational variability in MD simulations and experimental structures. a**, The root-mean-square deviation (RMSD) in NTD dimer Cα atom positions are shown for the MD simulations of isolated NTD dimers (left) and of NTD-LBD tridomains (middle), along with NTD dimer displacement torsions for the NTD-LBD tridomain simulations (right) for wt (top) and R163I (bottom). The LBD limits the NTD mobility in the tridomain simulations with R163I exhibiting transient excursions to displaced states with torsions of around −20°, instead of stabilizing around −25° in the NTD only simulations (see Fig. 4b). **b**, Displacement torsion angle progression is shown for 3 MD simulations of a parallel NTD dimer (PDB: 3O21 chains C/D; left). Two of these runs (cyan and purple) show stabilization at −10 degrees, due to binding of a chloride ion (right). The other (red) transitions to displaced. **c**, The resolved experimental R163I structure has an NTD dimer in an intermediate conformation between the phosphate-bound and apo structures as illustrated in the displacement graph (left) and the structural overlay (middle zoomed view and right inset tridomain).

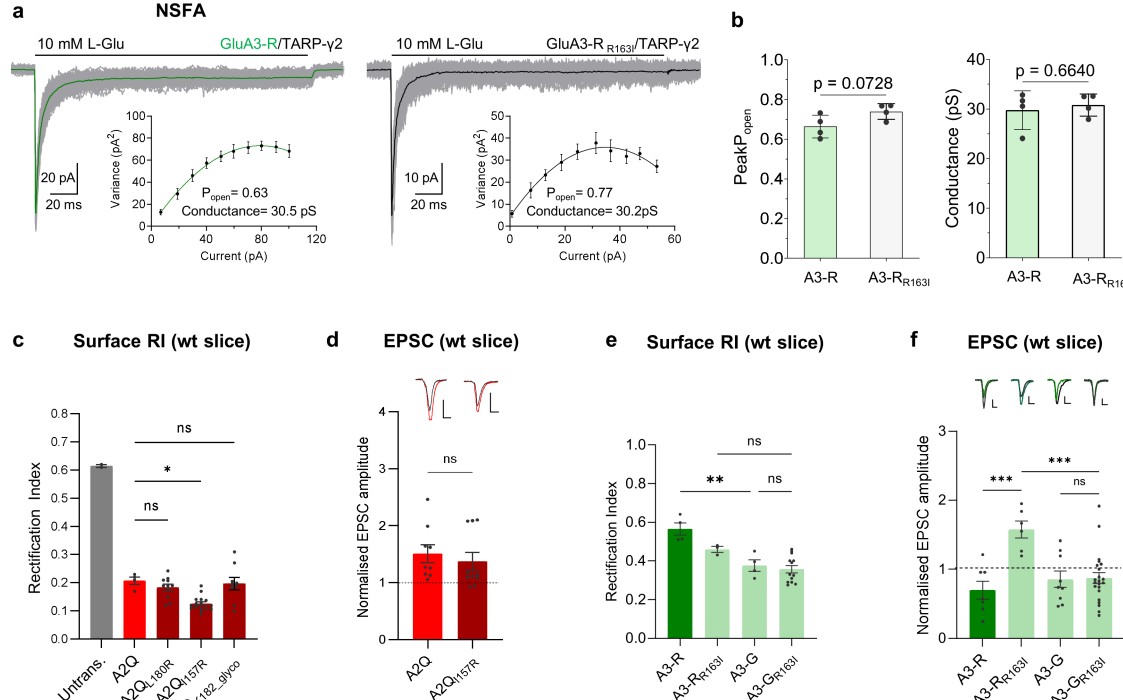

**Extended Data Fig. 11 | Unitary conductance and synaptic rectification properties of GluA3$_{R163I}$. a**, Current responses from an outside-out patch containing GluA3-R/TARP-γ2 (left panel) and GluA3-R$_{R163I}$/TARP-γ2 (right panel) to 10mM L-glutamate, 200 ms, holding voltage −60 mV. The green and black traces show the average traces from the ensemble of responses. Insets plots show the corresponding current-variance relationship, estimated channel conductance (γ), and open probability at the peak (Po). **b**, Bar plots show the mean conductance and open probability at the peak estimates for the GluA3-R/TARP-γ2 (n = 4 patches) and GluA3-R$_{R163I}$/TARP-γ2 (n = 4 patches). The height of the bar presents the mean value, error bars are SD. No difference between the means of the two groups was revealed with the unpaired two-tailed t-tests; t = 0.4566, df = 6, p = 0.6640 (conductance) t = 2.172, df = 6, p = 0.0728 (Peak P$_{open}$). **c**, Surface rectification index from somatic outside-out recordings of the

reciprocal mutations from Fig 5d in GluA2Q. Neither L180R, nor K182_glyco (=K182N R184T) affect surface trafficking of GluA2Q (GluA2Q: 0.21 ± 0.01, GluA2Q$_{L180R}$: 0.18 ± 0.01, GluA2Q$_{K182\_glyco}$: 0.20 ± 0.02). The R163I equivalent mutation in GluA2Q slightly enhances surface expression of GluA2 (GluA2QI$_{157R}$: 0.13 ± 0.01). **d**, A2Q$_{I157R}$ has no effect on the synaptic phenotype of GluA2Q (GluA2Q: 1.5 ± 0.16, GluA2QI$_{157R}$: 1.4 ± 0.16). Two-tailed t test (F = 1.14, DF = 9, p = 0.87) **e**, Rectification index from somatic outside-out recordings of GluA3-R (0.57 ± 0.03) and GluA3-G (0.38 ± 0.02) and the R163I mutation (A3-R$_{R163I}$ = 0.46 ± 0.02, A3-G$_{R163I}$ = 0.36 ± 0.02). One-way ANOVA (F$_{3,20}$ = 11.9). **f**, EPSC amplitude normalized to a neighbouring untransfected neuron. (A3-R = 0.69 ± 0.13, A3-R$_{R163I}$ = 1.6 ± 0.12, A3-G = 0.86 ± 0.11, A3-G$_{R163I}$ = 0.87 ± 0.08). One-way ANOVA (F$_{3,40}$ = 7.9). Bars represent mean ± SEM. Scale bars = 20 pA, 20 ms *p < 0.05, **p = 0.0024, ***p < 0.001. Source data are provided as a source data file.

**Extended Data Table 1 | Glutamate receptor current properties measured from the whole-cell recordings (mean±SD)**

| Receptor/ TARP-γ2 | $\tau_{w, recovery}$ (ms) (n) | $\tau_{w, desensitization}$ (ms) (n) | Steady-state (n) | Peak amplitude (-nA) (n) |
|---|---|---|---|---|
| **GluA3-G constructs[a]** | | | | |
| **GluA3-G** | 20.3± 2.3 (17) | 6.8± 0.7 (9) | 3.2 ± 0.8 (9) | 11 ± 5.2 (17) |
| **GluA3-R** | 23.0± 1.8 (8) p= 0.1586 | 5.4± 0.4 (18) p <0.0001 | 2.2 ± 0.5 (18) p= 0.0179 | 3.3 ± 2.1 (18) p <0.0001 |
| **GluA3-G$_{Q185glyco}$** | 35.0± 4.4 (8) p <0.0001 | 4.0± 0.5 (8) p <0.0001 | 1.2 ± 0.3 (8) p <0.0001 | 8.0 ± 5.0 (8) p= 0.6835 |
| **GluA3-G$_{K129C}$** | 20.4 ± 3.3 (7) p >0.9999 | 6.4 ± 0.6# (7) | 2.9 ± 0.6# (7) | 11.2 ± 4.3 (7) p >0.9999 |
| **GluA3-G$_{E458C}$** | 19.0 ± 2.8 (6) p > 0.9433 | 6.0 ± 0.39# (6) | 2.2 ± 0.6# (6) | 2.1 ± 1.5 (6) p <0.0001 |
| **GluA3-G$_{K129C/E458C}$** | 23.5±1.1 (6) p = 0.1071 | 10.0 ± 1.8# (6) | 29.0 ± 6.1# (6) | 6.3 ± 3.0 (6) p= 0.0635 |
| **GluA3-G$_{G235A}$** | 14.7 ± 2.2 (7) p= 0.0002 | 7.3 ± 1.1 (7) p= 0.4352 | 4.4 ± 1.3 (7) p= 0.2462 | 13 ± 7.0 (7) p= 0.9954 |
| **GluA3-G$_{G235C}$** | 12.6±2.0 (7) p <0.0001 | 7.8±0.7 (7) p = 0.0186 | 8.4±1.6 (7) p= 0.0002 | 6.2±6.0 p= 0.3734 (7) |
| **GluA3-G$_{T748C}$** | 30.8 ± 3.5 (7) p <0.0001 | 3.3 ± 0.8 (7) p <0.0001 | 1.8 ± 0.3 (7) p= 0.0021 | 5.5 ± 3.0 (7) p= 0.0167 |
| **GluA3-R constructs[b]** | | | | |
| **GluA3-R** | 22.7± 2.0 (9) | 5.50± 0.48 (23) | 2.4± 0.7 (23) | 3.5 ± 2.6 (23) |
| **GluA3-R $_{R163I}$** | 20.9 ± 1.9 (10) p= 0.3613 | 5.0 ± 0.4 (31) p= 0.0014 | 1.8 ± 0.5 (31) p= 0.0170 | 3.8 ± 2.1 (31) p= 0.9997 |
| **GluA3-R$_{E180A}$** | 20.3 ± 2.0 (6) p= 0.2256 | 5.4± 0.4 (13) p >0.9999 | 1.8 ± 0.7 (13) p= 0.1459 | 2.7 ± 2.3 (13) p= 0.9526 |
| **GluA3-R$_{R183A}$** | 25.6 ± 2.6 (7) p= 0.0715 | 5.4 ± 0.4 (14) p= 0.9998 | 2.1 ± 0.7 (14) p= 0.8551 | 5.5 ± 2.3 (14) p =0.3983 |
| **GluA3-R$_{R188Q}$** | 21.4 ± 1.51 (7) p= 0.7577 | 5.2 ± 0.4 (7) p= 0.6742 | 2.0 ± 0.8 (7) p= 0.7396 | 5.3 ± 3.0 (7) p= 0.6253 |
| **GluA3-R$_{R473T}$** | 28.7 ± 3.0 (4) p= 0.0003 | 5.0 ± 0.3 (13) p= 0.0146 | 1.8 ± 0.9 (13) p= 0.2550 | 1.0 ± 0.9 (13) p = 0.0014 |
| **GluA3-R$_{Q185glyco}$** | 31.1± 3.6 (6) p <0.0001 | 3.4 ± 0.3 (23) p <0.0001 | 1.0 ± 0.3 (23) p <0.0001 | 1.5 ± 0.7 (23) p = 0.0070 |
| **GluA3-R$_{G235C}$** | 12.2±1.7 (10) p <0.0001 | 7.5±0.7 (10) p <0.0001 | 7.7±2.1 (10) p= 0.0001 | 5.3±3.3 (10) p = 0.6901 |
| **GluA3-R$_{K129C/E458C}$** | ND | 9.2 ± 0.9 (9) p <0.0001 | 21.0± 4.9 (9) p <0.0001 | 0.5 ± 0.4 (9) p <0.0001 |
| **GluA2 constructs[c]** | | | | |
| **GluA2** | 16.4 ± 1.5 (9) | 12.0 ± 1.1 (9) | 10.0 ± 2.6 (9) | 17.0 ± 8.0 (9) |
| **GluA2$_{G232C}$** | 18.8 ± 1.7 (12) p= 0.0025 | 11.4 ± 0.7 (12) p= 0.1261 | 10.1 ± 2.0 (12) p= 0.8927 | 10.0 ± 1.0 (12) p= 0.0002 |

[a]**Comparisons with GluA3-G TARP-γ2**: The effect of mutations was determined by Ordinary one-way ANOVA, ($F_{(8, 64)}$=50.71); ****$p < 0.0001$ (recovery from desensitization), ordinary one-way ANOVA ($F_{5, 50}$=54.49); ****$p < 0.0001$ (desensitization entry), Welch's ANOVA ($W_{5, 19.27}$=38.27); ****$p < 0.0001$ (steady state) and Welch ANOVA ($W_{8, 24.16}$=9.203); ****$p < 0.0001$ (peak amplitude), all followed with the Dunnett's multiple comparisons test, p-values are indicated.

[b]**Comparisons with GluA3-R TARP-γ2:** The effect of mutations was determined by Ordinary one-way ANOVA, ($F_{7, 51}$=50.66); ****$p < 0.0001$ (recovery from desensitization), Welch's ANOVA ($W_{8, 40.45}$=131.1); ****$p < 0.0001$ (desensitization entry), Welch's ANOVA ($W_{8, 39.11}$=39.81); ****$p < 0.0001$ (steady state) and Welch's ANOVA ($W_{8, 42.18}$=17.53); ****$p < 0.0001$ (peak amplitude), all followed with the Dunnett's multiple comparisons test, p-values are indicated.

[c]**GluA2 constructs comparisons**: No difference between the means of the two groups was revealed with the unpaired two-tailed t-tests; t=1.6, df = 19, p=0.01261 (desensitization entry) and t=0.1366, df=19, p=0.8927 (steady state). Significant difference between the mean values of recovery from desensitization was revealed with the unpaired two-tailed t-tests; t=3.473, df = 19, p=0.0025). Significant difference between the mean values of peak amplitude was revealed with the unpaired two-tailed t-tests with Welch's correction; t=4.676, df = 16.79, p=0.0002.
[c]data used for statistical analysis in Fig. 2 and not taken into consideration.
ND - not determined.

# Reporting Summary

## Statistics

For all statistical analyses, confirm that the following items are present in the figure legend, table legend, main text, or Methods section.

| n/a | Confirmed | |
|---|---|---|
| ☐ | ☒ | The exact sample size (*n*) for each experimental group/condition, given as a discrete number and unit of measurement |
| ☐ | ☒ | A statement on whether measurements were taken from distinct samples or whether the same sample was measured repeatedly |
| ☐ | ☒ | The statistical test(s) used AND whether they are one- or two-sided<br>*Only common tests should be described solely by name; describe more complex techniques in the Methods section.* |
| ☒ | ☐ | A description of all covariates tested |
| ☐ | ☒ | A description of any assumptions or corrections, such as tests of normality and adjustment for multiple comparisons |
| ☐ | ☒ | A full description of the statistical parameters including central tendency (e.g. means) or other basic estimates (e.g. regression coefficient) AND variation (e.g. standard deviation) or associated estimates of uncertainty (e.g. confidence intervals) |
| ☐ | ☒ | For null hypothesis testing, the test statistic (e.g. *F*, *t*, *r*) with confidence intervals, effect sizes, degrees of freedom and *P* value noted<br>*Give P values as exact values whenever suitable.* |
| ☒ | ☐ | For Bayesian analysis, information on the choice of priors and Markov chain Monte Carlo settings |
| ☒ | ☐ | For hierarchical and complex designs, identification of the appropriate level for tests and full reporting of outcomes |
| ☒ | ☐ | Estimates of effect sizes (e.g. Cohen's *d*, Pearson's *r*), indicating how they were calculated |

*Our web collection on statistics for biologists contains articles on many of the points above.*

## Software and code

Policy information about availability of computer code

| Data collection | pClamp11.2, gromacs 2023, gromacs 5.0.4, plumed 2.1.3 |
|---|---|
| Data analysis | RELION 5.0, cryoSPARC v4.4.1, coot 0.9.8.95, PHENIX 1.20, REFMAC5, Servalcat, UCSF Chimera 1.14, ChimeraX-1.8, Pymol 1.8.2.0, MolProbity v4.2, ProDy 2.5.0, Clampfit 11.2, GraphPad Prism 10.3.1 |

For manuscripts utilizing custom algorithms or software that are central to the research but not yet described in published literature, software must be made available to editors and reviewers. We strongly encourage code deposition in a community repository (e.g. GitHub). See the Nature Portfolio guidelines for submitting code & software for further information.

## Data

Policy information about availability of data

All manuscripts must include a data availability statement. This statement should provide the following information, where applicable:

- Accession codes, unique identifiers, or web links for publicly available datasets
- A description of any restrictions on data availability
- For clinical datasets or third party data, please ensure that the statement adheres to our policy

Cryo-EM coordinates and corresponding EM maps are deposited in the PDB and EMDB under the following accession codes: Apo GluA3G439/γ2: 9HPD/EMD-52326 (LBD–TMD) and 9HPE/EMD-52327 (NTD-LBD); Active/open state GluA3G439/γ2: 9HPK/EMD-52332 (LBD–TMD) and 9HPC, EMD-52325 (NTD-LBD); Desensitized GluA3G439/γ2 NTD-LBD: 9HPF, EMD-52328; Apo GluA3G439,R163I/γ2 NTD-LBD: 9HPG/EMD-52329. Conventional MD and metadynamics simulations have been

deposited to the MDDB and accession codes will arrive following maintainer curation.

Source data for all electrophysiology experiments is also provided in a spreadsheet.

# Research involving human participants, their data, or biological material

Policy information about studies with human participants or human data. See also policy information about sex, gender (identity/presentation), and sexual orientation and race, ethnicity and racism.

| | |
|---|---|
| Reporting on sex and gender | N/A |
| Reporting on race, ethnicity, or other socially relevant groupings | N/A |
| Population characteristics | N/A |
| Recruitment | N/A |
| Ethics oversight | N/A |

Note that full information on the approval of the study protocol must also be provided in the manuscript.

# Field-specific reporting

Please select the one below that is the best fit for your research. If you are not sure, read the appropriate sections before making your selection.

☒ Life sciences          ☐ Behavioural & social sciences          ☐ Ecological, evolutionary & environmental sciences

For a reference copy of the document with all sections, see nature.com/documents/nr-reporting-summary-flat.pdf

# Life sciences study design

All studies must disclose on these points even when the disclosure is negative.

| | |
|---|---|
| Sample size | No statistical method was used to determine sample size.  Cryo-EM sample sizes were determined by available electron microscopy time and the number of particles on electron microscopy grids. The sample size is sufficient to obtain a structure at the reported resolution, as assessed by Fourier shell correlation. Electrophysiology sample sizes were determined based on literature review, previous experience with data of this sort, and reproducibility of results across independent experiments. The authors have extensive previous experience with data of this type (Zhang, Nature 2021; Herguedas, Science 2019; Herguedas, Science 2016; Cais, Cell Reports 2014)., therefore sample sizes were based on understanding of sample variabilities. |
| Data exclusions | During cryo-EM data processing, data were excluded using standard classification approaches in cryoSPARC and RELION to remove false picks and particle images without high resolution content. In electrophysiology experiments, data were excluded based on pre-established quality control criteria (rise time, holding current). |
| Replication | All cryo-EM structures were determined from independent half datasets, which were compared to assess the resolution of the reconstruction. All electrophysiology data sets were pooled from at least two independent experiments and all results were successfully replicated. |
| Randomization | For Cryo-EM, division of datasets into two random halves was done based on standard approach in RELION. Randomization is not relevant to electrophysiology. |
| Blinding | Blinding was not applicable to cryo-EM or MD simulations, because this type of study does not use group allocation. Researchers were not blinded for the acquisition or analysis of electrophysiology data as it was not technically or practically feasible to do so. Experimenter independence was ensured by application of defined exclusion criteria as stated above. |

# Reporting for specific materials, systems and methods

We require information from authors about some types of materials, experimental systems and methods used in many studies. Here, indicate whether each material, system or method listed is relevant to your study. If you are not sure if a list item applies to your research, read the appropriate section before selecting a response.

## Materials & experimental systems

| n/a | Involved in the study |
|-----|----------------------|
| ☒ | ☐ Antibodies |
| ☐ | ☒ Eukaryotic cell lines |
| ☒ | ☐ Palaeontology and archaeology |
| ☐ | ☒ Animals and other organisms |
| ☒ | ☐ Clinical data |
| ☒ | ☐ Dual use research of concern |
| ☒ | ☐ Plants |

## Methods

| n/a | Involved in the study |
|-----|----------------------|
| ☒ | ☐ ChIP-seq |
| ☐ | ☒ Flow cytometry |
| ☒ | ☐ MRI-based neuroimaging |

# Eukaryotic cell lines

Policy information about cell lines and Sex and Gender in Research

| | |
|---|---|
| Cell line source(s) | HEK293Tcells were purchased from ATCC and HEK-Expi293F cells from ThermoFisher Scientific (Cat# A14527). |
| Authentication | No further authentication was performed for cell lines used in this study. |
| Mycoplasma contamination | No mycoplasma testing was performed specifically for this study, the HEK293T cell line had been tested negative in the past. |
| Commonly misidentified lines (See ICLAC register) | HEK cells are listed in the register; however, our HEK cell lines come from reliable source and are the only secondary cell type used in this study, which minimizes the risk of any cross-contamination. |

# Animals and other research organisms

Policy information about studies involving animals; ARRIVE guidelines recommended for reporting animal research, and Sex and Gender in Research

| | |
|---|---|
| Laboratory animals | C57/Bl6 mice of both sexes were used in this study at age postnatal day 6-8. Animals were housed with unlimited access to food and water under a standard 12 hour light-dark cycle, at normal room temperature (approx 20-22 degrees Centigrade). Pregnant mothers were monitored daily, and P0 refers to the day of litter discovery. |
| Wild animals | No wild animals were used in this study. |
| Reporting on sex | Organotypic slices were prepared from pups of both sexes. There iare no reported or discernible differences between sexes in electrophysiological properties of slices prepared at age P6-8. |
| Field-collected samples | No field collected samples were used in this study. |
| Ethics oversight | All procedures were carried out under PPL PP5747704 in accordance with UK Home Office regulations. Experiments conducted in the UK are licensed under the UK Animals (Scientific Procedures) Act of 1986 following local ethical approval. |

Note that full information on the approval of the study protocol must also be provided in the manuscript.

# Plants

| | |
|---|---|
| Seed stocks | N/A |
| Novel plant genotypes | N/A |
| Authentication | N/A |

# Flow Cytometry

## Plots

Confirm that:

☒ The axis labels state the marker and fluorochrome used (e.g. CD4-FITC).

☒ The axis scales are clearly visible. Include numbers along axes only for bottom left plot of group (a 'group' is an analysis of identical markers).

☒ All plots are contour plots with outliers or pseudocolor plots.

☒ A numerical value for number of cells or percentage (with statistics) is provided.

## Methodology

| | |
|---|---|
| Sample preparation | Samples consisted of HEK293T cells transiently transfected with AMPAR subunits, with an HA tag inserted into the N-terminal domain following the signal peptide. Cells were harvested from 12-well plates using ice-cold PBS supplemented with 5% FBS, 1% BSA, and 0.05% sodium azide. Antibody incubation was performed in a final volume of 50 µL for 1 hour on ice. Following incubation, cells were washed three times with the same buffer and resuspended in 300 µL PBS containing 0.05% sodium azide immediately prior to analysis on an LSRFortessa flow cytometer. A minimum of 30,000 cells were collected per sample for analysis. |
| Instrument | BD LSRFortessa |
| Software | FlowJo (version 10.8.0). |
| Cell population abundance | A total of at least 30000 cells were taken for mean geometric fluorescence calculation |
| Gating strategy | Gating was used only to include the cells rather than debris. Further gating was not used. |

☒ Tick this box to confirm that a figure exemplifying the gating strategy is provided in the Supplementary Information.

