## [Peer Review File · Nature]

Architecture, dynamics and biogenesis of GluA3 AMPA glutamate receptors.

Corresponding Author: Dr Ingo Greger

Version 1:

Reviewer comments:

Referee #1

(Remarks to the Author)

In this study, Pokharna and colleagues present high-resolution cryo-electron microscopy structures of GluA3 AMPA receptors captured in closed, open, and desensitized conformational states. Their structural analysis reveals distinctive features of GluA3, particularly in the interfaces between N-terminal domains (NTD-NTD) and between the N-terminal and ligand-binding domains (NTD-LBD), which differentiate it from both homomeric GluA1 and GluA2 receptors, as well as from heteromeric GluA1/2 and GluA2/3 assemblies. Using targeted mutagenesis and electrophysiological recordings, the authors validate the functional significance of these structural states. Furthermore, they investigate how the GluA3 N-terminal domain influences both surface expression and synaptic targeting of AMPA receptors in neurons.

The findings are novel, and the technical quality of the data is exceptional, as expected from this lab. However, I'm unsure if the structural findings really allow as comprehensive a dissection of GluA3 physiology as the authors suggest. I have several criticisms:

First, the authors' characterization of GluA3 as "ultrafast" in the title and abstract lacks precise definition. The specific kinetic parameters that constitute this designation - presumably activation, deactivation, desensitization, or resensitization - are not clearly specified. Therefore, although the paper seems to promise structural details that explain unusually rapid kinetics of GluA3 (compared to e.g. the equivalent flip/flop or R/G editing forms of GluA2?), it neither presents evidence for unique kinetic behavior nor provides a structural argument for how it might come to pass.

Second, regarding GluA3 trafficking, while the role of arginine at the R/G 439 'trafficking checkpoint' is long-established, the current study does not substantially advance our mechanistic understanding. Although this residue is positioned near the LBD/NTD interface identified in desensitized receptors, the authors do not present structural modelling or a clear hypothesis explaining how this site regulates trafficking. Given that the G/R residue minimally affects mature receptor kinetics, the dramatic impact on surface expression likely involves mechanisms not revealed by the data.

Third, the functional significance of the desensitized GluA3 LBD-NTD interface remains questionable. While the LBD half of the interface incorporates elements of the flip/flop cassette near the R/G editing site, it doesn't seem like these contacts will be that important in mediating the functional effects of post-transcriptional processing given that GluA2, which lacks this interface, shows similar effects of the same modifications.

Fourth, the analysis of disease-associated mutations, while valuable, could be more comprehensive. The authors chose to examine artificial mutations (e.g., R188A, R473A) rather than patient-specific variants (R188Q, R473T). The significance given to certain structural proximities (such as A237T near to Gly235) may be coincidental given the number of NTD variants and their broad distribution. Additionally, the finding of Peng et al that E759G causes complete loss of function is more likely due to compromised agonist binding or protein folding than disruption of a desensitized LBD-NTD interface.

A limitation of the mutagenesis studies is the absence of reciprocal experiments in GluA2, (with the exception of domain transplantation). Testing equivalent mutations in GluA2 (e.g., at positions 163/164/180, the cys129/458 pair, Gly235 and Arg183) would have strengthened the authors' conclusions that these are uniquely important to the novel structural features of GluA3. i.e. if the functional/synaptic results of changes to GluA3 are solely due to modifying structural parameters, the equivalent changes in GluA2 should be functionally silent/have a different impact.

Minor Comments/Technical Points

Abstract. the phrase "...the ultrafast conducting GluA3 and GluA4 receptors, prevalent in interneurons and the brainstem" is used. The citation used to support this statement seems inappropriate as it does not mention ultrafast or 'ultrafast conductance', nor does it explicitly refer to the conductance of GluA3 or GluA4 or to brainstem neurons.

Line 64. "CP-AMPA subunits segregate into the slowly gating GluA1, and the fast GluA3 and GluA4 receptors, a subgroup

selectively enriched at sensory synapses in the thalamus and brain stem^{14,15}.” The term ‘fast’ is applied to GluA3 (and GluA4) in the context of gating, but the cited papers do not directly demonstrate slow and fast gating of these receptor groups.

Line 296. The statistical tests in Fig 5d have not been designed to support statements about GluA3-R vs -G.

Line 350. the term “...third AMPAR subtype ...” appears dogmatic and needs to be better qualified (i.e. based on the author’s structural studies).

Fig 2b. The cross-linking experiments lack a DTT control, which is essential for validating the specificity of the observed effects.

Fig 2c/d/methods. The methods section needs to provide a clear explanation of how the Q185glyco modification was achieved.

Fig 3d. It is unclear why H234 not G235 is labelled in the figure given mutational analysis was performed on G235. Also G235C is an odd choice of mutation given the potential reactivity of the thiol group. Presumably this was a relic of a cross-linking experiment that ultimately didn’t make the paper?

Fig 3f. AMPAR recovery has a lag before its exponential phase (e.g. Robert & Howe, 2003). By constraining an exponential fit to the end of the conditioning pulse, the approximation of the initial (and fastest changing) phase is poor – which is compounded by only recording a pulse every 10 ms. There seems little doubt that the G235C mutant recovers more quickly than the wild-type, but I would not be confident that the experiment as performed should be able to detect a <10% effect-size as reported for some of the other mutants in ED Table 2.

Extended data Fig 5a legend. For easier cross-referencing with databases such as Clinvar, the legend should note that the mouse sequence minus signal peptide corresponds to the full human sequence minus 28 – it took some time to unravel the numbering.

Referee #2

(Remarks to the Author)

AMPA glutamate receptors (AMPA) mediate the majority of fast excitatory synaptic transmission in the brain and are essential for synaptic plasticity, learning and memory. Among AMPAR subunits, GluA2 has been extensively studied and multiple GluA2 structures have been determined, however, GluA3, a subtype playing crucial roles in synaptic signaling, neuronal excitability, and disease pathology, is less understood. The GluA3 is interesting because it is an X-linked receptor found in both cortical and subcortical circuits, where it forms fast-conducting CP-AMPA receptors enriched in sensory processing regions such as the auditory brainstem. GluA3 shows different functional properties in comparison to GluA1 and GluA2.

This study by Pokharna et al. represents the first high-resolution structural and functional characterization of homomeric GluA3 in complex with TARP in apo, open and desensitized state. The data reveal different architecture and subunit interfaces than previously published AMPAR structures. They observed highly dynamic NTD layers in different functional states as previous studies. Overall, the structures are in high quality and provide valuable contribution to the Glutamate receptor field. However, there are some critical issues, such as physiological relevance of homomeric GluA3 and lack of discussion and comparison with native GluA3-containing heteromeric structures, that should be addressed.

1. All the structural comparisons in this manuscript are made between homomeric GluA2 and GluA3. However, native GluA3 subunit-containing AMPAR structures has been published in Yu et al, Nature, 2021. The authors should not ignore these structures, and should include them in the structural comparison and discussion. In fact, it is more meaningful to compare with GluA3-containing heteromer than with homomeric GluA2.
2. What is the physiological relevance and fraction of homomeric GluA3 in the brain? Are there specific brain regions or developmental stages where homomeric GluA3 is enriched? This should be discussed in the manuscript.
3. Previous structural studies of GluA2-TARP complexes show similar NTD layer expansion and upright conformation, a discussion of these similarities and differences would strengthen the manuscript.
4. The authors found a novel NTD-LBD interface in homomeric GluA3 by comparison with GluA2 structure. However, such an interface comparison should be made among all the available AMPAR/TARP structures.
5. It is vague about how they captured the open state and how they define the desensitized state of GluA3. A pore profile should be shown among different functional states.
6. There is no discussion about TARP-GluA3 interactions and if they are different than other AMPAR/TARP interactions.
7. What is the ligand shown in Figure 1?
8. The study shows that disrupting the Arg163 stacking interaction increases surface expression and promotes heteromer formation. What is the physiological implication of this finding?
9. The manuscript describes an unusual ‘flat’ organization of the NTD dimers, but this feature is not well illustrated in the figures. A structural panel highlighting this arrangement compared to other AMPAR subtypes would be helpful. Again, the discussion should not be only restricted to the homomeric GluA2.
10. Additional clarification of how the structural and functional properties that they observed correlate with GluA3’s role in rapid synaptic transmission is needed.
11. In the open state data, great heterogeneity is observed in NTD and LBD layers compared to the apo states, with fewer classes having NTDs interfacing with the LBD. What is the activation mechanism? What does the high dynamic of NTD and LBD mean for GluA3 activation?

Referee #3

(Remarks to the Author)

This paper reports new structures of the GluA3 homomer, for which there has been growing interest. The work reports key differences compared to other subtype structures and therefore will be of interest to researchers in the field, though the case

for significance beyond that is not so well made. The paper is clearly written. In the main I focus on the simulation aspect of the manuscript, which is generally fine, but I have the following points

1. Please say what the torsion angle being measured actually is in the main text. Also, the text in the methods around 598 is not sufficient to repeat the simulations. Please define exactly the Ca atoms in the upper and lower lobes (or even better provide the input files). The projection information following this is also too vague. Please provide complete scripts to reproduce this analysis. The authors say its deposited on MDDB, but there is no reference and MDDB is still in the pilot phase. It is also not clear where the analysis scripts are? A zenodo repository would guarantee a doi at the time of publication.
2. Authors say there are 3 runs in Fig. 4b, but its almost impossible to see that (looks like only 2 runs). Please chose better colors!
3. What is the mechanism behind R163I giving dimer displacement? Its not immediately obvious why Ile would necessary be less stable. Is it lack of counter-charge to E180? And if so, is E180 conserved in GluA2? The authors say that E180A gives rearrangements which are presumably different from R163I, but again why?
4. In figure 4C – the energy differences on this map are vast- some 160 kj mol (40 kcal/mol), but the wells also appear very narrow. Unfortunately, it is therefore not easy to see what kind of energy barriers exists between wells 1,2,3 and 4. It would be much more useful to see that zoomed in or re-coloured. This is essential as the authors say on line 257 that these results “emphasise the potential for the GluA3 NTD dimer to populate a wide range of states” but that would not be true if the barriers between states are so high. Given they have the data, the authors should report barrier heights. If the barrier heights are more than a few kT then its hard to see how the dimer would really populate a wide range of states and the above claim would need rephrasing.
5. Line 341 – what are CI and CP receptors?
6. Line 349 – the authors suggest that the R163 site could be targeted for drug-design but there is no indication of where there is a big enough pocket here to suggest that is anywhere near likely.

Version 2:

Reviewer comments:

Referee #1

(Remarks to the Author)

The authors have carefully considered the comments and provided substantial and important new data to support and extend their original study. In terms of my central criticisms:

- 1) Concerns and confusion about “Ultrafast”. The authors have clarified their meaning and provided impressive and informative new electrophysiological data. In terms of the 129/458 disulphide link, one alternative explanation for the altered kinetics observed could be a lower affinity of these receptors which would accelerate recovery kinetics due to fast unbinding of glutamate, and, if the peak current was ‘underestimated’ due to a slow risetime, could lead to large steady-state currents. No need for more experiments, but just confirmation that the risetimes of GluA3 and 129/458 are equivalent would confirm that the altered gating observed did not have an alternative explanation.
 - 2) Mechanisms surrounding R/G 439. Cryo-EM data is now provided for GluA3-R, and some more electrophysiological data demonstrating the effects of this site on heteromerization. There is no doubt that this site is important for the trafficking (or lack of) of GluA3 as homomers, but there is still no mechanism here – I do not agree that a shift of Taudes of 5.4 to 6.8 ms for the R/G forms is informative in this regard. That said, the visualization of aggregated particles is an interesting observation, so regardless of exactly why the R-form behaves differently, there is at least this physical evidence to add to existing knowledge. It might be worth a brief comment concerning GluA3 homomers in the brain – do the authors think that GluA3 homomers ever get to the surface, or are all CP-GluA3-containing AMPARs, heteromers with GluA1 or GluA4?
 - 3) Significance of desensitized LBD-NTD interface. The authors provide extra data in support of a functional role for this interface.
 - 4) Analysis of disease associated mutations. More data has now been added including for the specific variants mentioned in the original review. The adaptations to Extended Figure 5 are appreciated to help map disease variants from the human form to the signal peptide-lacking structural annotations.
 - 5) Reciprocal changes to GluA2. Again, some more useful experiments have been added.
- All minor points were satisfactorily addressed, even though the lack of effect of DTT on 129/458 is surprising – it doesn’t appear like the site of the cysteine bridge should be inaccessible to the reducing agent.

Referee #2

(Remarks to the Author)

The authors have adequately addressed my concerns and markedly improved the manuscript in the revised version. AMPA

homomeric receptors can function as independent units, and the high-resolution structures of fast-conducting CP-AMPA receptors composed of GluA3 subunits reveal unique features associated with calcium permeability and physiological function. I believe this work is of high quality and represents a valuable contribution to the iGluR field.

Response to reviewers 2024-12-26374A; Pokharna et al.

Referee #1:

In this study, Pokharna and colleagues present high-resolution cryo-electron microscopy structures of GluA3 AMPA receptors captured in closed, open, and desensitized conformational states. Their structural analysis reveals distinctive features of GluA3, particularly in the interfaces between N-terminal domains (NTD-NTD) and between the N-terminal and ligand-binding domains (NTD-LBD), which differentiate it from both homomeric GluA1 and GluA2 receptors, as well as from heteromeric GluA1/2 and GluA2/3 assemblies. Using targeted mutagenesis and electrophysiological recordings, the authors validate the functional significance of these structural states. Furthermore, they investigate how the GluA3 N-terminal domain influences both surface expression and synaptic targeting of AMPA receptors in neurons.

The findings are novel, and the technical quality of the data is exceptional, as expected from this lab. However, I'm unsure if the structural findings really allow as comprehensive a dissection of GluA3 physiology as the authors suggest. I have several criticisms:

1. First, the authors' characterization of GluA3 as "ultrafast" in the title and abstract lacks precise definition. The specific kinetic parameters that constitute this designation - presumably activation, deactivation, desensitization, or resensitization - are not clearly specified. Therefore, although the paper seems to promise structural details that explain unusually rapid kinetics of GluA3 (compared to e.g. the equivalent flip/flop or R/G editing forms of GluA2?), it neither presents evidence for unique kinetic behavior nor provides a structural argument for how it might come to pass.

We have addressed this point with additional experiments (Fig. 2b-d), and have revised both title and abstract.

With regard to 'ultrafast', our premise was that both GluA3 and GluA4 are capable of faithfully transmitting high-frequency action potential trains at sensory synapses. This is well established for auditory nerve synapses in the brain stem and lemniscal synapse in the thalamus, where these calcium permeable (CP) AMPAR types dominate, and where transmission frequencies of several hundred Hertz have been reported (eg., Antunes et al., J. Neurosci. 2020; ; reviewed in Trussell, Annu Rev Physiol., 1999;). Enrichment of both GluA3 and GluA4 at these fast synapses was demonstrated by the Wenthold lab (Wang et al., J Neurosci., 1998; Petralia et al. Hear Res 2000). Both synapses signal through (GluA2-lacking) CP AMPARs exhibiting fast recovery from desensitization i.e. they lack GluA1, whose slow recovery kinetics would lead to short-term depression.

Our aim was to explore the currently poorly understood 'fast' CP AMPARs - fast relative to the 'slow' CP GluA1 subtype. We investigated GluA3, as it is ubiquitously expressed in the brain at functionally distinct synapses, and is of great therapeutic interest (e.g. Singh et al., Nature 2022). The kinetic difference between slow and fast CP AMPARs is readily seen in trains of glutamate application to HEK293 cell patches, mimicking high-frequency signal transmission (**Rev Fig 1**). The depression

seen with GluA1 will largely result from its slow desensitisation recovery, which is particularly pronounced at higher frequencies (**Rev Fig 1**). We included the comparison between GluA1 and GluA3 in the revised paper (in Fig. 2b).

[This has been redacted.]

b

[This has been redacted.]

Rev Fig. 1. representative current traces from outside-out HEK293-cell patches elicited with 10 mM glutamate, 1 ms pulses at 5, 10, 20, 50 and 100 Hz frequencies, for GluA1 /TARP-γ2 (blue), GluA3-R/ TARP-γ2 (green) and GluA4/ TARP-γ2 (yellow). Currents are normalized to the first response in the application. Right panel shows zoomed in 100 Hz application. b, plot shows current amplitudes of the first five responses evoked with 100 Hz 10 mM glutamate application for GluA1 /TARP-γ2 (blue), GluA3-R/ TARP-γ2 (green) and GluA4/ TARP-γ2 (yellow). Currents are normalized to the first response in the application (P1 =1) and pooled together. Each dot presents the mean and the whiskers are SD.

Our structural argument is that the unique coupling of the GluA3 NTD and LBD is linked to faster kinetics. We had previously associated NTD dynamics structurally to kinetics, where greater NTD mobility is associated with slower kinetics in recent work with GluA1 (Zhang et al., Nature 2023), and GluA2 (Ivica et al., NSMB 2024). The current paper describes that the GluA3 NTD tier is less dynamic than GluA1 but more mobile than the GluA2 NTD, which can be schematised as follows (**Rev Fig. 2**):

Rev Fig. 2. Schematic of three AMPAR subtypes exhibiting a different arrangement of their NTD tier. GluA1 has highly flexible NTD dimers; these are constrained by the B/D interface in GluA2 and by the NTD-LBD interface in GluA3 (GluA2 and GluA3 NTD interfaces are denoted by black boxes).

In the paper, we had shown that mutating GluA3 NTD-LBD interface contacts leads to altered kinetics (Fig 2 and Table 2, in the original submission). We have now further elaborated this point in the revised paper, showing that stabilising the GluA3-specific NTD-LBD interface with the 129/458CC disulfide bridge further facilitates GluA3 responses to fast (100 Hz) input trains, relative to GluA3 wt. This facilitation requires the presence of both cysteines (the 129/458 double Cys mutant), and was not apparent with either of the two individual Cys mutations (**Rev Fig. 3**; rightmost panel).

Rev Fig. 3. **Coupling of GluA3 NTD-LBD with K129C/E458C facilitates responses to fast glutamate application.** Plot shows current amplitudes of the first five responses evoked with 100 Hz 10 mM glutamate application for GluA3-G/ TARP- γ 2 (green, 11 patches), GluA3-G_{K129C}/ TARP- γ 2 (yellow, 4 patches), GluA3-G_{E458C}/ TARP- γ 2 (blue, 3 patches) and GluA3-G_{K129C/E458C}/ TARP- γ 2 (orange, 6 patches). Currents are normalized to the first response in the application (P1 = 1) and pooled together. Each dot presents the mean and the whiskers are SD.

Hence, stabilising the GluA3-specific NTD-LBD domain coupling is facilitating, and is opposite to the interface disrupting 185 glyco-wedge mutant, which blunts GluA3 train responses (Fig. 2d in the revised paper).

Of note, there is a precedent for NTD proximity to the LBD affecting gating kinetics. We had shown earlier that shortening the GluA2 NTD-LBD linker, as is commonly performed in structural studies, leads to hastened desensitization recovery (Cais et al., Cell Rep. 2014).

To avoid distraction from the main message of our paper, we have now revised the title to:

'Architecture, dynamics and biogenesis of GluA3-type AMPA glutamate receptors'
And we also revised the abstract.

2. Second, regarding GluA3 trafficking, while the role of arginine at the R/G 439 'trafficking checkpoint' is long-established, the current study does not substantially advance our mechanistic understanding. Although this residue is positioned near the LBD/NTD interface identified in desensitized receptors, the authors do not present structural modelling or a clear hypothesis explaining how this site regulates trafficking. Given that the G/R residue minimally affects mature receptor kinetics, the dramatic impact on surface expression likely involves mechanisms not revealed by the data.

The GluA3 G/R439 trafficking checkpoint in the LBD was described in 2 papers by Coleman and Keinänen (JBC 285, 2010 and JBC 291, 2016), showing that GluA3-R is trafficking-incompetent and exhibits misfolding/aggregation in heterologous cells. To better understand the mechanism of ER retention by Arg439, and to complement our structural analysis of GluA3, we now determined an apo state structure of GluA3-R (the mammalian variety), also associated with TARP- γ 2, and contrast this to GluA3-G (the variety expressed in non-mammalian vertebrates) (outlined in Extended Data Fig.6 of the revised paper).

Apo GluA3-R markedly departs from GluA3-G. This is seen biochemically (GluA3-R is prone to aggregation; Extended Fig. 6b), and is immediately obvious in 3D classifications, where the R-form exhibits profound structural heterogeneity: GluA3-R features LBD tier instability, reflected in separation of one of the two dimers (within a receptor tetramer) across the majority of 3D classes. By contrast, apo-state GluA3-G is characterised by intact (resting-state) LBD dimers (Extended Fig. 6d versus e), as would be expected for an AMPAR determined in an apo state. This 'LBD dimer splitting' is a structural hallmark of AMPAR desensitization, and may contribute to the faster desensitization entry of GluA3-R, relative to GluA3-G (Extended Data Table 2). It may also explain the trafficking deficit of GluA3-R as gating in the ER has been linked to ER export (Grunwald and Kaplan, Neuropharmacology 2003; Mah et al., J Neurosci. 2005; Valluru et al., JBC 2005; Penn et al., EMBOJ.2006). Related to this, disulphide mutants trapping the novel desensitized state NTD-LBD interface lead to very small currents suggestive of poor trafficking (discussed in the next point).

We further confirm, using FACS analysis of transfected HEK293 cells, that GluA3-R is mostly intracellular even in the presence of TARP- γ 2, an established ER export factor (Extended Data Fig.6a). Hence, a large fraction of structurally resolved GluA3-R will originate from the ER. Despite the substantial flexibility in the LBD tier, NTD interfacing with the LBD is also apparent in GluA3-R: in a refined class exhibiting intact LBD dimers, the NTD-LBD arrangement matches the one of GluA3-G (Extended Data Fig.6g). This suggests that the NTD-LBD interface is a hallmark of all vertebrate GluA3 receptors. We also attempted a structure of GluA3-R desensitized state. The poor behaviour of the protein prevented a detailed structural analysis; however, the resolvable 3D classes featured the desensitized NTD-LBD interface similar to the one seen with GluA3-G (association of LBD helix J with the NTD). This suggests that both vertebrate GluA3 varieties undergo comparable NTD-LBD arrangements in both apo and desensitized states.

We propose that an unstable LBD tier in GluA3-R shifts subunit assembly towards heteromerization. We use patch clamp recordings in HEK293 cells, to document this, and co-expressed GluA3-R, -G or GluA1 with GluA2, using various plasmid ratios. Even at an excess of GluA3-R over GluA2 (a 4:1 ratio of GluA3-R:GluA2), heteromers greatly prevailed. By contrast, at this ratio we detected a substantial fraction of both GluA1 and GluA3-G homomers (Extended Data Fig. 6h,i). Lastly, we had shown that the difference in assembly preference between GluA3-R and GluA3-G is also apparent in neurons (as described in Fig. 5 of the original submission). To extend this neuronal dataset we provide further evidence that the R163I mutation alters heteromeric assembly. We show that the increased synaptic response of GluA3-RR_{163I} (in wt slices) is not seen when the mutation is introduced into GluA3-G (GluA3-GR_{163I}). This is readily explained by the fact that GluA3-G preferentially forms homomers and supports the notion that the R163I mutation alters heteromeric assembly of GluA3-R, which underlies the boost of synaptic transmission (Extended Data Fig. 11e and f).

3. Third, the functional significance of the desensitized GluA3 LBD-NTD interface remains questionable. While the LBD half of the interface incorporates elements of the flip/flop cassette near the R/G editing site, it doesn't seem like these contacts will be that important in mediating the functional effects of post-transcriptional processing given that GluA2, which lacks this interface, shows similar effects of the same modifications.

As the reviewer points out the difficulty here is that the LBD half also contributes to the well-studied D1 LBD dimer interface, which rearranges during gating and is a crucial determinant of gating kinetics across AMPARs (e.g. Sun et al., Nature 2002; Horning & Mayer, Neuron 2004). We had attempted to introduce disulfide bridges to lock the interface between NTD and LBD helix J (including G235C+T748C and K212C+L752C), but all LBD+NTD double cysteine mutants resulted in very small currents, preventing further functional analysis. Hence, we were left with mutants on the NTD half, and these yielded significant kinetic differences at the Gly235 interface contact point, which are now documented for both GluA3-G and GluA3-R. Moreover, these effects are *specific for GluA3*, and are negligible when introduced at the equivalent site in GluA2 (in response to point 5 of this reviewer) (**Rev Fig 4**).

Rev Fig. 4. Recovery from desensitisation from the pooled data: left panel A3-G/TARP- γ 2 ($n = 9$ cells) and A3-G_{G235C}/TARP- γ 2 ($n = 7$ cells), middle panel A3-R/TARP- γ 2 ($n = 9$ cells) and A3-R_{G235C}/TARP- γ 2 ($n = 10$ cells), fitted with two-component Hodgkin-Huxley equation. The slope for the fast component was fixed at 4, while the slope for the slower component was fixed at 1. Right panel GluA2/TARP- γ 2 (9 cells) fitted with two-component Hodgkin-Huxley equation. The slope for the fast component was fixed at 3, while the slope for the slower component was fixed at 2. GluA2_{G232C}/TARP- γ 2 (11 cells) fitted with Hodgkin-Huxley equation with the slope fixed to 2.

Structurally, our key observation of desensitized GluA3 was the common appearance of a unique NTD-LBD conformation, which has not been observed in other AMPAR structure before. This conformation greatly dominated in GluA3-G desensitized classes and it is similarly adopted by desensitized GluA3-R (recognisable albeit its poor resolution). The dominance of this conformation throughout 3D classes implies the existence of a preferred, low-energy desensitized state. As discussed in the paper, we hypothesise that GluA3, which lacks the stabilising GluA2 NTD B/D interface (Rev Fig 2) has evolved NTD-LBD interfaces for structural stability, enabling rapid gating kinetics. This is supported by *preservation of the NTD-LBD domain swap* in desensitized GluA3. This domain swap is a characteristic feature of iGluRs that is lost in GluA1 (Zhang et al., Nature 2023).

4. Fourth, the analysis of disease-associated mutations, while valuable, could be more comprehensive. The authors chose to examine artificial mutations (e.g., R188A, R473A) rather than patient-specific variants (R188Q, R473T). The significance given to certain structural proximities (such as A237T near to Gly235) may be coincidental given the number of NTD variants and their broad distribution. Additionally, the finding of Peng et al that E759G causes complete loss of function is more likely due to compromised agonist binding or protein folding than disruption of a desensitized LBD-NTD interface.

We have now repeated these recordings with the actual disease mutations locating to the apo-state interface, R188Q and R473T. The latter exhibited kinetic differences similar to the alanine mutations we had presented previously. Although still significantly different to the wt, fitting the data with a double Hodgkin-Huxley equation reduced the kinetic effect of R473T (Table 2 in the revised paper). Of note, both R473A and R473T exhibited significantly reduced peak amplitudes, suggestive of a trafficking phenotype (as shown for R473A in the FACS analysis below, **Rev Fig. 5**). An even more pronounced trafficking impediment was apparent for another disease mutation, R176H, locating to the GluA3 NTD dimer interface (**Rev Fig 5**); the resulting low amplitude responses precluded a kinetic analysis of this mutant. As the reviewer points out, disease mutations are broadly distributed across the NTD surface, with largely unknown function (Rinaldi et al., Brain 2024). This may reflect NTD interaction with synaptic protein partners critical for receptor signalling or

anchoring.

Rev Fig 5. Summary graph (mean \pm SD) for flow cytometry data of the surface expression of GluA3-R, GluA3-R_{R176H} and GluA3-R_{R473A}. Geometric mean fluorescence of each sample was normalized to the average value for GluA3-R on that day (indicated by green dashed line). Effect of mutation on the surface expression was

determined with ordinary ANOVA ($F_{(2, 11)} = 50.37$; **** $p < 0.0001$) followed by Dunnet's multiple comparison test, p values are indicated.

5. A limitation of the mutagenesis studies is the absence of reciprocal experiments in GluA2, (with the exception of domain transplantation). Testing equivalent mutations in GluA2 (e.g., at positions 163/164/180, the cys129/458 pair, Gly235 and Arg183) would have strengthened the authors' conclusions that these are uniquely important to the novel structural features of GluA3. i.e. if the functional/synaptic results of changes to GluA3 are solely due to modifying structural parameters, the equivalent changes in GluA2 should be functionally silent/have a different impact.

We have addressed this point by analysing equivalent mutations in GluA2. These were expressed in CA1 pyramidal neurons, where 'structural' GluA3 interface mutants exhibited a clear phenotype (Fig. 5 of the original submission). None of the GluA2 NTD-LBD interface mutants, neither L180R (R183A in GluA3) nor the 182 glyco mutation (185 in GluA3), affected surface trafficking of GluA2 (**Rev Fig 6a**) (shown in Extended Data Fig. 11c of the revised paper). The GluA2 NTD dimer interface mutant I157R (R163I in GluA3) had the *opposite* impact on surface trafficking (**Rev Fig 6b**). Thus an arginine at this position *elevated* surface expression in GluA2, but *reduced* trafficking of GluA3. However, contrasting with GluA3, this GluA2 mutation did not impact excitatory postsynaptic current (EPSC) amplitudes (**Rev Fig. 6c**). Taken together, reciprocal experiments with GluA2 were either of no effect, or worked differently when assessed in neurons.

Rev Fig 6. a, Rectification Index from somatic outside-out patch of CA1 hippocampal neurons expressing GluA2 mutants L180R, I157R and K182_glyco. Effect of mutations on rectification of surface receptors was determined with One-way ANOVA ($F_{5,44} = 1.88$) followed by Dunnet's multiple comparison test. * $p=0.0229$ b, Surface rectification of NTD dimer interface mutation I157R in GluA2 and R163I in GluA3. ($F_{4,49}=2.26$, * $p=0.048$ **** $p<0.0001$) c, EPSC amplitude of CA1 hippocampal neurons expressing NTD dimer interface mutants or wt GluA2 or GluA3-R, normalised to a neighbouring untransfected neuron. Effect determined by One-way ANOVA followed by Dunnet's multiple comparison test **** $p<0.0001$.

As pointed out above (Rev Fig 4), we also assessed the GluA2 G232C mutant locating to the desensitized NTD-LBD interface (equivalent to GluA3 G235C), which, contrary to GluA3, had no kinetic phenotype.

Minor Comments/Technical Points

Abstract. the phrase "...the ultrafast conducting GluA3 and GluA4 receptors, prevalent in interneurons and the brainstem" is used. The citation used to support

this statement seems inappropriate as it does not mention ultrafast or 'ultrafast conductance', nor does it explicitly refer to the conductance of GluA3 or GluA4 or to brainstem neurons.

Thank you for pointing this out, refs 14 and 23 (from the original submission) should have been cited at this point. As mentioned above, we have reworded both the title and abstract, and have cited Antunes et al., J. Neurosci., 2020 and Trussell, Annu Rev Physiol. 1999 at the respective position in the introduction.

Line 64. "CP-AMPA subunits segregate into the slowly gating GluA1, and the fast GluA3 and GluA4 receptors, a subgroup selectively enriched at sensory synapses in the thalamus and brain stem^{14,15}." The term 'fast' is applied to GluA3 (and GluA4) in the context of gating, but the cited papers do not directly demonstrate slow and fast gating of these receptor groups.

We have now cited the main glutamate receptor review at this point, which summarises kinetic properties of all AMPAR subunits (Hansen et al., Pharmacol Rev 2021).

Line 296. The statistical tests in Fig 5d have not been designed to support statements about GluA3-R vs -G.

Thank you for pointing this out. We have now rectified this and updated Figure 5d in the revision to show the Dunnett's test comparing to the mean of A3-R. The text now reads as follows: 'While GluA2(Q) readily expresses at the cell surface ($RI = 0.21 \pm 0.02$, $n = 13$; $RI_{untransfected} = 0.66 \pm 0.02$, $n = 14$)^{9,10}, this was not the case for GluA3-R ($RI = 0.59 \pm 0.01$, $n = 19$), but was apparent with GluA3-G ($RI = 0.38 \pm 0.02$, $n = 11$) (Fig. 5d), consistent with data from HEK293 cells (Extended Data Fig. 6a)' Details of all statistical tests are also provided in the Source data file.

Line 350. the term "...third AMPAR subtype ..." appears dogmatic and needs to be better qualified (i.e. based on the author's structural studies).

We reword this to clarify that we describe a *structurally* third AMPAR subtype at the level of the sequence-diverse NTD, which forms an interface with the (A/C) LBDs and is laterally expanded, relative to GluA1 and GluA2 receptors (depicted in Fig. Rev 2). We include this schematic in Fig. 5g.

The sentence now reads:

'The GluA3 homomer adopts an architecture not observed with other AMPARs. Its unique NTD/LBD-coupled organisation substantially differs from the detached NTD tier of GluA2 receptors, which acts as an efficient synaptic anchor^{9,10,59}, and from the highly mobile GluA1 NTDs (Fig. 5g)¹⁷.'

Fig 2b. The cross-linking experiments lack a DTT control, which is essential for validating the specificity of the observed effects.

We fully agree and we had tried DTT applications (5 mM DTT for > 5 mins) but surprisingly were unable to reverse the slowing of desensitization entry and increase of the equilibrium current. However, as we pointed out in the paper, the kinetic

changes apparent in the Cys bridge (double) mutant were eliminated when testing the individual mutations, implying the existence of a bridge. We further assessed this in the train responses, which selectively facilitated only in the double Cys mutant, as documented in Rev Fig 3.

Fig 2c/d/methods. The methods section needs to provide a clear explanation of how the Q185glyco modification was achieved.

We have clarified this in the Methods, thank you. In brief, N-glycans are introduced via the Asn-X-Ser/Thr sequon, which will lead to glycosylation of the Asn residue in the endoplasmic reticulum.

Fig 3d. It is unclear why H234 not G235 is labelled in the figure given mutational analysis was performed on G235. Also G235C is an odd choice of mutation given the potential reactivity of the thiol group. Presumably this was a relic of a cross-linking experiment that ultimately didn't make the paper?

We have now labelled G235 in Figure 3e. Re the choice of cysteine - yes, it was part of a bridge, as described in response to point 3, but a similar effect was apparent with the Ala mutant (G235A).

Fig 3f. AMPAR recovery has a lag before its exponential phase (e.g. Robert & Howe, 2003). By constraining an exponential fit to the end of the conditioning pulse, the approximation of the initial (and fastest changing) phase is poor – which is compounded by only recording a pulse every 10 ms. There seems little doubt that the G235C mutant recovers more quickly than the wild-type, but I would not be confident that the experiment as performed should be able to detect a <10% effect-size as reported for some of the other mutants in ED Table 2.

Thank you for this comment. We have now adjusted our protocol and data analysis, and have collected new data for all mutants (Table 2). In brief, we used two sampling rates such that the first 40 ms of recovery was collected in 2 ms intervals, while in the 50-240 ms time span, we increased the interpulse intervals by 10 ms (**Rev Fig 7a**). Since a one exponential fit did not fit the data well, we now use a sum of two Hodgkin-Huxley terms (Salazar et al., Biophys J. 2020), where k_1 and k_2 are the rates of recovery and m_1 and m_2 are the slopes. We achieved a good fit by fixing the values of the slopes m_1 and m_2 to 4 and 1. The y_{max} was constrained to 1. The weighted tau of recovery was calculated as $((t_1 * a_1) + t_2 * (y_{max} - a_1 - y_0)) / (y_{max} - y_0)$. This is documented for the G235C mutant below (**Rev Fig 7b**). We also present the revised recovery protocol in Ext. Data Fig. 9d of the revised paper.

Rev Fig. 7. **Protocol for recovery from desensitisation.** Example trace of paired pulse current elicited from whole cells expressing *GluA3-G/TARPy2* at -60 mV. The dashed black line represents fit to the peak ratios with two component Hodgkin-Huxley equation with a time constant (τ_w) of 20.4 ms. b) Recovery from desensitisation from the pooled data for *A3-G/TARPy2* ($n=9$ cells) and *A3-GG235C/TARPy2* ($n=7$ cells). In the left panel amplitudes are fitted with one exponential fit which clearly does not fit well to data. In the right panel the same data are fitted with two component Hodgkin-Huxley equation. The slope for the fast component was fixed at 4, while the slope for the slower component was fixed at 1.

Extended data Fig 5a legend. For easier cross-referencing with databases such as Clinvar, the legend should note that the mouse sequence minus signal peptide corresponds to the full human sequence minus 28 – it took some time to unravel the numbering.

We now include a table in Extended Data Fig 5b, listing the mutants in question both with signal peptide (matching Clinvar) and without signal peptide matching the structure and PDB file.

Referee #2:

AMPA glutamate receptors (AMPA) mediate the majority of fast excitatory synaptic transmission in the brain and are essential for synaptic plasticity, learning and memory. Among AMPAR subunits, GluA2 has been extensively studied and multiple GluA2 structures have been determined, however, GluA3, a subtype playing crucial roles in synaptic signaling, neuronal excitability, and disease pathology, is less understood. The GluA3 is interesting because it is an X-linked receptor found in both cortical and subcortical circuits, where it forms fast-conducting CP-AMPA receptors enriched in sensory processing regions such as the auditory brainstem. GluA3 shows different functional properties in comparison to GluA1 and GluA2.

This study by Pokharna et al. represents the first high-resolution structural and functional characterization of homomeric GluA3 in complex with TARP in apo, open

and desensitized state. The data reveal different architecture and subunit interfaces than previously published AMPAR structures. They observed highly dynamic NTD layers in different functional states as previous studies. Overall, the structures are in high quality and provide valuable contribution to the Glutamate receptor field. However, there are some critical issues, such as physiological relevance of homomeric GluA3 and lack of discussion and comparison with native GluA3-containing heteromeric structures, that should be addressed.

1. All the structural comparisons in this manuscript are made between homomeric GluA2 and GluA3. However, native GluA3 subunit-containing AMPAR structures has been published in Yu et al, Nature, 2021. The authors should not ignore these structures, and should include them in the structural comparison and discussion. In fact, it is more meaningful to compare with GluA3-containing heteromer than with homomeric GluA2.

Thank you for raising this. We hadn't discussed the native GluA2/3 heteromer as its conformation *closely resembles the GluA2 homomer* but *differs from the GluA3 homomer*. Hence the well-studied, and better resolved GluA2, also associated with four TARP- γ 2 subunits (thus more closely resembling our GluA3/ TARP- γ 2 structure) was used for comparison. The similarity between the native GluA2/3 heteromer and GluA2 homomer is due to the NTD B/D interface, formed by the inner GluA2 subunits in both receptor types (see Rev Fig. 2). This interface stabilises GluA2-containing receptors, rendering them conformationally compact and distinct from the GluA1 and GluA3 homomers. This NTD interface also influences their kinetics (Zhang et al., Nature 2023; Ivica et al., NSMB 2024), and contributes to the spacing between the NTD and LBD tiers that is absent in GluA3. Nevertheless, we agree with the reviewer that the GluA2/3 receptor structure should feature. We have included a comparison now, both in the text (pages 5+6) and as a figure (Extended Data fig. 4a).

2. What is the physiological relevance and fraction of homomeric GluA3 in the brain? Are there specific brain regions or developmental stages where homomeric GluA3 is enriched? This should be discussed in the manuscript.

As AMPARs can assemble into functional homomers (which is not the case for the related NMDA receptors), it is likely that homomers will exist in various brain regions and synapses that are dominated by CP AMPARs. The currently best accepted example is the transient appearance of GluA1 homomers in CA1 pyramidal neurons (reviewed in Purkey et al., Front. Synaptic Neurosci., 2020). Homomer formation will depend on the expression levels of partner subunits in a given cell, and rapid advances in RNA patch-seq data and single-cell proteomics will contribute to a better understanding of this question.

In the case of GluA3, the best evidence for the occurrence of homomers is the endbulb of Held (bushy cell) synapse in the ventral cochlear nucleus. As pointed out above it is well established that synapses in this brainstem nucleus are dominated by CP GluA3 and GluA4 AMPAR subtypes (and are mostly devoid of GluA1 and GluA2). High-resolution freeze-fracture imaging studies (Rubio et al., Brain Struct Funct 2017) revealed a central region in the endbulb synapse that is enriched in GluA3 and is devoid of GluA4, and is currently the best direct evidence for GluA3 homomers (Ref 18 in the revised paper). Other fast transmitting sensory synapses are also enriched in GluA3 and GluA4, and there is circumstantial genetic evidence

for GluA3 homomers operating at the lemniscal synapse in the thalamus (Wang et al., J. Neurosci., 2011; [Ref 16 in the revised paper]). For these reasons, the widely expressed GluA3 subunit likely also exists as both CP homomer at sensory synapses, and as CI heteromer throughout cortical synapses.

3. Previous structural studies of GluA2-TARP complexes show similar NTD layer expansion and upright conformation, a discussion of these similarities and differences would strengthen the manuscript.

We are not aware of any GluA2 structures exhibiting an NTD layer expansion like the one we describe for GluA3 (outlined in Fig. 1 of the original submission). Both GluA2 homomers, and GluA2-containing heteromers, have highly comparable, and upright, NTD tiers, dominated by the aforementioned NTD B/D interface (Rev Fig. 2). *Splayed* NTD dimers exist in GluA1 and can be induced in GluA2 by the F231A mutation or by low pH (Zhang et al., Nature, 2023; Ivica et al., NSMB 2024) but these are again very different to the arrangement we observe with resting-state GluA3.

4. The authors found a novel NTD-LBD interface in homomeric GluA3 by comparison with GluA2 structure. However, such an interface comparison should be made among all the available AMPAR/TARP structures.

This interface is unique to the GluA3 homomer. It is absent in currently available AMPAR structures, both in GluA2-containing receptors and in the GluA1 homomer (where the highly dynamic NTD tier remains poorly resolved). An approximation between NTD and LBD has been seen in recombinantly expressed GluA2 homomers, where the NTD-LBD linker had been shortened (to enable crystallisation), but this is not seen in recombinant wild-type receptors (with unaltered NTD linkers) nor in native AMPARs. Hence, the comparison we had made between GluA2 and GluA3 holds across AMPARs.

5. It is vague about how they captured the open state and how they define the desensitized state of GluA3. A pore profile should be shown among different functional states.

The open state was captured in the presence of the agonist L-glutamate and cyclothiazide (a positive allosteric modulator preventing desensitization), as has been described in previous open-state AMPARs (Chen et al., Cell 2017, Twomey et al, Nature 2017; Zhang et al., Nature 2021 and Nature 2023). The desensitized state was captured in the presence of agonist only. We have clarified this in the methods. As suggested by the reviewer, we now present pore profiles for the open versus the resting states in Extended Data Fig. 8e.

We have also amended the main text on p.9:

'To assess this gating transition in GluA3-G, we captured the receptor in an open state (in the presence of L-glutamate and the desensitization blocker cyclothiazide, as described⁴²⁻⁴⁴).

6. There is no discussion about TARP-GluA3 interactions and if they are different than other AMPAR/TARP interactions.

The LBD-TMD tiers are mostly comparable across current AMPAR structures associated with Type1 TARPs ($\gamma 2$ and $\gamma 3$), including our GluA3 structure. Structural information is limited in the TARP extracellular region, due to the flexible nature of the TARP loops. We had noted an interaction between the TARP- $\gamma 2$ beta 1 loop with the AC GluA3 subunits. These contacts involve LBD helix K, associated with flip/flop splicing. We have included a figure of this interaction in the revised manuscript (Extended Data Fig 8a), and describe it on page 10 of the revised paper.

7. What is the ligand shown in Figure 1?

Based on the electrostatic environment, and on coordination chemistry, we have modelled a chloride ion into this density (consistently observed across gating states). We had described this in line 140 of the original paper and in Extended Data Fig 4d. We have now updated this figure, to better document coordination of this putative chloride ligand with neighboring residues and predicted water densities (in Extended Data Fig 4e).

8. The study shows that disrupting the Arg163 stacking interaction increases surface expression and promotes heteromer formation. What is the physiological implication of this finding?

This is a key question. Based on our current data we surmise that conformational changes in the NTD tier are sensed in the ER, as has been reported for the LBD (Mah et al., J. Neurosci. 2005; Coleman et al., JBD 2006; Greger et al., Neuron 2006; Penn et al., EMBO J. 2008), and will determine ER export. Based on our R163I cryo-EM structure and on our MD simulations, the R163I mutation will alter NTD dimer conformation from 'flat' to more displaced (Fig. 4 of the original paper), and this conformation enables enhanced surface trafficking and delivery into the synapse (Fig 5d and f). Of note, GluA3 NTD dimer conformation is also modulated by endogenous ligands, such as PO₄ (Lee et al., Structure 2019; PDB: phosphate 6FPJ), which may alter interface structure and trafficking.

Regarding heteromer formation, the NTD dimer interface varies amongst AMPAR subtypes in both sequence and dimer affinity (Rossmann et al., 2011, Zhao et al., Elife 2017), and therefore plays a major role in heteromer assembly choice (Herguedas et al., Prog Mol Biol Transl Sci. 2013). GluA3 Arg163, an NTD interface residue unique to GluA3 (and highly conserved across GluA3 orthologs), renders *isolated* NTDs highly dynamic and structurally versatile (Sukumaran et al., EMBO J. 2011; Lee et al., Structure 2019). While in full-length GluA3 it forms an unusual Arg163 stacking interaction across the NTD dimer interface, enabled by association with the LBD. How the R163I mutation (Ile at this site is naturally existing in GluA2) directs the altered NTD interface towards preferred heteromer formation is currently unclear and under investigation.

9. The manuscript describes an unusual 'flat' organization of the NTD dimers, but this feature is not well illustrated in the figures. A structural panel highlighting this arrangement compared to other AMPAR subtypes would be helpful. Again, the discussion should not be only restricted to the homomeric GluA2.

This difference was meant to be documented in the superposition between the GluA2 and GluA3 NTD dimers (in Fig. 1e), where one protomer from each dimer was overlaid to highlight the NTD displacement of GluA2 relative to the non-displaced ('flat') GluA3 dimer. We had also documented this in Extended Data Fig. 4a, which included the GluA1/2 and GluA2/3 NTD heterodimers.

We now include the full-length GluA2/3 receptor heteromer in Extended Data Fig. 4, which exhibits a similar NTD dimer displacement that differs from the full-length GluA3 homomer. We also add additional text into main text Fig 1d, to explain the superposition underlying flat versus displaced NTD dimer, which is explained further in the text on p.6:

'This is evident upon superposition of the GluA3 and GluA2 (PDB: 9B68) NTD dimers using the A protomers, showing a ~30° displacement of GluA2 relative to GluA3 (protomers B; Fig. 1e).'

10. Additional clarification of how the structural and functional properties that they observed correlate with GluA3's role in rapid synaptic transmission is needed.

Please see our response to reviewer 1, point 1, where we expand on this point, for which we generated additional functional data on the GluA3 NTD-LBD interface (Fig. 2a, Extended Data Fig 11).

11. In the open state data, great heterogeneity is observed in NTD and LBD layers compared to the apo states, with fewer classes having NTDs interfacing with the LBD. What is the activation mechanism? What does the high dynamic of NTD and LBD mean for GluA3 activation?

This is another interesting question - what we can say at this stage is that conformational changes triggered by agonist binding to the LBD will lead to a partial rupture of the NTD-LBD interface. What this means for the activation mechanism is unclear. We do not observe differences in L-glutamate efficacy in GluA3-G wt versus the interface cross-linked C129-C458 mutant. As discussed, a complete rearrangement of this interface occurs when triggering larger conformational changes in the LBD tier, during desensitization, where the appearance of LBD monomers is apparent.

Referee #3:

This paper reports new structures of the GluA3 homomer, for which there has been growing interest. The work reports key differences compared to other subtype structures and therefore will be of interest to researchers in the field, though the case for significance beyond that is not so well made. The paper is clearly written. In the main I focus on the simulation aspect of the manuscript, which is generally fine, but I have the following points

1. Please say what the torsion angle being measured actually is in the main text.

This torsion angle is based on the centres of mass of the C α atoms of the upper and lower lobes of the two subunits. For the experimental structures and classical MD simulations, this torsion angle was measured using ProDy with the upper lobe

defined as residues 1 to 116 and 244 to 353 and the lower lobe defined as residues 117 to 243 and 354 to 380. This captures the rotation of the lower lobes relative about an axis passing between the upper lobes. We have clarified this in the revised main text and in the methods section.

Also, the text in the methods around 598 is not sufficient to repeat the simulations. Please define exactly the Ca atoms in the upper and lower lobes (or even better provide the input files). The projection information following this is also too vague. Please provide complete scripts to reproduce this analysis. The authors say its deposited on MDDDB, but there is no reference and MDDDB is still in the pilot phase. It is also not clear where the analysis scripts are? A zenodo repository would guarantee a doi at the time of publication.

We apologise that our metadynamics simulations were difficult to understand and repeat. We have now revised the methods and provide scripts and other input files on Zenodo with links available in the manuscript.

We also apologise that the MDDDB curators had not completed our depositions by the time of submission. This has now been completed and we have included IDs and DOI links for MDDDB and Zenodo entries in the data availability statement.

2. Authors say there are 3 runs in Fig. 4b, but its almost impossible to see that (looks like only 2 runs). Please chose better colors!

Thank you for the suggestion. We have now improved this.

3. What is the mechanism behind R163I giving dimer displacement? Its not immediately obvious why Ile would necessary be less stable. Is it lack of counter-charge to E180? And if so, is E180 conserved in GluA2? The authors say that E180A gives rearrangements which are presumably different from R163I, but again why?

Both R163I and E180A are aimed at destabilising the R163-E180A stack. Ile is too short to form stable contacts in the flat conformation and the remaining E180 pair indeed causes charge repulsion because of the lack of counter-charge. Likewise, E180A leads to charge repulsion of the R163 pair. The system rearranges and finds another stable state, which is a displaced one.

Ile is the corresponding residue in GluA2 and R163I destabilising the flat stacked conformation leads to formation of a stable GluA2-like displaced dimer. E180A also rearranges to a displaced dimer, but it is slightly different due to having different residues making contacts.

E180 is not conserved in GluA2 and is a Q instead. GluA2 Q176 potentially interacts with E149 in the displaced conformation (PDB: 3hsy). GluA3 has a compensatory change of E149 to Q155 that could potentially interact in the displaced conformation of GluA3 (PDB: 3p3w dimer BD, side chain not modelled due to low resolution). The E180-Q155 interaction is also seen for 20ns in one simulation of R163I.

We have adjusted the text to make this clearer by removing E180A.

4. In figure 4C – the energy differences on this map are vast- some 160 kj mol (40 kcal/mol), but the wells also appear very narrow. Unfortunately, it is therefore not

easy to see what kind of energy barriers exists between wells 1,2,3 and 4. It would be much more useful to see that zoomed in or re-coloured. This is essential as the authors say on line 257 that these results “emphasise the potential for the GluA3 NTD dimer to populate a wide range of states” but that would not be true if the barriers between states are so high. Given they have the data, the authors should report barrier heights. If the barrier heights are more than a few kT then its hard to see how the dimer would really populate a wide range of states and the above claim would need rephrasing.

The narrow wells and high barriers in our original metadynamics simulations are indeed on the order of 20 kJ/mol or 8 kT as can be seen in the zoomed view of the landscape below, which we have included in **Rev Fig 8a**.

Using the Arrhenius equation $k = Ae^{-\frac{E_a}{k_B T}}$ and taking A as 1 as there’s no collision rate to factor in, we get $k = e^{-8} = 3.36 \times 10^{-4}$ per second or about 1 transition per hour, which isn’t so low on the timescales of synaptic plasticity and learning. Given that there are about 20-60 receptors in a synapse (MacGillavry et al., 2013; Nair et al., 2013; Tang et al., 2016) and two NTD dimers in a receptor, this means a synapse could have an NTD dimer undergo a conformational transition every minute or so.

We note that these simulations were carried out with very low salt concentrations (10 mM) to demonstrate the intrinsic behaviour of the GluA3 NTD dimer. We expect this barrier to be lower under physiological conditions where phosphate or chloride ions can bind between the lower lobes at R163 and R184, as seen in new simulations that we present in **Rev Fig 8b**.

We have included a short description of these results in the revised text (page 12-13) and new Extended Data Figure 10b.

Rev Fig. 8. Zoomed landscape showing wells better (a) and new simulation results showing a chloride ion stabilising transition states (b). (a) The zoomed landscape shows an energy difference between the black wells (-160 kJ/mol) and the dark purple transition region (-140 kJ/mol; darkest part of box labelled TS) of 20

kJ/mol. (b) Simulations based on a crystal structure occupying the transition region (PDB: 3O21, dimer CD) often show transitions to displaced states (lower trace) as in Lee et al. Structure 2019 and Dutta et al. 2012, but two runs are stabilised at the transition state by chloride ions (orange) binding between the lower lobes (upper two traces and ribbon cartoon). The chloride ions bind the same site as phosphate between R163 and R184.

5. Line 341 – what are CI and CP receptors?

These are the two main types of AMPAR receptors, functionally different depending on the presence of the GluA2 subunit as alluded to briefly. We understand that this may not be so obvious for non-specialist readers and have taken this opportunity to expand and repeat the definition.

Calcium-impermeable (CI) receptors do not permit calcium to pass through the channel pore as a result of two arginine residues coming from endogenous GluA2, which is RNA-edited at the Q/R site.

Calcium-permeable (CP) receptors are those that lack GluA2 or only have a GluA2 that we added with the Q/R site unedited, leaving all four subunits containing glutamine, which permits calcium to permeate.

In the manuscript, these abbreviations are defined at their first occurrence in the abstract and again in the introduction.

6. Line 349 – the authors suggest that the R163 site could be targeted for drug-design but there is no indication of where there is a big enough pocket here to suggest that is anywhere near likely.

While this is not likely in the NTD dimer conformations resolved by CryoEM, we have shown in previous studies and again here that GluA3 NTD dimers have considerable flexibility. Reference 52 that we cite there shows larger pockets that can be occupied by multiple drug fragments. The new simulations with chloride bound also illustrate this.

Response to Reviewer #1:

The authors have carefully considered the comments and provided substantial and important new data to support and extend their original study. In terms of my central criticisms:

1) Concerns and confusion about “Ultrafast”. The authors have clarified their meaning and provided impressive and informative new electrophysiological data. In terms of the 129/458 disulphide link, one alternative explanation for the altered kinetics observed could be a lower affinity of these receptors which would accelerate recovery kinetics due to fast unbinding of glutamate, and, if the peak current was ‘underestimated’ due to a slow risetime, could lead to large steady-state currents. No need for more experiments, but just confirmation that the risetimes of GluA3 and 129/458 are equivalent would confirm that the altered gating observed did not have an alternative explanation.

We thank the reviewer for this comment. We have now analysed the 20% to peak rise times for both GluA3 and the K129C/E458C mutant using outside-out patch recordings with 10 mM glutamate (1 ms pulses) (Fig. 1a). The rise time was slightly faster for GluA3-G (0.12 ± 0.04 ms, $n = 11$) compared to the mutant (0.15 ± 0.03 ms, $n = 6$), but the difference was not statistically significant (unpaired t test: $t=1.288$, $df=15$, $p=0.2172$)

To analyse this further, we also compared the peak current responses to sub-saturating (0.3 mM) and saturating (10 mM) glutamate for both receptors (Fig. 1b). If the K129C/E458C mutant had significantly reduced glutamate efficacy, we would expect a lower current ratio at sub-saturating glutamate. However, this was not the case; the similar rise times and comparable current ratios indicate no significant difference in glutamate affinity between the mutant and GluA3-G.

Figure 1. a) box plot showing the rise time constants of the first response to 100 Hz glutamate application (10 mM, 1 ms pulse) to outside-out patches containing GluA3-G ($n=11$ patches) and K129C/E458C ($n=6$ patches) receptors. b) Left panel: Representative whole-cell current traces evoked by 10 mM (black traces) and 0.3 mM (red traces) glutamate for GluA3-G/TARPy2 and K129C/E458C/TARPy2. Right panel: Bar plots showing the current amplitude ratios (0.3 mM / 10 mM) for GluA3-G/TARPy2 ($n=5$ cells) and K129C/E458C/TARPy2 ($n=5$ cells).

2) Mechanisms surrounding R/G 439. Cryo-EM data is now provided for GluA3-R, and some more electrophysiological data demonstrating the effects of this site on heteromerization. There is no doubt that this site is important for the trafficking (or lack of) of GluA3 as homomers, but there is still no

mechanism here – I do not agree that a shift of T_{au} of 5.4 to 6.8 ms for the R/G forms is informative in this regard. That said, the visualization of aggregated particles is an interesting observation, so regardless of exactly why the R-form behaves differently, there is at least this physical evidence to add to existing knowledge. It might be worth a brief comment concerning GluA3 homomers in the brain – do the authors think that GluA3 homomers ever get to the surface, or are all CP-GluA3-containing AMPARs, heteromers with GluA1 or GluA4?

We thank the reviewer for the comment. As pointed out, we provide further evidence that Arg439 contributes to increased aggregation of GluA3, and show that this involves organisation of the LBD tier. We agree that the observed minor difference in desensitization is not particularly informative in this context, but it was not intended to highlight functional differences between the two GluA3 variants. Indeed, electrophysiological recordings show that their current profiles are very similar.

We also demonstrate that GluA3-G forms homomeric GluA3 receptors, even when co-expressed with GluA2. By contrast, under comparable conditions, GluA3-R heteromerizes with GluA2. Contrasting with NMDA receptors and with high-affinity kainate receptor subunits (GluK4 and -5), AMPARs and low-affinity kainate subunits (GluK1-3) can form homomers in heterologous cells. But whether these exist as homomers in the brain, in specific regions and developmental stages, remains unclear and will depend on the set of core and auxiliary subunits in a given cell type. Current biochemical techniques lack the sensitivity and selectivity needed to reliably detect homomers, but native cryo-EM structures offer promise to fill this gap. As stated in our manuscript, the existence of GluA3 homomers will be relevant in specific brain regions, such as the auditory system and certain thalamic nuclei, as well as in diverse interneurons and glia, which are dominated by CP AMPARs. We have added this to the conclusion section.

3) Significance of desensitized LBD-NTD interface. The authors provide extra data in support of a functional role for this interface.

Thank you.

4) Analysis of disease associated mutations. More data has now been added including for the specific variants mentioned in the original review. The adaptations to Extended Figure 5 are appreciated to help map disease variants from the human form to the signal peptide-lacking structural annotations.

Thank you.

5) Reciprocal changes to GluA2. Again, some more useful experiments have been added.

All minor points were satisfactorily addressed, even though the lack of effect of DTT on 129/458 is surprising – it doesn't appear like the site of the cysteine bridge should be inaccessible to the reducing agent.

We agree with the reviewer, but can only add that multiple experimental attempts have failed to obtain a consistent effect of DTT.

Response to Reviewer #2:

The authors have adequately addressed my concerns and markedly improved the manuscript in the revised version. AMPA homomeric receptors can function as independent units, and the high-resolution structures of fast-conducting CP-AMPA receptors composed of GluA3 subunits reveal

unique features associated with calcium permeability and physiological function. I believe this work is of high quality and represents a valuable contribution to the iGluR field.

Thank you.